# TimeSpot: Benchmarking Geo-Temporal Understanding in Vision–Language Models in Real-World Settings

**Azmine Toushik Wasi** [* 1 2]   **Shahriyar Zaman Ridoy** [* 1 3]   **Koushik Ahamed Tonmoy** [3]   **Kinga Tshering** [3]
**S. M. Muhtasimul Hasan** [3]   **Wahid Faisal** [1 2]   **Tasnim Mohiuddin** [4]   **Md Rizwan Parvez** [4]

## Abstract

Geo-temporal understanding, the ability to infer location, time, and contextual properties from visual input alone, underpins applications such as disaster management, traffic planning, embodied navigation, world modeling, and geography education. Although recent vision–language models (VLMs) have advanced image geo-localization using cues like landmarks and road signs, their ability to reason about temporal signals and physically grounded spatial cues remains limited. To address this gap, we introduce TIMESPOT, a benchmark for evaluating real-world geo-temporal reasoning in VLMs. TIMESPOT comprises 1,455 ground-level images from 80 countries and requires structured prediction of temporal attributes (season, month, time of day, daylight phase) and geographic attributes (continent, country, climate zone, environment type, latitude–longitude) directly from visual evidence. It also includes spatial–temporal reasoning tasks that test physical plausibility under real-world uncertainty. Evaluations of state-of-the-art open- and closed-source VLMs show low performance, particularly for temporal inference. While supervised fine-tuning yields improvements, results remain insufficient, highlighting the need for new methods to achieve robust, physically grounded geo-temporal understanding. TIMESPOT is available at: https: //TimeSpot-GT.github.io.

---

[*]Equal contribution  [1]Computational Intelligence and Operations Laboratory (CIOL), Bangladesh [2]Shahjalal University of Science and Technology (SUST), Sylhet, Bangladesh [3]North South University (NSU), Dhaka, Bangladesh [4]Qatar Computing Research Institute (QCRI), Doha, Qatar. Correspondence to: Shahriyar Zaman Ridoy <shahriyar.zaman01@gmail.com>, Md Rizwan Parvez <mparvez@hbku.edu.qa>.

*Proceedings of the 43$^{rd}$ International Conference on Machine Learning*, Seoul, South Korea. PMLR 306, 2026. Copyright 2026 by the author(s).

## 1. Introduction

Determining *where* and *when* a photograph was captured using visual information alone is a fundamental human cognitive skill, underpinning situational awareness, episodic memory, and contextual reasoning. This capability requires the integration of diverse and often subtle visual cues, including illumination and shadow geometry, seasonal vegetation patterns, architectural styles and materials, clothing and traffic conventions, as well as broader geographic regularities in the natural and built environment (Lin et al., 2013; Workman et al., 2015; Arandjelovic et al., 2016; Hu et al., 2018; Hu & Lee, 2020). We refer to this integrated ability as geo-temporal understanding. Beyond its cognitive significance, geo-temporal reasoning is central to high-impact applications such as disaster response and recovery (Mirowski et al., 2018), environmental and climate monitoring (Zhai et al., 2017), autonomous navigation and localization (Lynen et al., 2020; Sarlin et al., 2019), and media forensics and verification (Tian et al., 2017).

In the geospatial domain, substantial progress has been achieved through cross-view and street-view localization benchmarks (Vo & Hays, 2016). Early work on ground-to-aerial matching has evolved into large-scale, geographically diverse datasets and unified embedding frameworks, including VIGOR (Zhu et al., 2021), GeoCLIP (Vivanco Cepeda et al., 2023), OpenStreetView 5M (Astruc et al., 2024), Global Streetscapes (Hou et al., 2024), CV-Cities (Huang et al., 2024), panoramic cross-view settings (Xia et al., 2025), and recent embedding advances (Cai et al., 2025). Despite this progress, existing benchmarks almost exclusively focus on where, typically measured via retrieval ranks or coordinate error (Hu & Lee, 2020), while largely ignoring when. Explicit temporal inference (e.g., season, month, or local time), as well as internal geo-temporal consistency (e.g., avoiding *July* paired with winter conditions in the Northern Hemisphere), is rarely required.

Along with geospatial reasoning, accurate temporal inference from visual input is critical for real-world systems such as disaster response, transportation planning, and environmental monitoring, where the same location can imply fundamentally different risks and actions depending on sea-

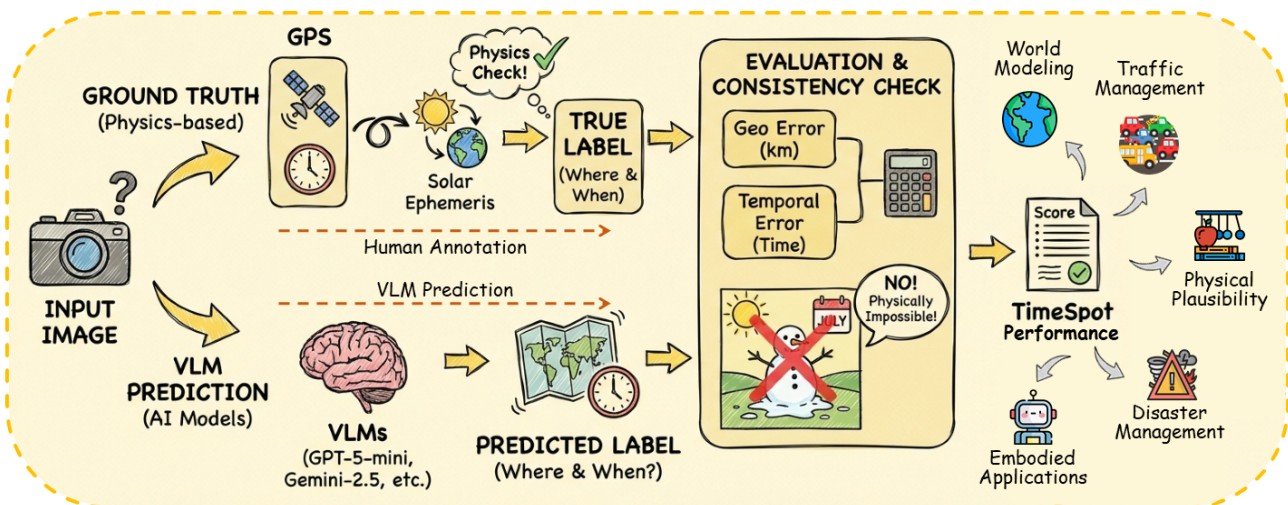

*Figure 1.* **TIMESPOT development and evaluation pipeline**, contrasting VLM predictions with expert-annotated ground truth for geo-temporal accuracy and physical consistency.

son, daylight phase, or time of day (Tehrani et al., 2026). From a world-modeling perspective, temporal reasoning underpins coherent environment representations for prediction, planning, and physical plausibility, and its absence has been shown to cause brittleness and physically implausible behavior in deployed vision and multimodal systems despite high spatial accuracy (Matta et al., 2025). Temporal information is essential across many domains, including complex navigation requiring temporal context, surveillance, visual event detection, embodied systems, and real-world decision support such as traffic management and disaster response.

However, when temporal information is considered in existing work, it is typically treated indirectly through proxy tasks such as basic navigation (Mirowski et al., 2018; Lynen et al., 2020), place recognition (Sarlin et al., 2019), or change detection (Sarlin et al., 2024), none of which require structured temporal outputs. Recent geospatial vision–language benchmarks in remote sensing, including HRVQA (Li et al., 2024b) and GEOBench-VLM (Danish et al., 2024), emphasize aerial imagery and focus on classification or segmentation rather than fine-grained temporal estimation or comprehensive geographic characterization from ground-level scenes. As a result, the field lacks a unified evaluation framework that probes joint geo-temporal reasoning, emphasizes subtle non-iconic cues over landmarks or text, and incorporates trustworthiness criteria such as schema validity, calibration, and robustness under distribution shift (Astruc et al., 2024; Hou et al., 2024; Huang et al., 2024; Xia et al., 2025).

This gap is consequential for four key reasons. **First**, temporal cues are often essential for disambiguation: solar geometry varies systematically with hemisphere (Ye et al., 2024), vegetation phenology separates climates at similar latitudes

(Wang et al., 2024c), and daylight phases shape urban lighting and activity patterns (Shi et al., 2020; Zhu et al., 2022). **Second**, real-world deployment requires *structured, verifiable predictions*, including minute-level precision for local time, kilometer-scale tolerance for coordinates, and explicit cross-field consistency checks (e.g., rejecting "snow in July" for a Northern Hemisphere prediction). Retrieval-centric protocols lack such safeguards, encouraging over-reliance on superficial or spurious cues. **Third**, robust global generalization demands *diversity-aware stress testing*. Hemisphere inversions, climate-region shifts, and out-of-distribution (OOD) splits expose brittle heuristics and dataset biases that iconic or text-dependent benchmarks fail to reveal (Astruc et al., 2024). **Fourth**, physically plausible world modeling requires coherent joint reasoning over time and space; without grounding in illumination and seasonal physics, models produce internally inconsistent inferences that undermine prediction and planning (Wu et al., 2025).

To address these limitations, we introduce TIMESPOT, a benchmark for evaluating joint geo-temporal understanding in vision–language models from natural, non-iconic images. TIMESPOT requires structured prediction over four temporal attributes and five geographic attributes, enabling fine-grained assessment beyond retrieval or coarse classification under uniform, metadata-free evaluation. Experiments across a diverse set of modern VLMs reveal systematically low temporal accuracy and frequent geo-temporal inconsistencies, highlighting the need for physically grounded geo-temporal benchmarks. Our core contributions are:

⇒ We introduce **TIMESPOT**, a diagnostic joint geo-temporal benchmark of 1,455 natural, non-iconic images across 80 countries, requiring structured prediction of four temporal and five geographic attributes. TIMESPOT

is verifiable (windowed temporal accuracy, geodesic distance), constrained, robust, and calibrated.

⇒ We perform a **rigorous evaluation** of state-of-the-art open- and closed-source VLMs under uniform, metadata-free, open-ended settings using precise temporal metrics, geodesic error, and consistency constraints, exposing substantial deficiencies in temporal grounding and joint reasoning.

⇒ We provide **comprehensive diagnostics** that reveal systematic spatial and temporal failure modes, including round-time anchoring, neighboring-country confusion, and geo-temporal inconsistency, and distill actionable directions for physically grounded reasoning.

⇒ We further conduct **supervised fine-tuning (SFT) as a diagnostic intervention** to assess whether explicit supervision improves geo-temporal understanding, analyzing gains, failures, and generalization trade-offs.

When evaluated on TIMESPOT, even the strongest VLMs achieve 77.59% country accuracy yet incur a median geodesic error of 892.54 km, while time-of-day accuracy peaks at only 33.74%, exposing severe deficiencies in joint geo-temporal reasoning. These results show that high coarse-grained localization can coexist with large metric and temporal errors, reflecting reliance on weak heuristics rather than physically grounded inference. By unifying spatial and temporal evaluation with calibration and consistency diagnostics, TIMESPOT establishes a foundation for trustworthy, real-world geo-temporal reasoning assessment.

## 2. Preliminaries and Related Work

TIMESPOT targets geo-temporal inference in *ground-level visual scenes*, where explicit textual cues and iconic landmarks are deliberately minimized and successful prediction depends on integrating weak, spatially distributed physical signals. Given an image, a model must produce a structured output schema comprising four temporal attributes: season, month, local time (HH:MM), and daylight phase, and five geographic attributes: continent, country, climate zone, environment type, and latitude–longitude coordinates. This formulation departs from retrieval-centric evaluation (Vivanco Cepeda et al., 2023; Astruc et al., 2024; Hou et al., 2024; Huang et al., 2024) by emphasizing *interpretable, verifiable, and physically grounded field-level predictions* that can be jointly audited for consistency.

**Relation to General Multimodal Benchmarks.** General-purpose multimodal benchmarks primarily evaluate broad perceptual and reasoning capabilities across images, text, charts, and diagrams, but provide limited assessment of *geo-temporal competence* (Table 1, 5). Benchmarks such as MMMU (Yue et al., 2024), M3Exam (Zhang et al., 2023), M4U (Wang et al., 2024a), MM-Vet (Yu et al., 2023), MME (Fu et al., 2023), MMBench (Liu et al., 2025), MM-

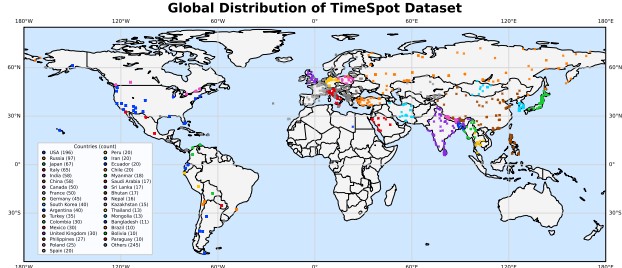

*Figure 2.* **Global coverage of TIMESPOT.** Locations of all 1,455 ground, level photos across 80 countries. Each marker denotes an image coordinate; colors indicate country (top contributors listed; "Others" aggregates the remainder).

STAR/Sphinx (Lin et al., 2023), SEED-Bench(Li et al., 2024a) span tasks from scientific reasoning to map and chart understanding, yet they neither require explicit prediction of when and where a scene occurs nor enforce cross-field geographic and temporal validity constraints. Similarly, foundational vision–language pretraining efforts, including CLIP (Radford et al., 2021), ViT (Dosovitskiy et al., 2020), ViLT (Kim et al., 2021), BLIP (Li et al., 2022), MG-P/SigLIP (Zhai et al., 2023), and Long-CLIP (Zhang et al., 2025), provide strong visual and textual priors but remain largely agnostic to the physical and ecological regularities governing temporal and geographic context.

**Geolocation and Spatial Reasoning Benchmarks.** Recent work has substantially advanced image-based geolocation using large vision–language models, but has focused predominantly on *spatial inference*. LLMGeo (Wang et al., 2024d) and IMAGEO-Bench (Li et al., 2025) benchmark country-, city-, or coordinate-level localization with structured prompting, while ETHAN (Liu et al., 2024b) and agent-based evaluations (Jay et al., 2025) demonstrate that chain-of-thought reasoning and navigation tools can improve spatial accuracy. FAIRLOCATOR (Huang et al., 2025) further exposes socio-economic and regional biases in spatial predictions. However, none of these benchmarks require explicit temporal prediction (e.g., month, local time, daylight phase) or enforce physical geo–temporal consistency constraints such as month–season–hemisphere alignment or time–daylight compatibility. TIMESPOT complements this line of work by assuming strong spatial recognition and explicitly probing whether models can *jointly infer time and place* from subtle physical cues while maintaining consistency across a nine-field geo-temporal schema. Our results show that models with competitive spatial accuracy often exhibit large temporal errors and frequent geo-temporal inconsistencies, revealing a critical and previously unmeasured failure mode. Additional related work is discussed in §A.

**Practical Significance of Temporal Reasoning.** Beyond

*Table 1.* **Comparison along TIMESPOT axes.** *Temporal*: temporal evaluation; *FineGeoG*: fine geo-graphics; *Subtle*: non-iconic cues; *HS/OOD*: hemisphere sanity or hard OOD; *FusionQs*: geo-temporal fusion tasks; *Schema*: structured outputs; *Calibration*: calibration/uncertainty; *Verifiable*: GPS/OSM-verifiable scoring; *Globality*: global coverage. Symbols: ✓ = yes; △ = partial; — = no.

| Benchmark / Dataset (year) | Temporal | FineGeoG | Subtle | HS/OOD | FusionQs | Schema | Calibration | Verifiable | Globality |
|---|---|---|---|---|---|---|---|---|---|
| LLMGeo (Wang et al., 2024d) | – | ✓ | △ | – | – | – | – | △ | ✓ |
| ETHAN (Liu et al., 2024b) | – | ✓ | △ | – | – | △ | – | △ | ✓ |
| Geo Inference (Jay et al., 2025) | – | ✓ | – | – | – | – | – | △ | ✓ |
| FAIRLOCATOR (Huang et al., 2025) | – | ✓ | △ | △ | – | – | – | △ | ✓ |
| IMAGEO-Bench (Li et al., 2025) | – | ✓ | △ | – | – | ✓ | △ | △ | ✓ |
| **TimeSpot (Ours, 2026)** | ✓ | ✓ | ✓ | ✓ | ✓ | ✓ | ✓ | ✓ | ✓ |

*Table 2.* **Dataset statistics.**

| Axis | Field | Categories (top shown) |
|---|---|---|
| Temporal | Season | Summer (400), Fall (399), Spring (335), Winter (321) |
| | Daylight phase | Afternoon (584), Night (287), Sunset (210), Morning (203), Midday (124), Sunrise (47) |
| | Month | 12 months represented; top: August (163), September (146), July (145), March (131) |
| | Hemispheric tag | Northern Hemisphere Summer (703), Northern Hemisphere Winter (615), Southern Hemisphere Winter (81), Southern Hemisphere Summer (56) |
| | Time coverage | Day (1182), Night (273) |
| | Hour range | Full 0–23; densest 08–18 |
| Geography | Continents | Asia (529), Europe (430), North America (326), South America (170) |
| | Countries | 80 unique; top: USA (196), Russia (97), Japan (67), Italy (65), China (58) |
| | Climate | Temperate (C) (582), Continental (D) (396), Tropical (A) (274), Arid (B) (180), Polar (E) (23) |
| | Environment type | Urban (648), Rural (202), Mountain (193), Coastal (181), Suburban (118), Desert (113) |
| | Lat/Lon span | lat -54.80 to 71.96, lon -173.24 to 170.31 |
| Cues | Primary temporal cues | Sun/Shadows (573), Vegetation (325), Other (289), Snow/Ice (122), Human Clothing (95), Agricultural Activity (51) |
| | Primary geolocation cues | Architecture (355), Natural Biome (354), Topography (Mountains/Coast) (295), Road Signage/Language (236), Vehicles (156), Other (58) |

cognitive interest, temporal inference directly affects systems where the same location implies different risks, operational states, or decisions depending on season, daylight phase, or time of day (Tehrani et al., 2026; Liao et al., 2025). Temporal misalignment has been identified as a source of physically implausible behavior in deployed vision systems even when spatial predictions are high (Matta et al., 2025). Our findings confirm this: VLMs frequently fail on temporal grounding even when spatial predictions are correct, reinforcing that geo-temporal reasoning is a prerequisite for trustworthy real-world deployment.

## 3. TimeSpot Benchmark

TIMESPOT is a benchmark for evaluating *geo-temporal reasoning* in vision–language models (VLMs) using natural, non-iconic, ground-level imagery. It targets inference from subtle physical cues, including illumination and shadow geometry, sky conditions, vegetation phenology, architectural

materials, and human activity, rather than from landmarks or textual artifacts. The benchmark comprises 1,455 images spanning 80 countries and requires prediction of a structured nine-field schema: four temporal attributes (season, month, local time (HH:MM), daylight phase) and five geographic attributes (continent, country, climate zone, environment type, latitude, longitude). By jointly evaluating spatial and temporal inference, TIMESPOT extends prior geolocation benchmarks beyond retrieval accuracy and coordinate error (Tian et al., 2017; Vo & Hays, 2016; Arandjelovic et al., 2016), and targets physically grounded reasoning under real-world uncertainty. Dataset statistics are reported in Table 2.

**Formal Definition.** Each image $x \in \mathcal{X}$ exposes observable cues such as illumination patterns, phenological signals, material and architectural styles, and traces of human activity. These cues map to a structured label $y = (y^{\text{temp}}, y^{\text{geo}})$, where $y^{\text{temp}} = (s, m, \tau, \phi)$ denotes season, month, local time, and daylight phase, and $y^{\text{geo}} = (C, \kappa, z, e, (\lambda, \varphi))$ denotes continent, country, climate zone, environment type, and coordinates. Given a fixed VLM, the task is to learn a mapping $f : \mathcal{X} \mapsto \widehat{\mathcal{Y}}$ that outputs predictions in the same schema. This formulation explicitly evaluates capabilities that generic VLMs (Radford et al., 2021; Li et al., 2022; Kim et al., 2021) and remote-sensing benchmarks do not directly test, namely joint reasoning over solar geometry, seasonal context, material cues, and geographic regularities. Geographic coverage is shown in Figure 2.

### 3.1. Benchmark Construction

TIMESPOT adopts a hybrid annotation strategy that combines deterministic programmatic labeling with structured human verification, enabling scale while preserving physical validity in ambiguous cases.
✓ *Image collection and curation.* Images are sourced from public web repositories and manual capture, with explicit suppression of landmark-centric and text-rich scenes. Curation prioritizes cases where inference depends on fine-grained physical evidence, such as seasonal vegetation changes, shadow-based solar estimation, or climate inference from the built and natural environment. Sampling is balanced across hemispheres, latitude bands, climate zones,

and environment types to ensure global coverage and reduce memorization shortcuts.

✓ *Annotation and Quality Control Process.* TIMESPOT targets objective physical quantities derived from metadata and geospatial databases rather than subjective semantic labels. Temporal attributes are computed from timestamps and solar ephemerides, and geographic attributes are mapped directly from coordinates. Human annotators act as verifiers, rejecting samples with corrupted metadata or irreconcilable visual evidence to ensure physically valid ground truth. Verification proceeds through context collection, image-level validation under formalized physical rules, expert adjudication of edge cases, and final constraint audits that produce proper annotation records.

✓ *Programmatic label derivation.* Ground-truth labels are derived deterministically from capture metadata and geographic priors. Months are taken from timestamps, seasons are assigned using meteorological definitions with hemisphere correction, daylight phases are computed from solar elevation using civil, nautical, and astronomical thresholds, and local time is derived from time zones and ephemerides. Climate zones follow the Köppen–Geiger classification (Peel et al., 2007), while continent, country, and coordinates are obtained directly from latitude and longitude, enforcing reproducibility and physical consistency across fields.

✓ *Human verification.* All samples undergo two-stage verification. Annotators first collect auxiliary geographic context and cross-check programmatic labels against visual cues such as shadows, illumination, vegetation, and weather. Senior annotators then adjudicate flagged cases, including twilight ambiguity and artificial lighting, by validating metadata against ephemeris calculations and scene consistency. The annotation team comprised 3 primary annotators and 2 senior annotators, all engineering graduates with geospatial domain experience; no crowdsourcing platforms were used. Primary annotation required approximately 576 hours across 6 weeks; senior adjudication of edge cases required an additional 30–40 hours.

✓ *Schema and normalization.* Annotations are stored in a canonical JSON schema. Model outputs are normalized for exact field-level scoring, including label canonicalization and signed decimal coordinates. Automated integrity checks enforce month–season–hemisphere alignment, daylight-phase compatibility with predicted time and location, and climate plausibility at $(lat, lon)$, yielding auditable error modes beyond retrieval failure.

## 4. Experiments and Evaluation

**Baselines.** We benchmark a diverse set of vision–language models (VLMs) to provide a comprehensive assessment of geo-temporal reasoning. The evaluated models fall into four families: *(i) proprietary* VLMs, including GPT-4o/mini, o3/o4-mini, Gemini-2/2.5-Flash, Claude 3.5 Haiku, and Mistral Medium (OpenAI, 2024; 2025; Team, 2025a; Anthropic, 2024); *(ii) open-source* VLMs spanning compact to large scales, including InternVL3, Qwen2.5-VL, Llama-3.2-Vision, Gemma-3, and GLM-4.5V (Zhu et al., 2025; Bai et al., 2025; Grattafiori et al., 2024; Gemma Team, Google DeepMind, 2025; Team et al., 2025); and *(iii) reasoning-augmented* variants that expose native *thinking* tokens while returning final structured predictions, such as o3/o4-mini, Gemini-2/2.5-Flash-Thinking, GLM-4.1V-Thinking, Kimi-VL-A3B-Thinking, and Step-3 (OpenAI, 2025; Team, 2025a; Team et al., 2025; Team, 2025b; Step-Fun, 2025). For open-source models, we further distinguish two parameter regimes: *small* ($\leq$11B) and *large* ($>$11B).

**Evaluation.** All proprietary model evaluations used the OpenRouter API; total inference cost was approximately 1,450 USD, with an estimated 400–500 million tokens across all runs (model pricing: 2–15 USD per million tokens depending on capability tier). We report categorical accuracy for continent, country, climate zone, and environment type; local-time accuracy within $\leq$1 hour and mean absolute error (MAE, minutes); coordinate MAE (degrees) and mean geodesic distance (MD, km) based on great-circle distance (Tian et al., 2017; Vo & Hays, 2016; Arandjelovic et al., 2016). To assess trustworthiness, we include explicit cross-field geo-temporal consistency diagnostics, calibration metrics such as expected calibration error and risk–coverage curves, and robustness analyses via stratification across continents, climate zones, and environment types, as well as hemisphere-flip tests and hard out-of-distribution splits (Zhu et al., 2021; Astruc et al., 2024; Hou et al., 2024; Huang et al., 2024). This evaluation isolates joint geo-temporal reasoning capabilities that generic VLM benchmarks (Radford et al., 2021; Kim et al., 2021; Li et al., 2022) and remote-sensing-focused suites such as GEOBench-VLM (Danish et al., 2024) do not explicitly test, namely calibrated and physically consistent inference of both time and place from subtle ground-level visual evidence. Formal metric definitions and scoring thresholds are provided in §D.2 and prompts in §E.

**Improving Visual Reasoning via Supervised Fine-Tuning (SFT).** As a diagnostic intervention, we investigate whether supervised fine-tuning improves geo-temporal reasoning on TIMESPOT. We fine-tune Qwen-VL2.5-3B-Instruct on country and time prediction tasks using a stratified 40% training split and evaluate on the remaining 60% (§H.1). The reduced training split is designed to test generalization rather than memorization of geographic distributions. While SFT improves both country and time prediction over the zero-shot baseline, cross-task evaluation reveals interference between spatial and temporal objectives under single-task LoRA adaptation. Joint SFT (§H.2) partially mitigates this trade-off but remains limited by competing gradient signals from illumination-invariant and illumination-sensitive cues.

*Table 3.* Performance of VLMs on **TIMESPOT** by questions. We bold and underline the best score within each model category. **Cnt.** → Continent, **Cou.** → Country, **Clim.** → Climate Zone; **Env.** → Environment Type, **Lat.°** → Latitude in degree, **Long.°** → Longitude in degree, **Dist.(km)** (MD) → mean distance from actual location in kilometers, **DLP** → Day-light phase. **Time** (Ac.) denotes accuracy, if the model predicted the time accurately within 1 hour window. **Time** (MAE) shows mean error in HH:MM format. Ac. denotes accuracy.

| Model | Geo-location Understanding | | | | | | | Temporal Understanding | | | | |
|---|---|---|---|---|---|---|---|---|---|---|---|---|
| | Cnt. | Cou. | Clim. | Env. | Lat.° | Long.° | Dist.(km) | Season | Month | Time | Time | DLP |
| | Ac.(↑) | Ac.(↑) | Ac.(↑) | Ac.(↑) | MAE (↓) | MAE (↓) | MD (↓) | Ac.(↑) | Ac.(↑) | Ac.(↑) | MAE (↓) | Ac.(↑) |
| *Proprietary Models* | | | | | | | | | | | | |
| GPT-4o-mini | 82.68 | 49.14 | 50.93 | 57.87 | 12.40 | 24.70 | 2827.07 | 47.08 | 22.34 | 30.32 | 3:54 | 31.55 |
| GPT-5-mini | 83.62 | 68.27 | **72.47** | 60.01 | 4.72 | 15.64 | 1389.79 | **58.43** | **34.27** | 21.55 | 4:10 | **44.60** |
| GPT-5.2 | 81.93 | 62.00 | 71.93 | 57.10 | 5.66 | 20.12 | 1770.00 | 47.17 | 19.17 | 29.35 | 3:42 | 41.66 |
| Gemini-2.0-Flash | 89.07 | 76.91 | 68.52 | 60.96 | 3.32 | 11.23 | 994.30 | 49.76 | 22.89 | 27.35 | 4:22 | 30.24 |
| Gemini-2.5-Flash | **90.51** | **77.25** | 71.34 | **64.32** | **3.05** | **10.38** | **917.61** | 50.92 | 23.91 | 25.15 | 3:56 | 41.92 |
| Claude 3.5 Haiku | 77.25 | 55.53 | 61.86 | 55.74 | 6.85 | 27.51 | 2269.86 | 44.12 | 19.04 | 23.09 | 4:14 | 30.93 |
| Mistral Medium 3.1 | 75.88 | 52.85 | 66.67 | 61.72 | 6.37 | 22.62 | 2045.61 | 36.84 | 15.26 | **30.73** | **3:36** | 36.01 |
| *Open-Source Models(≤11B)* | | | | | | | | | | | | |
| InternVL3.5-1B | 43.02 | 14.15 | 32.50 | 53.54 | 44.68 | 4378.92 | 7700.42 | 30.65 | 3.78 | 7.77 | 11:45 | 35.80 |
| InternVL3.5-2B | 60.00 | 29.41 | 51.82 | 57.80 | 13.11 | 43.71 | 3959.29 | 36.29 | 5.70 | 27.80 | 4:30 | 24.05 |
| Qwen-VL2.5-3B-Instruct | 22.40 | 13.47 | 18.83 | 44.53 | 16.18 | 130.98 | 8231.18 | 27.49 | 9.96 | 22.06 | 4:34 | 8.52 |
| InternVL3.5-4B | 60.79 | 30.12 | 57.77 | 56.74 | 15.34 | 44.15 | 4236.77 | 37.55 | 12.03 | **29.33** | 4:10 | 41.61 |
| Qwen-VL2.5-7B-Instruct | **85.70** | **73.96** | **70.86** | **75.21** | 32.94 | 21.46 | 4719.95 | **61.46** | **44.96** | 25.68 | **3:47** | **64.09** |
| Qwen3-8B-Instruct | 81.27 | 55.85 | 59.92 | 60.61 | 6.49 | **21.08** | **1897.80** | 43.18 | 15.15 | 21.14 | 4:32 | 32.51 |
| Llama-3.2-11B-Vision-Instruct | 74.22 | 55.73 | 57.12 | 57.61 | **5.85** | 26.57 | 2072.35 | 43.50 | 16.68 | 25.74 | 4:18 | 43.57 |
| *Open-Source Models (>11B)* | | | | | | | | | | | | |
| Gemma-3-27B-it | 79.59 | 54.02 | 60.41 | 53.12 | 6.83 | 23.58 | 2063.93 | 44.81 | 17.11 | 26.34 | 4:28 | 30.86 |
| Qwen-VL2.5-32B-Instruct | 78.56 | 57.11 | 62.95 | 60.82 | 6.27 | 24.02 | 2010.12 | 44.81 | 17.86 | **31.10** | 3:44 | **44.54** |
| Qwen3-32B-Instruct | 76.29 | 50.38 | 61.99 | 59.73 | 8.10 | 27.51 | 2423.00 | 42.27 | 16.56 | 23.73 | 4:03 | 43.09 |
| Internvl3-78b | 77.46 | 53.26 | **71.61** | 61.37 | 7.42 | 23.63 | 2180.29 | 45.91 | 16.43 | 29.64 | 4:07 | 34.91 |
| Qwen-VL2.5-72B-Instruct | 77.94 | 58.28 | 65.15 | 58.14 | **5.11** | 19.33 | 1711.42 | 44.47 | 18.28 | 28.71 | 4:00 | 36.84 |
| Llama-3.2-90B-Vision-Instruct | 78.08 | 53.54 | 63.85 | 59.04 | 7.05 | 26.79 | 2284.85 | 45.15 | **19.72** | 23.33 | 4:29 | 33.88 |
| GLM-4.5V-106B-MoE | **85.32** | **69.68** | 62.09 | **62.51** | **4.23** | **14.09** | **1280.87** | **57.55** | **36.04** | 30.51 | 4:09 | 42.45 |
| Qwen3-VL-235B-A22B-Instruct | 83.41 | 59.33 | 60.36 | 60.43 | 5.63 | 20.24 | 1778.00 | 45.56 | 19.48 | 26.22 | **3:36** | 41.91 |
| *Reasoning Models* | | | | | | | | | | | | |
| o4-mini | 82.39 | 71.82 | **73.06** | 66.64 | 4.85 | 15.39 | 1359.96 | **65.81** | **48.20** | 23.91 | 4:04 | **51.79** |
| Gemini-2-Flash-Thinking | 88.66 | 76.22 | 66.73 | 59.93 | 3.44 | 11.70 | 1024.14 | 49.28 | 22.68 | 27.49 | 4:22 | 29.76 |
| Gemini-2.5-Flash-Thinking | **90.31** | **77.59** | 66.47 | 64.47 | **3.04** | **9.85** | **892.54** | 51.13 | 24.26 | 22.19 | 4:03 | 36.56 |
| Kimi-VL-A3B-Thinking-2506 | 58.90 | 40.69 | 54.84 | 59.31 | 16.00 | 39.83 | 4034.15 | 39.72 | 12.65 | 32.23 | 4:18 | 25.70 |
| GLM-4.1V-9B-Thinking | 84.44 | 68.34 | 70.19 | **68.54** | 4.34 | 23.01 | 1788.77 | 58.02 | 38.88 | **33.74** | **3:58** | 47.76 |
| *Human Baselines* | | | | | | | | | | | | |
| Average (Undergrad) | 80.89 | 45.98 | 67.96 | 75.71 | 12.89 | 33.06 | 2800.49 | 68.89 | 28.39 | 41.92 | 2:41 | 77.89 |
| Average (Expert) | **94.06** | **67.89** | **86.39** | **87.39** | **5.16** | **12.22** | **1040.42** | **86.56** | **46.06** | **57.89** | **1:36** | **92.22** |

These findings motivate future RL-based or constraint-aware training strategies better aligned with physically grounded geo-temporal reasoning (Chen et al., 2025).

## 5. Results and Analysis

### 5.1. Performance Analysis

**Overall Performance.** Results in Table 3 show a clear advantage for proprietary models on spatial reasoning and metric localization. Among non-*Thinking* models, *Gemini–2.5–Flash* achieves the strongest overall performance, with continent accuracy 90.51%, country 77.25%, climate 71.34%, environment 64.32%, and the lowest median distance error (MD) of 917.61 km (latitude MAE 3.05°, longitude MAE 10.38°). Its reasoning-enhanced variant, *Flash–Thinking*, further improves coordinate precision (latitude MAE 3.04°, longitude MAE 9.85°, MD 892.54 km) while preserving

high country-level accuracy (77.59%).

**Open-source Models Performance.** Open-source models exhibit greater variance. *GLM–4.5V–106B–MoE* attains competitive country accuracy (69.68%) with an MD of 1280.87 km, indicating strong place recognition but weaker metric grounding. In contrast, *Qwen–VL2.5–7B–Instruct* performs well on categorical geography (continent 85.70%, country 73.96%, environment 75.21%) yet fails on coordinate estimation (latitude MAE 32.94°, MD 4719.95 km), revealing a pronounced gap between semantic place classification and precise spatial localization. Temporal inference remains a major bottleneck. Across all model families, time-of-day accuracy is low (typically 22–34%), with mean absolute errors of approximately four hours (3:36–4:30). Performance varies by temporal subtask: *o4–mini* achieves the highest calendar accuracy (season 65.81%, month 48.20%), *GLM–4.1V–9B–Thinking* leads on time-of-day prediction

*Table 4.* Consistency-violation and diagnostic rates across models on TIMESPOT. Lower is better. **Month & Season** reports the fraction of predictions where the predicted month is inconsistent with the predicted season under hemisphere rules; **Season & Month** reverses the dependency to test the inverse logical consistency.

| Model | Phase & Time (>1 h) (%) | Month & Season (%) | Season & Month (%) | Country & MD > 200 km (%) | Country & MD < 200 km (%) | Continent & Country (%) | MD > 1000 km (%) |
|---|---|---|---|---|---|---|---|
| Gpt5-Mini | 15.95 | 0.89 | 25.02 | 16.98 | 2.54 | 17.59 | 17.25 |
| intern_vl3_78B | 11.82 | 0.62 | 30.10 | 27.42 | 3.85 | 29.00 | 37.73 |
| QwenVL-3B | 0.21 | 0.82 | 18.35 | 12.78 | 0.00 | 8.93 | 95.19 |

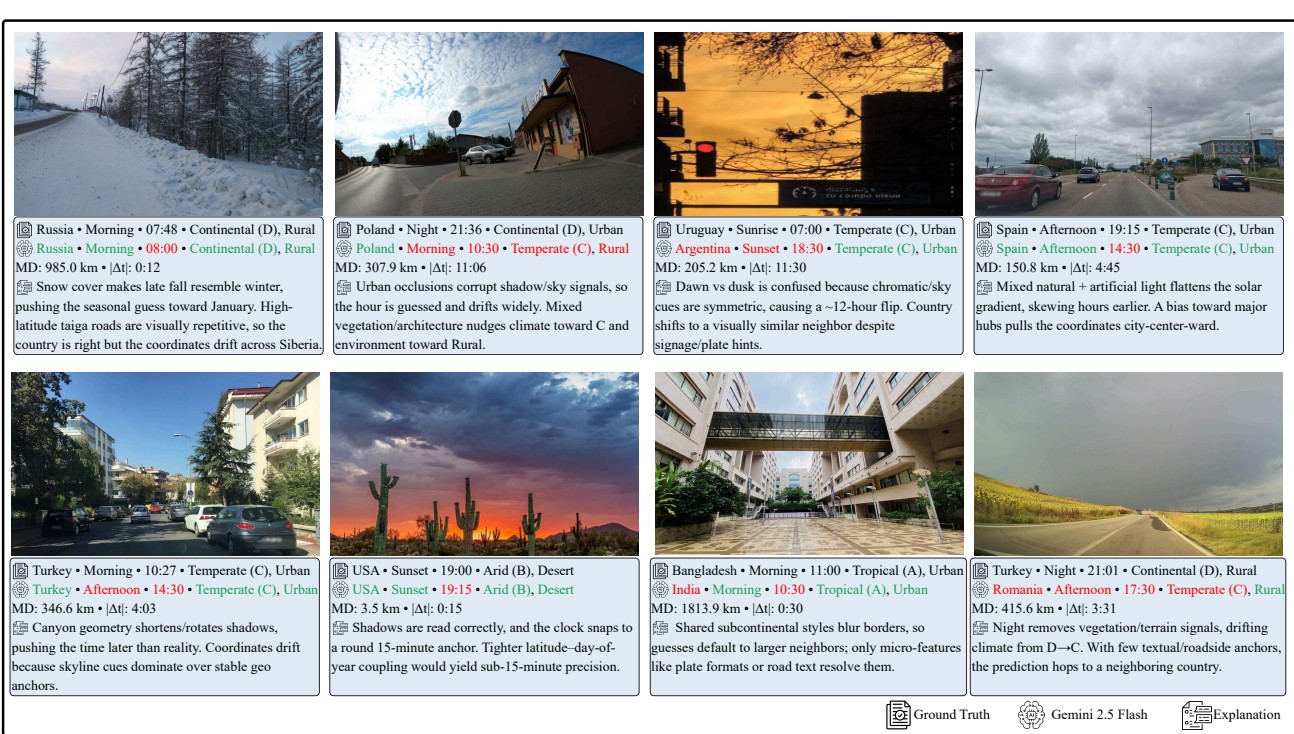

*Figure 3.* **Qualitative results on TIMESPOT (Gemini-2.5-Flash).** Each panel pairs the image with ground truth and model outputs, country, daylight phase, and local time, along with MD and $|\Delta t|$.

(33.74%), and *Qwen–VL2.5–7B–Instruct* performs best on daylight phase (64.09%). These results indicate that models can often infer coarse illumination states (e.g., day vs. dusk) but struggle to recover precise local time from subtle visual cues such as solar elevation, shadow geometry, sky luminance gradients, and lighting.

**Human performance baseline.** To establish empirical difficulty bounds for the benchmark tasks, we conducted a controlled human evaluation involving two groups given only the image (no metadata): average participants (undergraduates) and domain experts (geology and geography graduates). Expert participants achieved a Time MAE of 1:36 and absolute Time Accuracy (within 1 hour) of 57.89%, Season accuracy of 86.56%, Country accuracy of 67.89%, Climate accuracy of 86.39%, and Daylight Phase accuracy of 92.22%. Average participants scored lower across all fields (Time MAE: 2:41, Season: 68.89%, Country: 45.98%). These results confirm that (i) the 1-hour time accuracy threshold is empirically achievable by domain experts reasoning from physical cues alone; (ii) VLMs substantially lag human experts on all temporal fields (best VLM Time MAE: 3:36 vs. expert: 1:36); and (iii) the best VLMs match or exceed both human groups on coarse spatial memorization (e.g., Gemini-2.5-Flash country accuracy: 77.25% vs. expert: 67.89%), while lacking the continuous 4D physical-world understanding that underlies expert temporal inference. Full human baseline results are reported in Table 25.

**Impact of *Reasoning*.** As per Table 3, reasoning-oriented models consistently outperform their non-reasoning counterparts across both geo-location and temporal tasks, indicating a clear benefit from explicit multi-step inference. For example, Gemini-2.5-Flash-Thinking improves country accuracy to 77.59% (vs. 77.25% for Gemini-2.5-Flash) and reduces mean distance to 892 km, while o4-mini achieves strong gains in season (65.81%) and daylight phase accu-

racy (51.79%), surpassing most proprietary baselines. These improvements suggest that reasoning models better integrate weak, low-salience cues (e.g., illumination, seasonal context, spatial consistency), yielding more physically consistent and robust geo-temporal predictions than standard instruction-following VLMs.

**Geographic Understanding Overview.** Across continents (§F.5), model performance is driven primarily by data density, visual regularity, and cue stability rather than model scale alone. Europe and North America exhibit the highest and most stable accuracy, reflecting dense pretraining exposure, standardized urban infrastructure, and uniform signage systems. Asia shows the greatest variance, with strong performance in coastal and urban regions but sharp failures in inland, mountainous, and high-altitude areas where terrain and illumination vary widely. South America displays a pronounced coastal advantage, while Andean and inland regions remain challenging due to high elevation, heterogeneous lighting, and mixed land cover. Across all continents, island nations frequently reach ceiling accuracy, whereas landlocked or climatically complex regions expose brittle generalization, and large parameter counts do not ensure robustness. These continent-level disparities motivate TimeSpot as a benchmark that explicitly probes fine-grained, physically grounded geo-temporal reasoning beyond coarse geographic heuristics.

**Temporal Reasoning Overview**. Across temporal attributes, we observe systematic and complementary failure modes. Climate zone prediction (§F.3) is strongest in Tropical, Arid, and especially Temperate regions, but degrades sharply in Continental and Polar climates, revealing strong mid-latitude and low-elevation biases and limited robustness to extreme seasonality and illumination physics. Seasonal inference (§F.2) is reliable in summer but degrades in spring and winter due to transitional phenology and regional variability, while autumn collapses entirely across all models, exposing a fundamental weakness in vegetation- and color-based seasonal reasoning not captured by daylight cues alone. Daylight phase prediction (§F.1) shows strong phase-specific specialization but poor robustness across the diurnal cycle, with night and sunrise remaining particularly challenging. Together, these results indicate limited temporal grounding, as models rely on isolated appearance cues rather than coherent reasoning about solar dynamics, phenology, and long-term environmental change.

**Consistency Overview**. Table 4 shows that even strong models exhibit frequent geo-temporal consistency violations, with substantial rates of phase–time mismatch, season–month inconsistency, and extreme spatial errors. Notably, low inconsistency in one dimension (e.g., phase–time for QwenVL-3B) does not preclude severe failures elsewhere (e.g., MD>1000km), revealing that isolated accuracy

gains do not translate into coherent, physically consistent geo-temporal reasoning.

**Calibration**. We assess confidence reliability using Expected Calibration Error (ECE) and risk–coverage curves (§D.1). ECE increases with task granularity, from as low as 0.021–0.042 for continent prediction to 0.038–0.063 for daylight phase in proprietary models, and up to 0.095–0.110 in open-source models. While proprietary systems are consistently better calibrated, all models exhibit systematic overconfidence on fine-grained temporal tasks. It shows that reliable geo-temporal deployment requires not only higher accuracy but also improved confidence calibration, particularly for temporally ambiguous settings.

**Performance Deviation in Different Cues**. Cue-conditioned analysis shows that models perform best when high-salience human-centric cues are present: for example, *GPT-5-mini* reaches 87.6% continent and 79.2% country accuracy with architectural cues, yet achieves only 47.0% accuracy within 200 km for coordinates (Table 15). Performance degrades substantially for natural biome and topographic cues, where coordinate accuracy drops below 40% even for strong models and to near-zero for smaller ones, revealing weak physical grounding. Temporal cue–conditioned analysis (Table 16) shows that salient signals such as sun/shadows and snow/ice substantially improve coarse temporal inference but do not yield reliable fine-grained time estimation. For example, *GPT-5-mini* achieves up to 60.5% season accuracy from sun/shadow cues and 69.4% from snow/ice, yet time-within-1h accuracy remains below 25% across all cues. This gap suggests that models primarily utilize temporal cues as semantic correlates, rather than grounding them in physically consistent reasoning about solar dynamics and clock time. More detailed analysis is provided in §F.6.

### 5.2. Error Analysis

**Summary of Failure Modes.** Across TimeSpot (1,455 images), we observe consistent breakdowns in fine-grained geo-temporal reasoning (full diagnostics in §C). High continent-level accuracy frequently coexists with sharp drops at the country level, indicating limited use of micro-geographic cues. Temporal inference is particularly fragile: minute-level time prediction collapses toward common hour anchors despite moderate MAE, while models with similar mean geodesic distance can differ substantially in the mass of extreme errors (MD>1000 km), which dominate unusable predictions. We also observe persistent cross-field inconsistencies between daylight phase, local time, and longitude, alongside cue imbalance in which daylight phase is inferred more reliably than clock time and climate or environment predictions default toward frequent classes such as Temperate and Urban.

**Qualitative and Quantitative Patterns.** Figure 3 illustrates representative successes and failures in geo-temporal reasoning on TIMESPOT. When salient physical cues are present, such as clean solar geometry in open or arid landscapes, models align closely with ground truth (e.g., desert sunset with $|\Delta t| = 0{:}15$ and MD$= 3.5\,\text{km}$), indicating effective use of shadow direction and sky color. In contrast, low or ambiguous illumination severely degrades temporal accuracy: at night, predictions collapse to popular evening anchors (e.g., 20:30), while at dawn or dusk, symmetric chromatic cues induce Sunrise–Sunset confusions that yield large time errors (e.g., Uruguay→Argentina with $|\Delta t| \approx 11\,\text{h}$) despite limited spatial drift. Urban street scenes expose a second failure mode, where occlusions and canyon geometry distort shadow cues and compress apparent sun elevation, shifting predicted times later than reality (e.g., Turkey morning misclassified as afternoon with $|\Delta t| \approx 4\,\text{h}$). Spatially, models frequently identify the correct continent but substitute a neighboring country (e.g., Bangladesh→India), reflecting reliance on broad stylistic features rather than micro-geographic signals such as signage typography, license plates, utility hardware, or road markings. Additional errors include climate and environment drift under low light and systematic underuse of coastal or elevation cues, where visible shorelines are ignored and predictions move inland, producing large geodesic errors. Overall, these examples reveal a clear divide between scenes governed by strong physical constraints and those requiring integration of subtle, distributed cues, motivating explicit solar-geometry modeling, phase-aware temporal heads, and stronger geo- and topographic priors.

### 5.3. Dataset Stability

To assess dataset stability in low-frequency areas, we analyze frequency-weighted country accuracy with rank-stability tests (§G). Our analysis reveal that, across all models, accuracy increases monotonically with per-country sample size, while relative model rankings remain consistent across low-, medium-, and high-resource regimes. These statistical tests confirm that observed trends are not driven by isolated samples or dominant countries, demonstrating that TIMESPOT yields a stable and robust evaluation despite uneven geographic distributions.

### 5.4. Performance Improvement with SFT

Figure 4 reports supervised fine-tuning results on TIMESPOT (details in §H.1). Fine-tuning *Qwen2.5-VL-3B-Instruct* on a stratified 40% subset improves both *country accuracy* (14.20% → 19.24% by epoch 4) and *time accuracy* (20.27% → 24.79% by epoch 5), showing that SFT strengthens geo-semantic prediction and temporal grounding. However, country accuracy saturates early, while time accuracy fluctuates across epochs, indicating less stable

learning of fine-grained temporal cues. Cross-task evaluation reveals feature interference: country-tuned models reduce time accuracy (22.06% → 21.78%), while time-tuned models reduce country accuracy (13.47% → 12.98%). This reflects competing visual demands: country prediction relies on illumination-invariant structural cues, whereas time prediction depends on illumination-sensitive features such as shadows and sky luminance. Joint SFT partially mitigates this trade-off (country: 14.23% → 15.72%; time: 20.27% → 22.36%), but remains below single-task peaks, suggesting gradient conflict under shared LoRA parameters. These findings motivate training methods that better separate or balance spatial and temporal gradient signals (Buschoff et al., 2025; Binz et al., 2025; Chen et al., 2025).

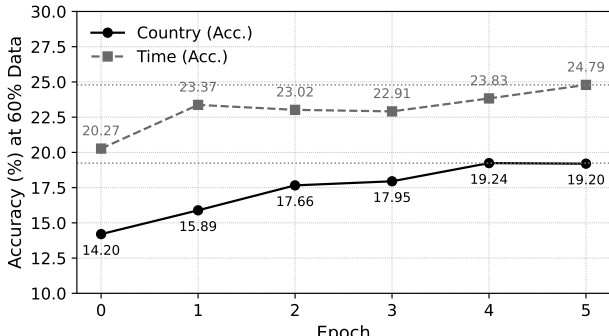

*Figure 4.* SFT performance trends over epochs.

## 6. Concluding Remarks

Geo-temporal reasoning remains a major challenge for vision–language models in unconstrained, real-world images. In this work, we show that joint geo-temporal reasoning remains a core unresolved challenge for vision–language models operating on unconstrained real-world images. We introduce TIMESPOT, a benchmark of 1,455 ground-level images from 80 countries that requires explicit prediction of *when* and *where* a scene occurs through structured temporal and geographic attributes. Our analysis exposes failure modes that are invisible to spatial-only or retrieval-based benchmarks. Empirically, even the strongest models exhibit large geodesic errors and low time-of-day accuracy, with frequent violations of solar, hemispheric, and daylight consistency. These errors reveal over-reliance on coarse spatial priors and weak grounding in physical cues such as illumination, shadows, and seasonal dynamics. Importantly, temporal reasoning is not ancillary but essential for coherent world modeling, physically plausible inference, and downstream decision-making in real-world systems. Future works should prioritize explicit geo-temporal inductive biases, constraint-aware reasoning, and supervision that couples spatial context with temporal physics and dynamics.

## Impact Statement

This paper introduces TimeSpot, a diagnostic benchmark for evaluating geo-temporal reasoning in vision–language models. The primary goal of this work is to advance the scientific understanding of how current models infer time and place from visual input, and to expose limitations that affect reliability, calibration, and physical consistency.

By highlighting systematic failure modes in temporal grounding and joint geo-temporal reasoning, this benchmark can support the development of more robust and trustworthy models for applications such as environmental monitoring, disaster response, navigation, and context-aware decision support. At the same time, we recognize that geo-temporal inference capabilities may raise concerns if misused for surveillance or intrusive location tracking. TimeSpot mitigates these risks by relying exclusively on publicly available, non-sensitive imagery, avoiding personally identifiable information, and framing evaluation around diagnostic analysis rather than deployment.

A central contribution of TimeSpot is its explicit focus on temporal reasoning from visual input, including season, month, daylight phase, and fine-grained local time. Temporal inference is particularly sensitive because it requires models to reason about physical processes such as solar geometry, illumination continuity, and seasonal cycles, rather than relying on static semantic cues alone. These capabilities are essential across many practical domains, including navigation systems that depend on temporal context, surveillance and visual event detection, embodied agents operating in dynamic environments, and real-world decision support such as traffic management and disaster response. By exposing systematic failures in temporal understanding, TimeSpot aims to prevent the uncritical deployment of models that may appear spatially competent but lack physically grounded understanding of *when* an image was captured. We view this diagnostic emphasis on time as central to safe world modeling, since temporal inconsistencies can propagate downstream errors in high-stakes, real-world systems.

Overall, we believe this work contributes positively to responsible machine learning by encouraging transparency, rigorous evaluation, and awareness of model limitations, thereby helping prevent overconfident or unsafe real-world use of vision–language systems.

## Ethics Statement

TimeSpot is designed to study geo-temporal reasoning in everyday, publicly observable scenes and to benchmark vision–language models using structured, verifiable outputs. Images were sourced under licenses permitting research use or captured by the authors; items with unclear rights were excluded, and license and attribution metadata are recorded in the release. To mitigate privacy risks, we remove embedded metadata, blur or mask personally identifying details (e.g., faces, license plates, house numbers), and exclude scenes dominated by sensitive content. Geographic annotations are limited to regional granularity, and exact dwelling locations are never provided.

Annotation guidelines prohibit demographic inference or stereotyping, and dataset composition was audited for geographic and environmental balance to reduce bias. Annotators were trained, fairly compensated, and allowed to decline any image. We acknowledge potential dual-use risks in location inference; accordingly, the dataset license restricts use to non-commercial research, prohibits identification of individuals or private property, and documents known failure modes to discourage unsafe deployment.

## Software and Data

All equations, hyperparameters, models and data resources required for reproducibility are fully specified in the paper. TimeSpot is available at: https://TimeSpot-GT.github.io.

## Conflict of Interest Disclosure

The authors declare that they have no financial or other substantive conflicts of interest that could reasonably be perceived to influence this work.

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

# A. Extended Related Work

**Remote sensing and aerial geospatial VLMs.** A large body of recent work on geospatial vision–language models focuses on aerial and satellite imagery, advancing captioning, visual question answering, detection, and change analysis at planetary scale. Benchmarks such as EarthVQA (Wang et al., 2024b), RS-LLaVA (Bazi et al., 2024), RSBench/VRSBench (Li et al., 2024c), GeoChat (Kuckreja et al., 2024), RemoteCLIP (Liu et al., 2024a), RS5M/GeoRSCLIP (Zhang et al., 2024), and HRVQA (Li et al., 2024b) evaluate multi-modal reasoning and perception over remote-sensing imagery, while GEOBench-VLM (Danish et al., 2024) aggregates diverse geospatial tasks, including non-optical data and segmentation. Although these efforts provide valuable coverage for Earth observation, they primarily emphasize aerial viewpoints and do not require models to jointly reason about fine-grained temporal attributes (e.g., season, month, time, daylight phase) together with ground-level geographic context, where physical cues are weaker, noisier, and more ambiguous.

**Ground-level localization and cross-view benchmarks.** At ground level, cross-view localization and place-recognition research has developed powerful spatial representations through retrieval and matching pipelines. Early ground-to-aerial methods (Lin et al., 2013; Workman et al., 2015; Tian et al., 2017) and NetVLAD-style aggregation (Arandjelovic et al., 2016) laid the foundations for metric localization, followed by larger and more geographically diverse benchmarks such as CVM/Net (Hu et al., 2018; Hu & Lee, 2020). Subsequent datasets, including VIGOR (Zhu et al., 2021), OpenStreetView 5M (Astruc et al., 2024), Global Streetscapes (Hou et al., 2024), CV-Cities (Huang et al., 2024), and panoramic cross-view settings (Xia et al., 2025), stress robustness to viewpoint, appearance, and domain gaps across continents. However, evaluation in these benchmarks is almost exclusively spatial, focusing on retrieval accuracy or coordinate error (Lin et al., 2013; Workman et al., 2015), and does not require calibrated predictions of *when* an image was captured, nor the validation of physical constraints such as month–season–hemisphere alignment or daylight plausibility (Hu & Lee, 2020).

**Image-based geolocation with large VLMs.** More recently, large vision–language models have significantly advanced ground-level image geolocation, but remain largely focused on *spatial inference*. LLMGeo (Wang et al., 2024d) and IMAGEO-Bench (Li et al., 2025) evaluate country-, city-, or coordinate-level localization with sophisticated prompting and structured reasoning protocols. ETHAN (Liu et al., 2024b) and subsequent agent-based evaluations (Jay et al., 2025) show that chain-of-thought reasoning and navigation tools can substantially improve spatial accuracy. FAIRLOCATOR (Huang et al., 2025) introduces a complementary axis by revealing socio-economic and regional biases in geolocation performance. Despite their breadth and rigor, these benchmarks do not require explicit prediction of temporal attributes (e.g., month, local time, daylight phase), nor do they enforce cross-field physical consistency constraints such as month–season–hemisphere or time–daylight compatibility.

**Positioning of TIMESPOT.** TIMESPOT is complementary to this line of work. Rather than re-evaluating whether modern VLMs can localize images spatially, it assumes strong spatial priors and instead probes whether models can *jointly infer where and when* from subtle, low-salience physical cues present in natural ground-level imagery, as illustrated in Figure 5. By requiring structured prediction over a nine-field geo–temporal schema and auditing internal consistency across geographic and temporal attributes, TIMESPOT exposes failure modes that are invisible to spatial-only benchmarks. Our results show that even state-of-the-art models with competitive geolocation accuracy exhibit large temporal errors and frequent geo–temporal inconsistencies, highlighting a critical gap in current evaluation protocols and a clear opportunity for future model and benchmark development.

# B. Details on TimeSpot Development

**Evaluation and Trustworthiness.** TIMESPOT evaluates categorical fields using top-1 accuracy, local time using minute-window accuracy and mean absolute error (MAE), and geographic coordinates using great-circle distance (mean, median, and thresholded ranges), following established geolocation protocols (Tian et al., 2017; Vo & Hays, 2016; Arandjelovic et al., 2016) while extending them to explicitly temporal prediction. To encourage *trustworthy inference*, we incorporate cross-field consistency constraints and diagnostics, including hemisphere–season agreement, daylight-phase compatibility with predicted time and location, and climate plausibility at $(lat, lon)$, building on practices from large-scale cross-view and geolocation benchmarks (Zhu et al., 2021; Astruc et al., 2024; Hou et al., 2024). We further report calibration metrics, including expected calibration error (ECE) and risk–coverage curves, to assess confidence reliability in multi-field outputs, addressing failure modes observed in prior cross-view and place-recognition systems (Lin et al., 2013; Workman et al., 2015; Hu & Lee, 2020), where predictions can be confident yet physically inconsistent.

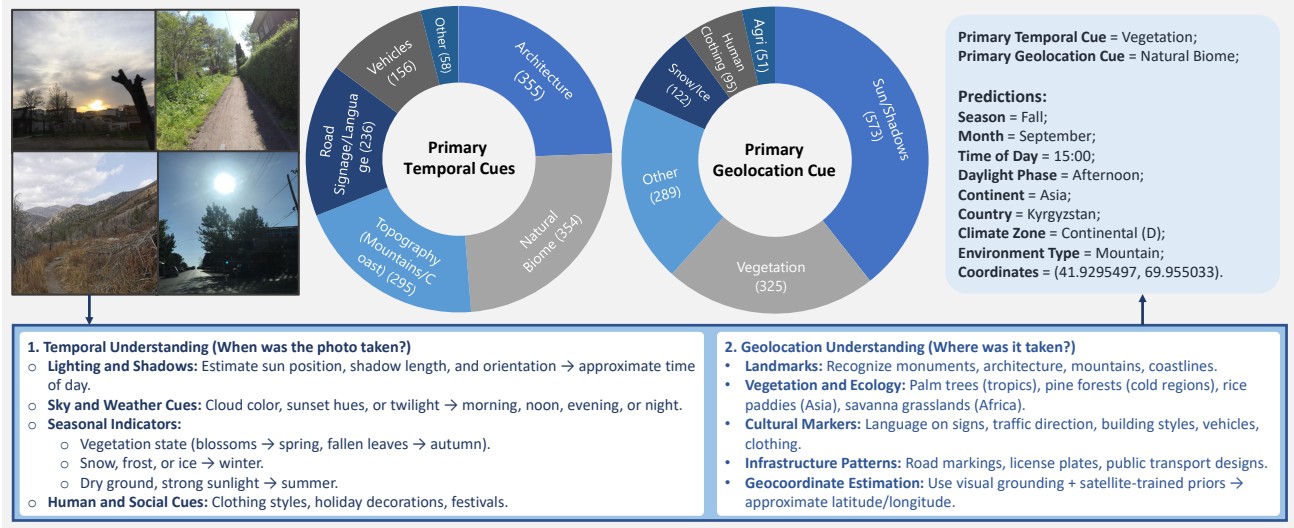

*Figure 5.* **Illustration of the TIMESPOT benchmark for geo-temporal understanding**. Models must infer temporal attributes (season, month, time of day, daylight phase) and geographic attributes (continent, country, climate zone, environment type, coordinates) directly from visual input. Left: example images. Center: distributions of primary temporal cues (e.g., architecture, natural biome, topography) and geolocation cues (e.g., sun/shadows, vegetation, snow/ice). Right: an example prediction, highlighting the integration of diverse and subtle cues required for reliable reasoning.

**Generalization and Robustness.** To evaluate robustness, TIMESPOT provides stratified analysis across continents, climate zones, and environment types, alongside *hemisphere-flip* tests and hard out-of-distribution (OOD) splits that suppress landmark shortcuts and amplify reliance on physical and ecological cues (Astruc et al., 2024; Hou et al., 2024; Huang et al., 2024; Xia et al., 2025). The benchmark also includes fusion questions that require coherent integration of spatial and temporal evidence, exposing independence assumptions that commonly arise in modular VLM decoders (Lin et al., 2013; Hu et al., 2018; Zhu et al., 2021). Relative to generic multimodal evaluations and remote-sensing-focused geospatial suites, TIMESPOT occupies a distinct and currently missing niche: a *ground-level*, *joint geo-temporal* benchmark with structured outputs, internal consistency checks, calibration analysis, and systematic robustness diagnostics. Empirically, we observe that state-of-the-art open and closed VLMs, despite strong performance on broad multimodal benchmarks (Vivanco Cepeda et al., 2023) and cross-view geolocation tasks (Lin et al., 2013; Workman et al., 2015; Hu & Lee, 2020), exhibit low temporal accuracy and frequent geo-temporal inconsistencies, highlighting substantial remaining headroom for physically grounded visual reasoning.

## C. More Details on Error Analysis

**Metrics and diagnostics.** Beyond per-field accuracy, we analyze error structure using complementary diagnostics that explicitly probe spatial precision, temporal fidelity, and cross-field consistency. In addition to categorical accuracy for discrete fields, we report local-time accuracy within a $\leq 1$-hour tolerance alongside mean absolute error (MAE in minutes), and geographic accuracy using mean and median great-circle distance (MD). To capture practically consequential failures, we introduce targeted indicators for physically implausible predictions, including Continent✓ & Country✗ errors, Country✗ with MD<200 km (boundary confusions), Phase✓ with $|\Delta t| > 120$ min (daylight–time inconsistency), and extreme spatial outliers (MD>1000 km). These diagnostics isolate failure modes that are obscured by aggregate metrics but dominate downstream risk. We further compute correlations between temporal error and spatial error to assess whether mistakes in time and place co-occur or arise independently.

**Spatial error structure.** Spatial errors are dominated by *near-miss geography*, where predictions fall within the correct continent or regional cluster but cross national boundaries. These failures frequently occur between visually similar neighboring countries, suggesting reliance on coarse regional cues rather than micro-geographic signals. Examples include Bangladesh–India, Uruguay–Argentina, and inland–coastal confusions in Southern Europe. Although such errors may yield moderate MD, they are semantically severe for country-level tasks. A second class of spatial failures exhibits heavy-tailed behavior, with errors exceeding 1000 km in scenes lacking distinctive architectural, linguistic, or ecological markers. These

*Table 5.* **Extended Comparison along TIMESPOT axes.** *Temp*: Season/Month/Time/Daylight phase. *FineGeo*: Continent/Country/Climate/Environment/Lat–Lon. *Subtle*: non-iconic cue emphasis. *HS/OOD*: hemisphere sanity or hard/OOD splits. *FusionQs*: geo-temporal fusion tasks. *Schema*: structured field outputs. *Calib*: calibration/uncertainty metrics. *Verif*: GPS/OSM-verifiable scoring. *Globality*: multi-continental coverage. Symbols: ✓ = explicit support; △ = partial/limited; — = not present.

| Benchmark / Dataset (year) | Temp | FineGeo | Subtle | HS/OOD | FusionQs | Schema | Calib | Verif | Globality |
|---|---|---|---|---|---|---|---|---|---|
| OpenStreetView–5M (Astruc et al., 2024) | – | △ | △ | ✓ | – | – | – | ✓ | ✓ |
| Global Streetscapes (Hou et al., 2024) | △ | △ | – | △ | – | – | – | ✓ | ✓ |
| CV–Cities (Huang et al., 2024) | – | △ | – | ✓ | – | – | – | ✓ | ✓ |
| VIGOR (Zhu et al., 2021) | – | △ | – | ✓ | – | – | – | ✓ | △ |
| CVACT (Liu & Li, 2019) | – | △ | – | ✓ | – | – | – | ✓ | △ |
| CVUSA (Workman et al., 2015) | – | △ | – | ✓ | – | – | – | ✓ | – |
| University–1652 (Zheng et al., 2020) | – | △ | – | △ | – | – | – | ✓ | – |
| GeoText–1652 (Chu et al., 2024) | – | △ | △ | ✓ | – | △ | – | ✓ | – |
| GeoCLIP Eval (Vivanco Cepeda et al., 2023) | – | △ | – | ✓ | – | – | – | ✓ | ✓ |
| Panoramic Cross–View (Xia et al., 2025) | – | △ | – | ✓ | – | – | – | ✓ | ✓ |
| HRVQA (Li et al., 2024b) | – | – | – | △ | △ | – | – | – | ✓ |
| VRSBench (Li et al., 2024c) | – | – | – | ✓ | △ | – | – | – | ✓ |
| GEOBench–VLM (Danish et al., 2024) | △ | – | – | ✓ | △ | △ | – | – | ✓ |
| EarthVQA (Wang et al., 2024b) | – | – | – | △ | △ | – | – | – | ✓ |
| FIT–RSFG / RSRC (Luo et al., 2024) | – | – | – | △ | △ | △ | – | – | ✓ |
| RemoteCount (Liu et al., 2024a) | – | – | – | △ | △ | – | – | – | ✓ |
| SkyEyeGPT / SkyBench (Zhan et al., 2025) | – | – | – | △ | △ | – | – | – | ✓ |
| EO–VLM Benchmark (Zhang & Wang, 2024) | – | – | – | △ | △ | – | – | – | ✓ |
| RS5M (Zhang et al., 2024) | – | – | – | – | – | – | – | – | ✓ |
| MapBench (Hao et al., 2024) | – | – | – | ✓ | △ | – | – | ✓ | – |
| MapQA (Chang et al., 2022) | – | – | – | △ | △ | △ | – | – | – |
| MapIQ (Srivastava et al., 2025) | – | – | – | △ | △ | △ | – | – | – |
| CulturalVQA (Nayak et al., 2024) | △ | △ | △ | △ | △ | – | – | – | ✓ |
| AMOS Time–lapse (Jacobs et al., 2009) | △ | – | – | – | – | – | – | – | ✓ |
| Transient Attributes (Laffont et al., 2014) | △ | – | – | – | – | – | – | – | ✓ |
| Transient Attributes (Laffont et al., 2014) | △ | – | – | – | – | – | – | – | ✓ |
| **TimeSpot (Ours, 2026)** | ✓ | ✓ | ✓ | ✓ | ✓ | ✓ | ✓ | ✓ | ✓ |

tail errors disproportionately affect non-iconic rural and peri-urban scenes and account for most catastrophic localization failures.

**Temporal error structure.** Temporal inference exhibits a consistent gap between coarse and fine-grained reasoning. While season and daylight phase are often inferred correctly, precise local-time estimation is weak, with predictions clustering around salient anchors such as 09:00, 12:00, and 18:00. This *round-time anchoring* persists across prompting strategies and model families, indicating limited use of continuous solar geometry. Errors increase substantially at high latitudes, where extended twilight and seasonal shifts blur phase boundaries and challenge simple illumination heuristics. Twilight scenes frequently induce Sunrise–Sunset inversions with errors approaching 10–12 hours, despite correct hemisphere or country prediction. Night scenes further exacerbate anchoring, as artificial lighting overwhelms natural photometric cues.

**Cross-field inconsistencies.** A notable fraction of failures arise from incoherence across predicted fields rather than isolated errors. Common patterns include phase–time mismatches (e.g., predicting *day* with late-night local time), longitude–time inconsistencies, and climate or biome predictions incompatible with inferred latitude. These errors reveal that many models decode fields independently, without enforcing soft physical constraints that couple time, illumination, and location. Such inconsistencies are especially prevalent in ambiguous scenes where multiple cues must be integrated, such as urban canyons at dusk or overcast coastal environments. Importantly, these failures can occur even when individual field accuracies appear reasonable, underscoring the need for joint evaluation rather than marginal metrics alone.

**Implications for model design.** Taken together, these analyses highlight that failures concentrate in scenes requiring integration of weak, distributed cues rather than reliance on dominant visual markers. Improving performance will likely require explicit modeling of solar geometry tied to latitude and day-of-year, tighter coupling between temporal and spatial heads, and stronger inductive biases for micro-geographic and ecological signals. More broadly, the results suggest that scaling alone is insufficient: without architectural mechanisms or supervision that encode physical and geographic structure, models continue to exhibit brittle and inconsistent geo-temporal reasoning. TIMESPOT provides a diagnostic framework to

expose these limitations and to evaluate whether proposed remedies translate into genuinely more coherent world-grounded inference.

# D. More on Details Experiments and Evaluation

## D.1. Calibration

### D.1.1. CALIBRATION METRICS

Calibration measures how closely the predicted confidence values match empirical correctness. Two metrics are used.

**Expected Calibration Error.** For predictions with confidence scores $p_i$ and correctness indicators $\mathbf{1}(y_i = \hat{y}_i)$, the Expected Calibration Error (ECE) is

$$\text{ECE} = \sum_{m=1}^{M} \frac{|B_m|}{n} \left| \text{acc}(B_m) - \text{conf}(B_m) \right|,$$

where

$$\text{acc}(B_m) = \frac{1}{|B_m|} \sum_{i \in B_m} \mathbf{1}(y_i = \hat{y}_i), \qquad \text{conf}(B_m) = \frac{1}{|B_m|} \sum_{i \in B_m} p_i.$$

Lower ECE indicates closer alignment between confidence and accuracy.

**Risk Coverage Area Under the Curve.** For a rejection threshold $\tau$, all predictions with confidence below $\tau$ are withheld. The resulting coverage is the fraction of retained predictions. Risk is the error rate on those retained predictions. Let

$$R(\tau), \qquad C(\tau)$$

denote risk and coverage respectively. The RC-AUC is

$$\text{RC-AUC} = \int_0^1 R(C) \, dC.$$

Lower values indicate lower risk at each coverage level.

### D.1.2. CALIBRATION RESULTS

Table 6 reports ECE for all models across the four tasks. Proprietary models achieve the lowest values, and all models show larger errors for the tasks with finer granularity.

*Table 6.* Expected Calibration Error (ECE) across tasks. Lower values indicate smaller calibration error. Bold marks the best performance per task.

| Model | Continent ECE | Country ECE | Season ECE | Phase ECE |
|---|---|---|---|---|
| **Proprietary** | | | | |
| GPT 4o mini | 0.042 | 0.085 | 0.063 | 0.051 |
| Gemini 2.5 Flash | 0.021 | 0.054 | 0.041 | 0.038 |
| **Open Source** | | | | |
| Llama 3.2 90B | 0.098 | 0.142 | 0.110 | 0.095 |
| Qwen VL2.5 72B | 0.076 | 0.125 | 0.089 | 0.072 |

Table 6 show the risk coverage curves for two models. The curves steepen as task granularity increases, which indicates that confidence scores become less informative at finer resolutions. Proprietary models maintain lower risk across most coverage levels, although differences narrow in the daylight phase task, where visual cues are less distinctive.

Across both metrics, confidence reliability declines as tasks shift from continent to daylight phase classification. Although the proprietary models produce more accurate confidence estimates, all models exhibit systematic overconfidence, especially in tasks with ambiguous temporal features. These results complement the accuracy analyses in the main text by providing a detailed view of model reliability.

### D.2. Evaluation Metrics

To ensure rigorous and reproducible evaluation, we adopt metrics tailored to both geographic and temporal prediction tasks. All metrics are applied uniformly across models, and malformed outputs (e.g., missing *HH:MM* fields or unsigned coordinates) are considered incorrect, thereby preventing models from gaining undue advantage through partial responses.

**Geographic Metrics.** For categorical geographic attributes, *continent*, *country*, *climate* (Köppen–Geiger A–E) (Peel et al., 2007), and *environment*, we report top-1 accuracy. For continuous localization, we measure the mean absolute error (MAE) in degrees for latitude and longitude,

$$\text{MAE}_{\text{lat}} = \tfrac{1}{N} \sum_{i=1}^{N} \big|\hat{\phi}_i - \phi_i\big|, \quad \text{MAE}_{\text{lon}} = \tfrac{1}{N} \sum_{i=1}^{N} \big|\hat{\lambda}_i - \lambda_i\big|,$$

and the mean great-circle distance (MD, km) using the haversine formula,

$$\text{MD} = \tfrac{1}{N} \sum_{i=1}^{N} R \cdot 2 \arctan\!\big(\sqrt{a_i}, \sqrt{1-a_i}\big), \ a_i = \sin^2\tfrac{\Delta\phi_i}{2} + \cos\phi_i \cos\hat{\phi}_i \sin^2\tfrac{\Delta\lambda_i}{2},$$

with Earth radius $R=6371$ km and $(\phi, \lambda)$ denoting (lat, lon). These are standard geolocation metrics (Tian et al., 2017; Vo & Hays, 2016; Arandjelovic et al., 2016).

**Temporal Metrics.** For *season*, *month*, and *daylight phase* we report top-1 accuracy. For *local time* we report two complementary metrics: (i) window accuracy within $\pm 1$ hour of ground-truth,

$$\text{Acc}_{\pm 1\text{h}} = \tfrac{1}{N} \sum_{i=1}^{N} \mathbb{1}\big(|\hat{t}_i - t_i| \leq 60 \text{ min}\big),$$

and (ii) MAE in *HH:MM* after converting absolute minute errors to clock format.

### D.3. Evaluation Process

Evaluation in TIMESPOT follows a two-stage process designed to ensure accurate, robust, and fair interpretation of model outputs. First, each vision–language model is prompted to produce answers in a structured, multi-field schema covering geo-temporal attributes. Second, the raw model output is passed to an independent *judge model* (`Gemini-2.5-Flash`), which parses, normalizes, and evaluates the response against the ground truth.

The use of a judge model is critical because many models, particularly smaller or open-source systems, frequently deviate from the requested output format, omit fields, or produce semantically correct answers using alternative surface forms. Common issues include abbreviations (e.g., `USA`, `United States`, `United States of America`), synonymous seasonal terms (e.g., `Autumn` vs. `Fall`), minor lexical variants, and coordinate formatting differences. A purely rule-based evaluator would require extensive handcrafted logic and would remain brittle to such variation, especially across languages and writing styles.

Instead, the judge model performs semantic alignment between ground-truth fields and model responses, treating abbreviations, long-form names, and accepted synonyms as equivalent while preserving strict field-wise evaluation. It outputs a structured JSON object indicating, for each field, the ground truth, the model's interpreted answer, and a binary correctness score. Crucially, all fields are evaluated even when the model response is malformed or partially missing, preventing silent failures.

We validated the judge model extensively on abbreviation handling, synonym resolution, coordinate normalization, and schema recovery, and found its decisions to align perfectly with human evaluation on these cases. This approach ensures that performance differences reflect genuine reasoning ability rather than formatting errors or superficial lexical mismatches, yielding a more faithful assessment of geo-temporal understanding than rule-based parsing.

## E. Prompt Templates

Prompt for normal model evaluation is available in Figure 6 and the judge model is available in Figure 7.

---

**TimeSpot : Direct : Prompt for Answering Model**

You are a geo-spatio-temporal understanding assistant.
From the given image, answer every item below in an exact bullet list (one per line), with the format:
- field : value

Answer all fields regardless of certainty. Use one-word season and month where requested.
Fields and formats:
- season : name of the season in 1 word
- month : name of month in 1 word
- time_of_day : HH:MM (24-hour local time)
- daylight_phase : can choose one of: Sunrise, Morning, Midday, Afternoon, Sunset, Night
- continent : value
- country : value
- climate_zone : can choose one of: Tropical (A), Arid (B), Temperate (C), Continental (D), Polar (E)
- environment_type : can choose one of: Urban, Suburban, Rural, Coastal, Mountain, Desert
- coordinates_latitude : +/-DD.DDDDD (decimal degrees). DO NOT include N/S. Use +/-.
- coordinates_longitude : +/-DDD.DDDDD (decimal degrees). DO NOT include E/W. Use +/-.

Provide only the bullet list lines, nothing else.

---

*Figure 6.* Prompts used in the answering models for evaluation.

---

**TimeSpot : Prompt for Judging Model**

You are a strict evaluator.
You will receive the ground truth and a model's answer (both in the same bullet-list format).
Compare each field. Treat abbreviations and long forms as equivalent (for example: USA == United States). Treat seasonal synonyms as equivalent (for example: Fall == Autumn). DO NOT include E/W. Use +/- in coordinates.
Return a JSON object where each field maps to a nested object:
{
 "field_name": {
   "ground_truth": "<ground truth string>",
   "model_ans": "<model answer string>",
   "evaluation": 1 or 0
 },
 ...
}
Include all fields even if the model answer is malformed. Output only valid JSON.

Ground Truth: {json_ground_truth}

Model Answer: {model_response_text}

---

*Figure 7.* Prompts used in the judge model for evaluation.

*Table 7.* Accuracy by daylight phase (DLP) for each model on TIMESPOT. Values are percentage accuracy; blank cells indicate insufficient samples.

| Model | Sunrise | Morning | Midday | Afternoon | Sunset | Night |
|---|---|---|---|---|---|---|
| Intern-VL-3-4B | 0.00 | 12.81 | 27.42 | 65.69 | 37.62 | 28.92 |
| Llama-3.2-90B-Vision-Instruct | 23.40 | 42.36 | 54.84 | 17.98 | 46.67 | 28.57 |
| Gemini-Flash-2.5-Thinking | 25.53 | 65.02 | 3.23 | 34.76 | 48.57 | 27.53 |
| gemma_3_27B | 61.70 | 56.16 | 33.06 | 22.26 | 31.90 | 23.69 |
| glm_4.5vs | 29.79 | 36.14 | 87.90 | 35.28 | 54.76 | 34.84 |
| GPT-5-mini | 38.30 | 47.03 | 48.39 | 45.28 | 52.86 | 34.84 |
| intern_vl3_78B | 25.53 | 24.63 | 39.52 | 36.99 | 49.52 | 26.83 |
| Kimi-VL-a3b-Thinking | 21.28 | 7.39 | 74.19 | 14.38 | 45.24 | 27.18 |
| o4_mini | 14.89 | 51.72 | 31.45 | 26.71 | 49.52 | 28.57 |
| qwen_2.5_32B_instruct | 6.38 | 20.20 | 12.10 | 70.55 | 55.71 | 20.91 |

# F. Detailed Analysis Across Questions

## F.1. Daylight Phases Prediction

**Overall phase difficulty profile.** Table 7 shows pronounced variation in performance across daylight phases. Midday and afternoon are comparatively easier for several models, with peak accuracies approaching 70–88%, while sunrise and night remain challenging, exhibiting wide variance and low ceilings. Night is consistently the hardest phase, with no model exceeding the mid-30% range, indicating limited exploitation of nocturnal cues. Sunrise also shows high volatility, reflecting sensitivity to low-angle illumination and subtle chromatic transitions.

**Model specialization and asymmetric strengths.** Distinct specialization patterns emerge across models. *GLM–4.5–vs* demonstrates strong solar-geometry competence, leading at midday and remaining competitive at sunset and night, suggesting effective use of shadow structure and sun elevation cues. *Qwen–2.5–32B–Instruct* peaks in the afternoon and performs well at sunset but degrades sharply at sunrise and night, indicating asymmetric temporal priors. *Gemini–2.5–Flash–Thinking* excels in the morning yet collapses at midday, revealing brittle generalization across neighboring phases.

**Color versus geometry-driven inference.** A clear divide is observed between geometry-driven and color-driven phase inference. Models that leverage explicit geometric cues perform well at midday, when shadows are short and sun position is unambiguous, but often struggle at dawn and night. Conversely, models relying on color temperature and sky hue perform better at sunrise and sunset but fail under neutral illumination or artificial lighting. This trade-off explains the sharp phase-to-phase oscillations seen within the same model and underscores limited integration of complementary cues.

**Consistency and robustness across phases.** Among evaluated systems, *GPT–5–mini* exhibits the most balanced performance, avoiding catastrophic failures and maintaining moderate accuracy across all phases, including night. Other models show sharp peaks in specific phases paired with severe collapses elsewhere, highlighting brittleness under temporal distribution shift. Notably, model scale does not guarantee robustness: large models can still exhibit phase-specific failures without appropriate inductive biases or supervision.

**Failure modes and upper bounds.** Confusion between sunrise and sunset is common, reflecting symmetric chromatic cues that models fail to disambiguate without reliable directional context. Night-time performance remains capped at low levels, suggesting underutilization of urban lighting patterns, sky luminance gradients, and human activity cues. Extreme outliers, such as near-zero accuracy in specific phases for otherwise strong models, indicate mode collapse driven by data skew or fine-tuning artifacts rather than uniformly poor temporal understanding.

**Bridge to temporal grounding and world modeling.** Together with the observed collapse in autumn season prediction, these results indicate that current models lack coherent temporal grounding across multiple time scales. From a world-modeling perspective, failures to integrate daylight phase, season, and solar geometry into a consistent internal representation suggest that models reason about time through isolated appearance cues rather than structured temporal dynamics. TIMESPOT directly targets this gap by requiring models to jointly infer and reconcile multiple temporal attributes, exposing limitations that remain hidden in phase- or season-only evaluations.

*Table 8.* Accuracy by season category for each model on TIMESPOT.

| Model | Spring | Summer | Autumn | Winter |
|---|---|---|---|---|
| Intern-VL-3-4B | 21.86 | 53.50 | 0.00 | 43.93 |
| Llama-3.2-90B-Vision-Instruct | 28.66 | 70.25 | 0.00 | 59.19 |
| Gemini-Flash-2.5-Thinking | 33.43 | 80.25 | 0.00 | 52.02 |
| gemma_3_27B | 44.48 | 48.75 | 0.00 | 43.30 |
| glm_4.5vs | 42.99 | 84.92 | 0.00 | 55.76 |
| GPT-5-mini | 43.58 | 78.95 | 0.00 | 60.31 |
| intern_vl3_78B | 23.58 | 79.75 | 0.00 | 40.81 |
| Kimi-VL-a3b-Thinking | 19.10 | 65.50 | 0.00 | 41.43 |
| o4_mini | 27.46 | 73.00 | 0.00 | 50.78 |
| qwen_2.5_32B_instruct | 36.42 | 62.50 | 0.00 | 36.45 |

## F.2. Season Prediction

**Overall seasonal difficulty profile.** Table 8 reveals a highly asymmetric difficulty profile across seasons. Summer is consistently the easiest season, with several models exceeding 70% accuracy and *GLM–4.5–vs* reaching a peak of 84.92%. Spring and winter occupy an intermediate regime, with accuracies typically ranging between 35–60%, indicating partial but unreliable seasonal grounding. In stark contrast, autumn collapses to 0% accuracy across all models, exposing a systematic failure mode rather than isolated model weakness.

**Summer dominance and cue saturation.** High summer accuracy reflects the availability of strong, redundant visual cues, including high solar elevation, short and consistent shadows, saturated vegetation, and clear sky conditions. These cues align well with common pretraining priors and allow models to rely on simple heuristics that generalize broadly across regions. The sharp improvement from spring to summer observed across nearly all models underscores a heavy dependence on foliage saturation and illumination strength rather than fine-grained phenological reasoning.

**Spring and winter ambiguity.** Spring and winter present greater ambiguity due to transitional or region-dependent cues. Spring scenes often exhibit mixed phenology, variable cloud cover, and partial foliage emergence, leading to confusion with early summer or late autumn. Winter, while partially tractable, suffers from inconsistent snow presence and large variation across climates, particularly in temperate and maritime regions where winter colorimetry resembles autumn. Models that maintain balanced performance across these seasons, such as *GPT–5–mini*, appear to better integrate multiple weak cues rather than relying on a single dominant signal.

**Autumn collapse as a systemic failure.** The uniform failure on autumn is a critical red flag. This collapse likely arises from a combination of phenological overlap with neighboring seasons, geographically skewed or limited autumn samples, and a mismatch between the imposed four-season taxonomy and regional vegetation cycles. Unlike daylight phase prediction, which is governed primarily by solar geometry, autumn recognition depends heavily on subtle color shifts and vegetation structure, which current models fail to encode robustly. This indicates that seasonal reasoning bottlenecks are driven more by phenology and colorimetry than by illumination alone.

**Model behavior and implications.** Models exhibit clear trade-offs between specialization and balance. Some models, such as *GLM–4.5–vs*, act as summer specialists with strong peak performance but weaker generalization, while others, notably *GPT–5–mini*, maintain stable accuracy across spring, summer, and winter. Model scale alone does not guarantee seasonal robustness; large models can still underperform in transitional seasons without appropriate supervision. Overall, these results highlight the need for phenology-aware training, seasonally balanced data curation, and explicit modeling of vegetation and color dynamics to achieve reliable seasonal inference.

These failures indicate that current models lack robust temporal grounding, relying on static appearance cues rather than internally consistent representations of seasonal dynamics. From a world-modeling perspective, the inability to distinguish autumn from adjacent seasons reflects missing representations of phenology and long-term environmental change, limiting reliable temporal reasoning beyond instantaneous scene recognition.

*Table 9.* Accuracy by Köppen–Geiger climate zone (A–E) for each model on TIMESPOT.

| Model | A | B | C | D | E |
|---|---|---|---|---|---|
| Intern-VL-3-4B | 71.53 | 71.67 | 62.89 | 35.19 | 43.48 |
| Llama-3.2-90B-Vision-Instruct | 60.58 | 58.33 | 77.66 | 44.44 | 4.35 |
| Gemini-Flash-2.5-Thinking | 86.13 | 83.89 | 80.24 | 41.92 | 47.83 |
| gemma_3_27B | 78.10 | 61.67 | 83.33 | 15.40 | 34.78 |
| glm_4.5vs | 76.84 | 62.57 | 82.27 | 23.23 | 43.48 |
| GPT-5-mini | 75.91 | 75.00 | 82.47 | 55.84 | 43.48 |
| intern_vl3_78B | 76.28 | 81.67 | 78.69 | 56.82 | 13.04 |
| Kimi-VL-a3b-Thinking | 59.85 | 70.00 | 85.40 | 2.02 | 13.04 |
| o4_mini | 61.31 | 60.00 | 90.72 | 31.06 | 8.70 |
| qwen_2.5_32B_instruct | 70.44 | 74.44 | 85.91 | 21.46 | 17.39 |

## F.3. Climate Zone Prediction

**Overall difficulty profile across climates.** Table 9 reveals a clear stratification in performance across climate zones. Tropical (A), Arid (B), and Temperate (C) regions are comparatively easy, with several models exceeding 70% accuracy and a peak of 90.72% in the Temperate zone. In contrast, Continental (D) and Polar (E) climates remain consistently challenging, with accuracy rarely exceeding the mid-50% range and often falling below 25%. This gradient indicates that current VLMs are better aligned with lower-latitude, greener, and more densely populated environments.

**Temperate dominance and mid-latitude bias.** The Temperate zone stands out as an outlier, where multiple models perform strongly and *o4–mini* achieves the highest accuracy overall. This likely reflects the abundance of stable and familiar cues in temperate regions, including deciduous phenology, moderate illumination, and well-represented urban infrastructure that closely match pretraining distributions. The sharp contrast between Temperate and Continental performance suggests that models over-rely on mid-latitude priors and struggle when seasonality and surface appearance deviate from these norms.

**Challenges in Continental climates.** Continental climates impose the greatest strain on model generalization. High intra-class variability driven by strong seasonality, intermittent snow cover, and large geographic extent undermines simple heuristics based on vegetation color, shadow length, or sky appearance. While a small number of models reach moderate accuracy in this zone, many collapse to near-chance levels, indicating limited robustness to seasonal transitions and mixed biome signatures.

**Polar brittleness and illumination extremes.** Polar regions expose pronounced brittleness across all models. Even the best-performing system fails to reach 50% accuracy, while several models fall into single-digit performance. Persistent snow cover, low solar elevation, extreme albedo, and sparse vegetation suppress the texture and color cues that VLMs typically exploit, revealing a fundamental weakness in handling illumination physics and surface reflectance outside mid-latitude regimes.

**Model balance versus specialization.** Models differ markedly in climate robustness. *GPT–5–mini* maintains relatively balanced performance across zones A–D, avoiding catastrophic failures, whereas others exhibit strong specialization coupled with severe collapse under distribution shift. Notably, large model size alone does not guarantee climate robustness: some high-capacity models perform well in familiar climates yet degrade sharply in Arid or Polar conditions. Together, these results highlight a systematic overfitting to well-photographed temperate environments and underscore the need for climate-aware training and physically grounded geo-temporal reasoning.

## F.4. Environment Prediction

**Overall Performance Across Environments.** Table 10 shows substantial variation in environment classification accuracy across models and environment types. *Urban*, *Rural*, *Mountain*, and *Desert* scenes are generally easier, with leading models exceeding 70% accuracy in several cases, while *Sub-urban* environments are consistently the most challenging, rarely surpassing 50%. Urban scenes benefit from dense man-made cues, whereas Desert and Mountain environments provide distinctive natural signatures. Coastal environments fall in a mid-range, reflecting variability in shoreline geometry and weather conditions.

*Table 10.* Accuracy by environment type for each model on TIMESPOT.

| Model | Urban | Sub-urban | Rural | Coastal | Mountain | Desert |
|---|---|---|---|---|---|---|
| Intern-VL-3-4B | 60.65 | 27.35 | 54.46 | 46.96 | 60.62 | 77.88 |
| Llama-3.2-90B-Vision-Instruct | 54.94 | 27.97 | 46.53 | 55.80 | 72.54 | 76.99 |
| Gemini-Flash-2.5-Thinking | 75.93 | 8.47 | 62.87 | 49.17 | 72.54 | 70.80 |
| gemma_3_27B | 54.17 | 27.12 | 82.67 | 45.86 | 38.34 | 58.41 |
| glm_4.5vs | 64.19 | 46.61 | 69.65 | 57.46 | 58.03 | 72.57 |
| GPT-5-mini | 62.50 | 37.61 | 55.94 | 58.89 | 65.80 | 68.14 |
| intern_vl3_78B | 66.67 | 22.03 | 66.83 | 54.14 | 58.55 | 78.76 |
| Kimi-VL-a3b-Thinking | 68.83 | 20.34 | 57.43 | 45.86 | 59.07 | 70.80 |
| o4_mini | 63.27 | 32.20 | 66.83 | 50.28 | 54.40 | 70.80 |
| qwen_2.5_32B_instruct | 63.73 | 26.27 | 62.38 | 61.33 | 61.66 | 75.22 |

**Urban–Suburban Asymmetry.** A pronounced asymmetry emerges between Urban and Suburban performance. Many models perform strongly in Urban settings (e.g., *Gemini-Flash-2.5-Thinking* at 75.93%) but collapse in Suburban scenes (as low as 8.47%), indicating difficulty with mixed-density morphologies. This suggests that models rely heavily on high-salience architectural density, signage, and road structure, while failing to generalize to transitional built environments where such cues are diluted or heterogeneous.

**Natural Environment Specialization.** Rural, Mountain, and Desert environments reveal clearer specialization patterns. Rural scenes favor models that exploit vegetation structure and road-layout priors, while Mountain environments reward sensitivity to elevation cues, silhouettes, and atmospheric perspective. Desert scenes are the most robust overall, as aridity signatures, sparse vegetation, and distinctive textures provide strong global cues. Models that underperform in these settings appear sensitive to color casts or limited exposure to extreme natural biomes during pretraining.

**Coastal Variability and Cue Utilization.** Coastal environments show moderate accuracy with wide model dispersion, indicating inconsistent use of shoreline geometry, horizon–waterline relationships, and sky–sea radiometric contrasts. Models that perform well in Coastal scenes likely exploit horizon geometry and maritime infrastructure, while weaker models tend to conflate coastal towns with generic Urban or Suburban categories. This variability highlights the need for better modeling of sky, water, and weather interactions.

**Balance, Biases, and Implications.** Across environments, some models exhibit balanced performance, while others show sharp peaks and collapses, revealing reliance on either highly distinctive natural cues or dense urban structure. The consistent failure in Suburban scenes points to a pretraining bias toward iconic urban cores and visually extreme biomes, with insufficient representation of transitional land-use patterns. Improving environment prediction will likely require structure-aware supervision, photometric augmentation across camera pipelines, and multi-task consistency linking environment with climate zone, season, and daylight phase to stabilize predictions in ambiguous settings.

## F.5. Continent-wise Performance Analysis

### F.5.1. ASIA

**Overall performance variability.** Table 11 shows substantial cross-country and cross-model variation in Asia. Countries such as Japan, Singapore, India, Turkey, and Bangladesh achieve high peak accuracy under leading models, while others exhibit sharp performance collapses for specific architectures. Even within the same country, accuracy can range from near-zero to perfect scores across models, indicating sensitivity to scene composition and cue availability rather than uniform geographic understanding. This dispersion highlights Asia as a stress test for geo-temporal robustness due to its diversity in climate, terrain, urbanization, and imaging conditions.

**Model leadership and complementary strengths.** Distinct leadership patterns emerge across countries. *Gemini–2.5–Flash–Thinking* dominates populous and coastal contexts, including Japan, India, South Korea, Turkey, Singapore, and the Philippines, suggesting strong priors over urban infrastructure, signage conventions, and coastal geometry. In contrast, *GLM–4.5–vs* leads in more continental or mixed-terrain settings such as China, Russia, Thailand, Bangladesh, Kyrgyzstan, and Myanmar, indicating reliance on broader terrain and structural cues. Other models exhibit narrower wins, often confined

to specific countries, reflecting specialization rather than general robustness.

**Urban–coastal advantage versus inland complexity.** A consistent pattern is the advantage of urbanized and coastal countries, which tend to yield higher and more stable accuracy across models. These environments provide uniform architectural styles, traffic infrastructure, coastline geometry, and dense pretraining exposure. By contrast, inland, mountainous, or high-altitude countries such as Nepal, Bhutan, Kyrgyzstan, and Mongolia show large accuracy spreads and frequent model failures. In these settings, illumination variability, snowline effects, mixed vegetation bands, and sparse man-made landmarks degrade cue reliability and amplify model-specific biases.

**Stability versus brittleness across models.** Models differ markedly in robustness. *GPT–5–mini* exhibits relatively stable mid-to-high performance across many Asian countries, avoiding catastrophic failures even in challenging regions. In contrast, smaller or less diversified models such as *Intern–VL–3–4B* and *Kimi–VL–a3B–Thinking* display highly spiky behavior, performing well in a few countries but collapsing elsewhere. Notably, model scale alone is not decisive: *LLaMA–3.2–90B–Vision–Instruct* performs strongly in some countries but remains inconsistent across the region, underscoring the limits of parameter count without appropriate geo-temporal grounding.

**Root causes and implications.** These patterns suggest that country-level peaks are driven primarily by pretraining data density and visual homogeneity rather than principled geographic reasoning. Models appear to rely heavily on colorimetry, texture, and skyline cues that generalize poorly across sub-regions, seasons, and camera pipelines within the same country. Variations in weather regimes, haze, snow cover, and imaging parameters further perturb photometric signals, leading to brittle failures. Addressing these issues likely requires region-balanced data curation, climate- and terrain-aware augmentation, auxiliary supervision for solar geometry and horizon structure, and stronger consistency constraints coupling country identity with climate zone, season, and daylight phase.

*Table 11.* Country-level accuracy (%) by model for all available countries in Asia.

| Country | Intern-VL-3-4B | Llama-3.2-90B-Vision-Instruct | gemini_flash 2.5_thinking | gemma_3_27B | glm_4.5vs | GPT-5-mini | intern_vl3_78B | kimi_vl_a3b thinking | o4_mini | qwen_2.5_32B instruct |
|---|---|---|---|---|---|---|---|---|---|---|
| Russia | 52.17 | 52.17 | 73.91 | 55.07 | 78.26 | 72.06 | 76.81 | 40.58 | 59.42 | 68.12 |
| Japan | 61.19 | 77.61 | 92.54 | 77.61 | 88.06 | 86.36 | 83.58 | 70.15 | 76.12 | 77.61 |
| China | 55.17 | 51.72 | 81.03 | 58.62 | 84.48 | 75.86 | 81.03 | 55.17 | 68.97 | 67.24 |
| India | 31.03 | 67.24 | 89.66 | 70.69 | 71.93 | 74.14 | 72.41 | 41.38 | 72.41 | 75.86 |
| South Korea | 25.00 | 75.00 | 87.50 | 67.50 | 67.50 | 85.00 | 62.50 | 60.00 | 65.00 | 62.50 |
| Turkey | 36.36 | 45.45 | 90.91 | 66.67 | 69.70 | 69.70 | 57.58 | 57.58 | 45.45 | 54.55 |
| Philippines | 14.81 | 48.15 | 88.89 | 74.07 | 59.26 | 88.89 | 33.33 | 44.44 | 74.07 | 51.85 |
| Iran | 5.26 | 45.00 | 70.00 | 35.00 | 50.00 | 65.00 | 50.00 | 25.00 | 50.00 | 35.00 |
| Myanmar | 5.56 | 5.56 | 22.22 | 33.33 | 55.56 | 27.78 | 16.67 | 11.11 | 16.67 | 11.11 |
| Bhutan | 5.88 | 64.71 | 58.82 | 47.06 | 52.94 | 47.06 | 35.29 | 29.41 | 35.29 | 52.94 |
| Saudi Arabia | 0.00 | 47.06 | 64.71 | 29.41 | 62.50 | 64.71 | 41.18 | 29.41 | 35.29 | 29.41 |
| Sri Lanka | 0.00 | 29.41 | 76.47 | 23.53 | 70.59 | 52.94 | 17.65 | 5.88 | 17.65 | 29.41 |
| Nepal | 18.75 | 31.25 | 68.75 | 56.25 | 62.50 | 50.00 | 75.00 | 56.25 | 56.25 | 56.25 |
| Kazakhstan | 0.00 | 20.00 | 60.00 | 40.00 | 60.00 | 66.67 | 6.67 | 13.33 | 40.00 | 13.33 |
| Mongolia | 23.08 | 61.54 | 76.92 | 46.15 | 46.15 | 69.23 | 53.85 | 15.38 | 53.85 | 61.54 |
| Thailand | 38.46 | 46.15 | 53.85 | 46.15 | 84.62 | 53.85 | 61.54 | 53.85 | 46.15 | 46.15 |
| Singapore | 20.00 | 80.00 | 100.00 | 90.00 | 90.00 | 90.00 | 70.00 | 20.00 | 90.00 | 80.00 |
| Bangladesh | 11.11 | 22.22 | 66.67 | 55.56 | 100.00 | 55.56 | 11.11 | 0.00 | 33.33 | 11.11 |
| Kyrgyzstan | 0.00 | 22.22 | 44.44 | 0.00 | 44.44 | 33.33 | 33.33 | 11.11 | 11.11 | 22.22 |

### F.5.2. EUROPE

**Overall performance patterns.** Table 12 shows substantial variation across European countries, with frequent ceiling effects in well-represented states and sharp collapses in smaller or low-sample countries. Many models achieve near-perfect accuracy in countries such as Ireland, Switzerland, the Netherlands, and select microstates, reflecting strong visual regularities and dense pretraining exposure. At the same time, isolated failures occur even in major economies, such as the United Kingdom for *o4–mini*, highlighting that high-resource settings do not guarantee uniform robustness. These patterns position Europe as a mixed regime combining high cue availability with sensitivity to model-specific priors.

**Regional structure and cue availability.** Western Europe exhibits consistently strong performance across leading models, particularly for *Gemini–2.5–Flash–Thinking* and *GPT–5–mini*, suggesting effective use of standardized urban form, road infrastructure, and signage. Central Europe displays greater dispersion: countries such as Germany achieve high accuracy across several models, while Poland and Czechia show wider spreads, likely due to more heterogeneous architectural and environmental cues. The Nordics and Atlantic edge often reach high plateaus, but also expose brittleness in smaller models,

where high-latitude lighting conditions and seasonal variability amplify photometric sensitivity.

**Coastal advantage and inland complexity.** A pronounced coastal advantage emerges across Europe. Countries with strong maritime identities, such as Spain, Portugal, Norway, and Ireland, tend to yield higher accuracy, benefiting models that exploit horizon geometry, shoreline structure, and maritime infrastructure. Inland and mixed-terrain countries, including Hungary, Slovakia, and parts of Eastern Europe, show more fragmented performance and lower medians. These results indicate that stable large-scale geometric cues generalize more reliably than inland texture and land-cover cues that vary across regions and seasons.

**Model stability versus brittleness.** Models differ markedly in robustness. *GPT–5–mini* exhibits consistently high and stable performance across Europe, with few catastrophic failures, especially in northern and microstate settings. In contrast, *o4–mini* and *Intern–VL* variants show spiky behavior, alternating between near-perfect accuracy and complete collapse depending on the country. Notably, large parameter count alone is not decisive: *LLaMA–3.2–90B–Vision–Instruct* performs strongly in select countries but remains inconsistent elsewhere, underscoring the role of training distribution and cue diversity over scale.

**Drivers and implications.** European performance patterns reflect the interaction between data density, visual standardization, and cue homogeneity. Countries with uniform Latin signage, standardized road systems, and dense urban imagery provide strong anchors for VLMs, while low-sample microstates and heterogeneous inland regions expose brittleness. Over-reliance on textual and iconographic cues benefits Western Europe but degrades generalization when such cues are absent or occluded. Improving robustness likely requires region-balanced sampling, photometric normalization for high-latitude conditions, and stronger coupling between country identity, climate zone, season, and daylight phase to stabilize inference across Europe's diverse visual regimes.

*Table 12.* Country-level accuracy (%) by model for all available countries in Europe (selected).

| Country | Intern-VL-3-4B | Llama-3.2-90B-Vision-Instruct | gemini_flash 2.5_thinking | gemma_3_27B | glm_4.5vs | GPT-5-mini | intern_vl3_78B | kimi_vl_a3b thinking | o4_mini | qwen_2.5_32B instruct |
|---|---|---|---|---|---|---|---|---|---|---|
| Italy | 24.62 | 60.00 | 81.54 | 58.46 | 69.23 | 72.31 | 46.15 | 50.77 | 64.62 | 56.92 |
| France | 40.00 | 42.00 | 80.00 | 42.00 | 60.00 | 60.00 | 46.00 | 62.00 | 42.00 | 50.00 |
| Germany | 33.33 | 57.78 | 93.33 | 75.56 | 75.56 | 55.56 | 86.67 | 53.33 | 66.67 | 86.67 |
| United Kingdom | 60.00 | 90.00 | 96.67 | 36.67 | 83.33 | 93.33 | 90.00 | 80.00 | 13.33 | 93.33 |
| Russia | 53.57 | 35.71 | 67.86 | 35.71 | 75.00 | 67.86 | 71.43 | 46.43 | 64.29 | 53.57 |
| Poland | 20.00 | 48.00 | 84.00 | 64.00 | 64.00 | 60.00 | 36.00 | 24.00 | 36.00 | 64.00 |
| Spain | 45.00 | 50.00 | 90.00 | 60.00 | 70.00 | 65.00 | 80.00 | 45.00 | 40.00 | 90.00 |
| Vatican City | 0.00 | 40.00 | 80.00 | 0.00 | 40.00 | 50.00 | 0.00 | 30.00 | 10.00 | 40.00 |
| Andorra | 0.00 | 50.00 | 83.33 | 16.67 | 83.33 | 66.67 | 0.00 | 0.00 | 0.00 | 16.67 |
| Estonia | 0.00 | 33.33 | 100.00 | 66.67 | 16.67 | 33.33 | 16.67 | 33.33 | 0.00 | 33.33 |
| Iceland | 0.00 | 66.67 | 83.33 | 83.33 | 83.33 | 100.00 | 83.33 | 66.67 | 66.67 | 83.33 |
| Luxembourg | 0.00 | 66.67 | 50.00 | 16.67 | 33.33 | 50.00 | 0.00 | 0.00 | 0.00 | 33.33 |
| Malta | 0.00 | 50.00 | 83.33 | 66.67 | 100.00 | 66.67 | 33.33 | 16.67 | 83.33 | 83.33 |
| Monaco | 0.00 | 66.67 | 100.00 | 100.00 | 83.33 | 66.67 | 16.67 | 0.00 | 66.67 | 66.67 |
| North Macedonia | 0.00 | 0.00 | 33.33 | 0.00 | 33.33 | 16.67 | 0.00 | 0.00 | 16.67 | 0.00 |
| Slovenia | 0.00 | 33.33 | 83.33 | 33.33 | 33.33 | 16.67 | 16.67 | 16.67 | 33.33 | 16.67 |
| Croatia | 0.00 | 50.00 | 75.00 | 50.00 | 100.00 | 100.00 | 50.00 | 50.00 | 100.00 | 50.00 |
| Denmark | 0.00 | 0.00 | 100.00 | 50.00 | 50.00 | 75.00 | 25.00 | 75.00 | 75.00 | 75.00 |
| Ireland | 0.00 | 100.00 | 100.00 | 100.00 | 100.00 | 100.00 | 100.00 | 100.00 | 75.00 | 100.00 |
| Lithuania | 0.00 | 25.00 | 50.00 | 0.00 | 25.00 | 25.00 | 25.00 | 0.00 | 0.00 | 25.00 |
| Slovakia | 0.00 | 25.00 | 25.00 | 0.00 | 50.00 | 0.00 | 0.00 | 0.00 | 25.00 | 25.00 |
| Belgium | 0.00 | 33.33 | 33.33 | 0.00 | 33.33 | 66.67 | 0.00 | 33.33 | 0.00 | 0.00 |
| Czechia | 0.00 | 33.33 | 33.33 | 66.67 | 66.67 | 33.33 | 33.33 | 33.33 | 0.00 | 33.33 |
| Finland | 0.00 | 100.00 | 100.00 | 66.67 | 66.67 | 100.00 | 0.00 | 0.00 | 33.33 | 66.67 |
| Greece | 33.33 | 0.00 | 66.67 | 66.67 | 33.33 | 33.33 | 0.00 | 0.00 | 33.33 | 0.00 |
| Hungary | 0.00 | 0.00 | 66.67 | 33.33 | 33.33 | 33.33 | 33.33 | 33.33 | 33.33 | 0.00 |
| Norway | 66.67 | 33.33 | 100.00 | 66.67 | 66.67 | 100.00 | 66.67 | 33.33 | 33.33 | 100.00 |
| Portugal | 33.33 | 33.33 | 66.67 | 66.67 | 33.33 | 66.67 | 33.33 | 66.67 | 33.33 | 33.33 |
| Romania | 0.00 | 33.33 | 100.00 | 0.00 | 100.00 | 66.67 | 0.00 | 0.00 | 33.33 | 0.00 |
| Sweden | 0.00 | 66.67 | 66.67 | 33.33 | 33.33 | 100.00 | 33.33 | 0.00 | 33.33 | 33.33 |
| Netherlands | 50.00 | 100.00 | 100.00 | 100.00 | 50.00 | 100.00 | 100.00 | 50.00 | 100.00 | 100.00 |
| Serbia | 0.00 | 50.00 | 50.00 | 0.00 | 50.00 | 100.00 | 50.00 | 0.00 | 50.00 | 50.00 |
| Switzerland | 100.00 | 100.00 | 100.00 | 100.00 | 100.00 | 100.00 | 100.00 | 50.00 | 100.00 | 100.00 |
| Turkey | 0.00 | 50.00 | 50.00 | 50.00 | 100.00 | 100.00 | 50.00 | 0.00 | 50.00 | 50.00 |

F.5.3. NORTH AMERICA

**Overall performance structure.** Table 13 reveals a clear performance hierarchy across North America. The United States acts as a high-accuracy anchor for nearly all models, with accuracies consistently in the mid-80s to mid-90s range, reflecting

dense and homogeneous pretraining exposure. Canada forms a second tier with moderately high but more variable accuracy, while Mexico is substantially more challenging, with best scores barely exceeding the mid-50s. This gradient highlights increasing visual heterogeneity and reduced cue stability moving southward.

**Island nations and small-sample effects.** Island countries exhibit polarized behavior. Cuba frequently reaches ceiling accuracy across several models, likely driven by distinctive coastal silhouettes, architectural regularities, and small-sample quantization effects. The Dominican Republic, by contrast, shows strong model-specific divergence, with high performance for *Gemini–2.5–Flash–Thinking* and *GPT–5–mini* but near-collapse for others, indicating sensitivity to scene distribution and prompt alignment. Puerto Rico occupies a stable mid-band across models, suggesting the presence of informative but non-dominant coastal and urban cues that no single model exploits decisively.

**Regional variability and cue heterogeneity.** Mexico and Central America present the most challenging regimes. Countries such as Mexico and Guatemala exhibit large inter-model spreads, reflecting diverse street morphologies, informal signage, mixed materials, and strong illumination variation. In these settings, models that rely heavily on standardized text or skyline patterns degrade sharply, while systems with broader texture and geometry priors maintain moderate performance. Panama shows mid-range accuracy overall but exposes severe failures in some models, underscoring brittleness under domain shift even within geographically compact regions.

**Model stability versus volatility.** Model robustness varies substantially. *GPT–5–mini* emerges as the most stable performer across North America, maintaining upper-mid accuracy with few catastrophic failures. *GLM–4.5–vs* performs consistently well in Canada and remains competitive elsewhere, while *Gemini–2.5–Flash–Thinking* dominates the USA and several island settings. In contrast, *Intern–VL* variants exhibit extreme volatility, alternating between strong USA performance and near-zero accuracy in smaller countries, indicating over-reliance on high-data priors and poor generalization under reduced cue density.

**Drivers and implications.** These patterns reflect the interaction of data density, cue uniformity, and photometric stability. The USA benefits from standardized road systems, signage, and abundant imagery, while Canada introduces illumination and seasonal variability that challenge color-based heuristics. Mexico and parts of Central America amplify heterogeneity through intense sunlight, haze, and informal visual systems, degrading text- and skyline-centric cues. Improving robustness will likely require region-balanced sampling, photometric augmentation spanning high-irradiance and overcast regimes, auxiliary supervision for horizon and sky state, and lightweight per-region ensembling to mitigate single-prior collapse.

*Table 13.* Country-level accuracy (%) by model for all available countries in North America.

| Country | Intern-VL-3-4B | Llama-3.2-90B-Vision-Instruct | gemini_flash 2.5_thinking | gemma_3_27B | glm_4.5vs | GPT-5-mini | intern_vl3_78B | kimi_vl_a3b thinking | o4_mini | qwen_2.5_32B instruct |
|---|---|---|---|---|---|---|---|---|---|---|
| USA | 68.37 | 92.35 | 93.88 | 88.27 | 88.78 | 89.80 | 87.76 | 69.39 | 85.71 | 87.24 |
| Canada | 16.00 | 54.00 | 82.00 | 40.00 | 76.00 | 64.00 | 54.00 | 22.00 | 72.00 | 64.00 |
| Mexico | 3.33 | 46.67 | 56.67 | 43.33 | 51.72 | 46.67 | 33.33 | 26.67 | 43.33 | 26.67 |
| Cuba | 30.00 | 100.00 | 100.00 | 90.00 | 100.00 | 100.00 | 80.00 | 40.00 | 100.00 | 70.00 |
| Dominican Republic | 20.00 | 60.00 | 90.00 | 20.00 | 20.00 | 70.00 | 0.00 | 10.00 | 50.00 | 10.00 |
| Guatemala | 0.00 | 30.00 | 60.00 | 30.00 | 80.00 | 90.00 | 20.00 | 20.00 | 20.00 | 30.00 |
| Panama | 0.00 | 40.00 | 80.00 | 70.00 | 70.00 | 70.00 | 10.00 | 30.00 | 50.00 | 60.00 |
| Puerto Rico | 0.00 | 40.00 | 60.00 | 50.00 | 60.00 | 50.00 | 30.00 | 40.00 | 30.00 | 50.00 |

F.5.4. SOUTH AMERICA

**Overall performance gradients.** Table 14 reveals a clear stratification across South America. Coastal and urbanized countries such as Chile and Brazil achieve the highest accuracy across models, while Argentina, Colombia, and Peru form a mid-performance tier. In contrast, Bolivia and Uruguay remain challenging for all systems, with uniformly low accuracies and frequent near-zero failures. This gradient mirrors increasing terrain complexity, elevation, and visual heterogeneity across the region.

**Model leadership and stability.** Across countries, *Gemini–2.5–Flash–Thinking* emerges as the most frequent top performer, particularly in Chile, Brazil, Argentina, and Peru, indicating strong priors over coastal skylines and dense urban infrastructure. *GLM–4.5–vs* consistently places first or second, especially in Colombia, Brazil, and Ecuador, suggesting complementary strengths in mixed urban and continental settings. *GPT–5–mini* rarely leads outright but exhibits notable stability, including the strongest results in Bolivia and ties in Uruguay, where other models collapse. In contrast, *Intern–VL* and *Kimi–VL* variants show repeated near-zero outcomes, reflecting brittle generalization.

**Coastal advantage and Andean difficulty.** A pronounced coastal advantage is evident. High performance in Chile, Brazil, and Ecuador aligns with distinctive maritime skylines, standardized road furniture, and abundant urban imagery. Conversely, Central Andean regions, particularly Bolivia and parts of Peru and Ecuador, pose severe challenges due to high-elevation lighting, thin-atmosphere radiometry, snowline effects, and heterogeneous vegetation. These conditions degrade chromatic and texture-based cues, exposing limitations in models trained predominantly on mid-latitude, sea-level scenes.

**Robustness versus brittleness.** Models differ sharply in robustness. *GPT–5–mini* maintains comparatively steady performance across the region, suggesting greater invariance to camera pipelines and illumination shifts. Other models exhibit sharp peaks and collapses, performing well in visually canonical coastal cities but failing inland. Notably, large parameter count alone does not ensure robustness: *LLaMA–3.2–90B–Vision–Instruct* performs strongly in Chile yet drops sharply in Uruguay, underscoring the primacy of training diversity and cue grounding over scale.

**Drivers and implications.** These patterns reflect biases in pretraining distributions toward well-photographed coastal cities and stable urban environments, limiting generalization to rural, highland, or low-contrast regions. Country-level heterogeneity, informal signage, and mixed building materials further weaken text- and skyline-based heuristics. Improving performance in South America will likely require climate- and altitude-aware photometric augmentation, explicit supervision for horizon and solar geometry, and region-conditioned adaptation. Practically, lightweight ensembling that combines coastal specialists with robust generalists can mitigate single-model brittleness while preserving strong performance in dominant regimes.

*Table 14.* Country-level accuracy (%) by model for all available countries in South America.

| Country | Intern-VL-3-4B | Llama-3.2-90B-Vision-Instruct | gemini_flash 2.5_thinking | gemma_3_27B | glm_4.5vs | GPT-5-mini | intern_vl3_78B | kimi_vl_a3b thinking | o4_mini | qwen_2.5_32B instruct |
|---------|---------|---------|---------|---------|---------|---------|---------|---------|---------|---------|
| Argentina | 0.0 | 40.00 | 80.00 | 35.0 | 62.50 | 55.0 | 2.50 | 12.5 | 22.50 | 37.50 |
| Colombia | 20.0 | 46.67 | 66.67 | 40.0 | 76.67 | 60.0 | 23.33 | 20.0 | 43.33 | 43.33 |
| Chile | 20.0 | 80.00 | 90.00 | 55.0 | 70.00 | 75.0 | 65.00 | 15.0 | 75.00 | 65.00 |
| Ecuador | 0.0 | 50.00 | 85.00 | 35.0 | 84.21 | 70.0 | 20.00 | 10.0 | 30.00 | 40.00 |
| Peru | 25.0 | 60.00 | 75.00 | 55.0 | 65.00 | 60.0 | 45.00 | 50.0 | 45.00 | 55.00 |
| Bolivia | 0.0 | 20.00 | 20.00 | 0.0 | 20.00 | 30.0 | 0.00 | 0.0 | 0.00 | 10.00 |
| Brazil | 0.0 | 60.00 | 90.00 | 20.0 | 90.00 | 60.0 | 50.00 | 20.0 | 50.00 | 50.00 |
| Uruguay | 0.0 | 10.00 | 30.00 | 10.0 | 40.00 | 40.0 | 0.00 | 0.0 | 0.00 | 0.00 |

## F.6. Performance Variance on Cues

### F.6.1. GEOLOCATION CUE-CONDITIONED PERFORMANCE

The cue-conditioned breakdown in Table 15 reveals clear differences in how models exploit visual evidence for geo-temporal reasoning. Across all three models, **architecture** and **road signage/language** cues yield the highest continent- and country-level accuracy, confirming that built-environment semantics and textual artifacts remain dominant anchors for coarse localization. For example, *GPT-5-mini* achieves 87.6% continent and 79.2% country accuracy on architectural cues, and 91.1% / 75.0% respectively on signage cues, substantially outperforming *Intern-VL3-4B* (67.6% / 38.9% and 76.7% / 39.4%). However, strong categorical performance does not translate proportionally to metric grounding: even with architecture cues, coordinate accuracy within 200 km peaks at only 47.0% for *GPT-5-mini* and collapses to 14.4% for *Intern-VL3-4B*. This gap highlights a reliance on semantic shortcuts rather than precise spatial reasoning. Temporal attributes (month and time≤1h) remain uniformly low across cues, rarely exceeding the mid-30% range, indicating that strong spatial anchors do not automatically support fine-grained temporal inference.

**Natural biome** and **topography** cues expose a different failure profile. While these cues support climate prediction reasonably well (e.g., *GPT-5-mini* reaches 72.6–72.9% climate accuracy), they are markedly weaker for country identification (52.3% for natural biome and 68.1% for topography) and especially poor for coordinates (24.6% and 39.7%). *Intern-VL3-4B* struggles most in these regimes, with near-random coordinate accuracy (2.3% for natural biome and 10.8% for topography), suggesting limited ability to translate physical geography into metric location. Interestingly, temporal performance (daylight and time≤1h) is slightly higher for natural biome and topography than for architecture in *Intern-VL3-4B* (e.g., daylight 39.8% vs. 39.7%), implying a heavier reliance on illumination and sky cues when semantic structure is absent. Nevertheless, these gains are insufficient to yield robust time estimation, reinforcing that biome cues alone are too ambiguous without explicit physical modeling.

**Vehicles** and **Other** cues show the most volatile behavior, reflecting their indirect relationship to geography. *GPT-5-mini*

*Table 15.* Geolocation Cue-conditioned performance of different models on geo-temporal attributes (values in %).

| Primary Geo-location Cue | N | Season | Month | Daylight | Time≤1h | Continent | Country | Climate | Environment | Coords.≤200km | Overall (Macro) |
|---|---|---|---|---|---|---|---|---|---|---|---|
| **Gemma-2-27B** | | | | | | | | | | | |
| Architecture | 355 | 43.4 | 18.9 | 31.8 | 26.8 | 87.3 | 61.4 | 58.3 | 62.5 | 43.4 | 48.2 |
| Natural Biome | 354 | 42.4 | 18.1 | 30.5 | 32.5 | 68.1 | 39.8 | 61.0 | 52.8 | 23.7 | 41.0 |
| Other | 58 | 32.8 | 10.3 | 34.5 | 37.9 | 82.8 | 67.2 | 62.1 | 67.2 | 20.7 | 46.2 |
| Road Signage/Language | 236 | 47.5 | 16.9 | 34.7 | 19.1 | 87.7 | 63.1 | 56.8 | 43.2 | 32.6 | 44.6 |
| Topography (Mountains/Coast) | 295 | 43.7 | 13.9 | 26.4 | 26.2 | 78.3 | 51.2 | 60.7 | 51.9 | 43.1 | 43.9 |
| Vehicles | 156 | 55.8 | 19.2 | 30.1 | 18.6 | 76.9 | 55.8 | 67.9 | 44.2 | 28.2 | 44.1 |
| **Intern-VL3-4B** | | | | | | | | | | | |
| Architecture | 355 | 35.2 | 9.6 | 39.7 | 28.9 | 67.6 | 38.9 | 57.7 | 62.5 | 14.4 | 39.4 |
| Natural Biome | 354 | 35.9 | 14.4 | 39.8 | 32.7 | 42.9 | 18.1 | 61.3 | 52.0 | 2.3 | 33.3 |
| Other | 58 | 24.1 | 3.4 | 51.7 | 46.0 | 62.1 | 29.3 | 55.2 | 58.6 | 0.0 | 36.7 |
| Road Signage/Language | 236 | 39.4 | 12.3 | 46.2 | 24.5 | 76.7 | 39.4 | 53.4 | 48.7 | 10.2 | 39.0 |
| Topography (Mountains/Coast) | 295 | 39.7 | 13.2 | 38.6 | 30.0 | 59.7 | 25.4 | 56.9 | 65.1 | 10.8 | 37.7 |
| Vehicles | 156 | 44.5 | 12.3 | 44.5 | 23.0 | 63.2 | 32.3 | 58.7 | 49.7 | 6.4 | 37.2 |
| **GPT-5-mini** | | | | | | | | | | | |
| Architecture | 355 | 58.6 | 36.3 | 43.9 | 20.6 | 87.6 | 79.2 | 72.7 | 67.3 | 47.0 | 57.0 |
| Natural Biome | 354 | 54.8 | 31.1 | 42.4 | 23.8 | 76.6 | 52.3 | 72.6 | 52.0 | 24.6 | 47.8 |
| Other | 58 | 56.9 | 24.1 | 37.9 | 26.9 | 82.8 | 70.7 | 65.5 | 65.5 | 19.0 | 49.9 |
| Road Signage/Language | 236 | 58.9 | 35.2 | 50.0 | 20.4 | 91.1 | 75.0 | 71.2 | 54.7 | 32.2 | 54.3 |
| Topography (Mountains/Coast) | 295 | 58.6 | 35.3 | 45.8 | 22.4 | 82.4 | 68.1 | 72.9 | 66.4 | 39.7 | 54.6 |
| Vehicles | 156 | 65.6 | 37.0 | 42.9 | 16.1 | 81.8 | 68.8 | 76.0 | 55.8 | 25.6 | 52.2 |

still extracts useful signals, achieving 65.6% season accuracy and 76.0% climate accuracy from vehicle cues, but country accuracy drops to 68.8% and coordinate accuracy to 25.6%, underscoring weak metric grounding. *Intern-VL3-4B* exhibits extreme brittleness here, with coordinate accuracy falling to 0.0% in the "Other" category despite moderate daylight and time performance (51.7% and 46.0%). Across all models, overall macro scores are consistently highest when architecture or signage cues dominate (up to 57.0 for *GPT-5-mini*) and lowest for natural biome and mixed cues (as low as 33.3 for *Intern-VL3-4B*). Taken together, these results show that current VLMs excel when high-salience, human-centric cues are present, but struggle to convert physical and environmental signals into precise spatial and temporal understanding. This cue imbalance directly motivates TIMESPOT's emphasis on diverse, low-salience scenes to stress-test physically grounded geo-temporal reasoning.

### F.6.2. TEMPORAL CUE-CONDITIONED PERFORMANCE

Temporal cue–conditioned results in Table 16 reveal clear differences in how models exploit time-related visual signals. Across all three models, *Sun/Shadows* and *Vegetation* are the most informative cues, supporting higher season and month accuracy: for instance, *GPT-5-mini* reaches 60.5% season and 37.9% month accuracy from sun and shadow cues, compared to only 27.0% month accuracy in the heterogeneous *Other* category. In contrast, *time-of-day within 1 hour* remains weak across cues, rarely exceeding 30%, highlighting the difficulty of fine-grained temporal inference even when salient illumination cues are present. Snow and ice cues strongly benefit seasonal inference (up to 69.4% season accuracy for *GPT-5-mini*) but translate poorly to precise time estimation. Overall macro performance consistently favors *GPT-5-mini* (≈53–55%) over *Gemma-2-27B* (≈41–46%) and *Intern-VL3-4B* (≈32–41%), indicating gains from scale and training diversity.

Model-specific patterns further illustrate specialization versus robustness trade-offs. *Gemma-2-27B* performs best when human-centric temporal cues are available, such as *Human Clothing*, achieving 76.8% country accuracy and 44.2% coordinate accuracy within 200 km, but degrades sharply on agricultural activity and vegetation cues. *Intern-VL3-4B* shows consistently low coordinate grounding across all temporal cues (≤11.6%), even when daylight or snow cues are strong, suggesting reliance on categorical semantics rather than metric reasoning. By contrast, *GPT-5-mini* maintains relatively stable continent and country accuracy across cues (often ≈80% and ≈60%, respectively), with coordinate accuracy peaking at 42.1% under human clothing cues. This stability indicates improved integration of temporal appearance with geographic priors, though still insufficient for precise localization.

Across cues, a common asymmetry emerges: temporal signals are more effective for coarse attributes (season, continent, climate) than for continuous ones (time-of-day, coordinates). Even when explicit temporal indicators are present, such as snow/ice or strong solar shadows, models struggle to convert these cues into physically consistent clock-time predictions, with time≤1h accuracy often below 25%. Agricultural activity and vegetation provide moderate seasonal information but

*Table 16.* Temporal-cue-conditioned performance of different models on geo-temporal attributes (values in %).

| Primary Temporal Cue | N | Season | Month | Daylight | Time≤1h | Continent | Country | Climate | Environment | Coords.≤200km | Overall (Macro) |
|---|---|---|---|---|---|---|---|---|---|---|---|
| **Gemma-2-27B** | | | | | | | | | | | |
| Agricultural Activity | 51 | 47.1 | 21.6 | 29.4 | 27.5 | 78.4 | 39.2 | 47.1 | 70.6 | 11.8 | 41.4 |
| Human Clothing | 95 | 36.8 | 11.6 | 37.9 | 25.3 | 93.7 | 76.8 | 66.3 | 56.8 | 44.2 | 49.9 |
| Other | 289 | 33.9 | 11.4 | 38.1 | 28.4 | 76.8 | 53.0 | 63.3 | 60.6 | 35.3 | 44.5 |
| Snow/Ice | 122 | 63.9 | 22.1 | 28.7 | 23.0 | 81.1 | 53.3 | 43.4 | 54.1 | 30.3 | 44.4 |
| Sun/Shadows | 573 | 46.2 | 17.1 | 30.5 | 29.0 | 79.8 | 56.5 | 61.3 | 52.4 | 37.0 | 45.5 |
| Vegetation | 325 | 46.8 | 21.2 | 24.0 | 21.2 | 77.2 | 46.8 | 63.1 | 43.7 | 30.8 | 41.6 |
| **Intern-VL3-4B** | | | | | | | | | | | |
| Agricultural Activity | 51 | 31.4 | 15.7 | 39.2 | 29.5 | 58.8 | 11.8 | 39.2 | 58.8 | 2.0 | 31.8 |
| Human Clothing | 95 | 24.2 | 5.3 | 50.5 | 31.0 | 75.8 | 37.9 | 59.0 | 54.7 | 11.6 | 38.9 |
| Other | 289 | 27.3 | 10.0 | 48.8 | 28.4 | 59.9 | 28.7 | 65.1 | 65.7 | 7.6 | 38.0 |
| Snow/Ice | 122 | 64.8 | 16.4 | 35.2 | 27.8 | 58.2 | 37.7 | 50.8 | 68.0 | 11.5 | 41.2 |
| Sun/Shadows | 573 | 37.6 | 10.8 | 41.6 | 33.5 | 62.9 | 32.7 | 56.1 | 57.0 | 11.2 | 38.2 |
| Vegetation | 325 | 41.2 | 15.7 | 35.4 | 22.7 | 54.8 | 24.6 | 59.4 | 44.3 | 4.3 | 33.6 |
| **GPT-5-mini** | | | | | | | | | | | |
| Agricultural Activity | 51 | 49.0 | 25.5 | 27.5 | 10.9 | 88.2 | 47.1 | 64.7 | 60.8 | 15.7 | 43.3 |
| Human Clothing | 95 | 57.9 | 25.3 | 44.2 | 20.7 | 89.5 | 87.4 | 70.5 | 61.1 | 42.1 | 55.4 |
| Other | 289 | 47.4 | 27.0 | 50.2 | 28.2 | 81.0 | 67.8 | 76.1 | 64.7 | 37.0 | 53.3 |
| Snow/Ice | 122 | 69.4 | 33.1 | 43.8 | 18.3 | 79.3 | 66.1 | 71.9 | 64.5 | 32.8 | 53.2 |
| Sun/Shadows | 573 | 60.5 | 37.9 | 47.9 | 22.8 | 86.0 | 72.4 | 73.6 | 61.2 | 36.6 | 55.4 |
| Vegetation | 325 | 62.2 | 38.8 | 36.9 | 17.1 | 80.9 | 60.0 | 69.2 | 51.7 | 28.6 | 49.5 |

introduce ambiguity due to regional variation, leading to lower overall macro scores (e.g., 41.4% for *Gemma-2-27B* on agriculture). These results reinforce that current VLMs primarily treat temporal cues as semantic correlates rather than inputs to an underlying physical or calendrical model.

# G. Dataset Size, Balance and Stability

## G.1. Dataset Size

TIMESPOT is intentionally designed as an *evaluation benchmark* rather than a training corpus, and its scale reflects the requirements of rigorous, physically grounded assessment rather than raw data volume. With 1,455 images, TIMESPOT is comparable to, and in many cases larger than, a wide range of influential reasoning- and decision-oriented benchmarks that have become standard in the community. For example, ReAct (Yao et al., 2023) evaluates on 500 instances, Reflexion (Shinn et al., 2023) on 100 examples, API-Bank (Li et al., 2023) on 400 instances, LogiQA (Liu et al., 2023) on 641 examples, SpatiaLab on 1400 problems (Wasi et al., 2026), OSWorld (Xie et al., 2024) on 369 problems, and MapEval (Dihan et al., 2025) on 700 tasks. These benchmarks demonstrate that for evaluation, *task difficulty, coverage, and annotation fidelity* are more decisive than sheer scale.

A defining feature of TIMESPOT is that each sample is annotated with *verifiable physical ground truth*, including exact temporal labels derived from solar geometry and precise geographic coordinates. Producing such annotations requires non-trivial computation, cross-field consistency checks, and human verification, making large-scale noisy expansion undesirable. Larger datasets often trade physical validity and diagnostic power for quantity, whereas TIMESPOT prioritizes correctness, interpretability, and the ability to expose structured reasoning failures.

The dataset exhibits natural imbalance across certain attributes, such as daylight phase (1,182 day images versus 273 night images), reflecting the physics and observability of real-world photography rather than a sampling artifact. Night scenes inherently lack many of the solar and shadow cues required for precise temporal inference, which is central to the task studied. Rather than enforcing artificial uniformity, this distribution allows us to examine model robustness under varying levels of cue availability, closely mirroring real-world deployment conditions.

TIMESPOT reliably separates model performance, exposing consistent differences across architectures, training strategies, and reasoning approaches in terms of geo-temporal accuracy, calibration, and physical consistency. This empirical separation demonstrates that the benchmark provides sufficient statistical signal for meaningful comparison. More broadly, TIMESPOT adopts an evaluation philosophy aligned with modern reasoning benchmarks, emphasizing verifiable annotations and diagnostic depth rather than raw scale, enabling trustworthy assessment of real-world geo-temporal understanding.

## G.2. Geographic Imbalance and Frequency-Weighted Accuracy

The geographic distribution of images in our dataset is uneven, with some countries contributing substantially more samples than others. To explicitly account for this imbalance and to clarify how per-country sample size affects country-level accuracy, we perform an additional analysis based on *frequency-weighted accuracy* stratified by per-country sample size. Specifically, we group countries into four bins according to their per-country sample size $n_c$: 1–10 images, 11–30 images, 31–60 images, and $\geq 61$ images.

Within each bin, we compute *frequency-weighted accuracy* by aggregating all examples from countries belonging to that bin. This aggregation reflects the effective evaluation setting in which countries contribute proportionally to their number of samples, while still allowing us to isolate how performance evolves across low-resource to high-resource regimes. Table 17 reports bin-wise, frequency-weighted accuracy (%) for all evaluated models.

*Table 17.* Bin-wise frequency-weighted accuracy (%) grouped by per-country sample size $n_c$.

| Per-country ($n_c$) | GPT5-Mini | Gemini-2.5-Flash | InternVL3-78B | LLaMA-3.2-11B | Qwen2.5-VL-32B | O4-Mini | GLM-4.1V-9B |
|---|---|---|---|---|---|---|---|
| 1–10 | 54.5 | 60.4 | 21.1 | 34.9 | 32.0 | 62.1 | 49.0 |
| 11–30 | 63.3 | 72.3 | 42.7 | 48.9 | 48.3 | 65.1 | 66.9 |
| 31–60 | 67.8 | 85.1 | 59.6 | 59.4 | 63.3 | 74.3 | 67.6 |
| $\geq 61$ | 82.3 | 86.8 | 77.9 | 72.4 | 75.8 | 81.9 | 83.3 |

Across all evaluated models, accuracy increases monotonically as we move from the 1–10 image bin to the $\geq 61$ image bin. This trend is consistent with the expected reduction in statistical variance as per-country sample size increases. Strong models such as Gemini-2.5-Flash, GPT5-Mini, O4-Mini, and GLM-4.1V-9B all exhibit this behavior, indicating that improved performance is not confined to a single model family nor limited to heavily represented countries. While low-resource countries naturally exhibit higher variance, the relative ordering and qualitative behavior of models remain stable across low-, medium-, and high-resource regimes. This suggests that the observed performance trends are not driven solely by a small number of high-frequency countries, but instead reflect consistent model behavior under varying degrees of geographic sparsity. Overall, this frequency-weighted, bin-wise analysis provides a clearer characterization of how geographic imbalance interacts with country-level accuracy and confirms that our conclusions are robust to uneven country distributions.

## G.3. Stability tests for frequency-binned accuracy

Here we perform statistical tests to verify the stability.

Let $\mathcal{M} = \{m_1, \ldots, m_M\}$ denote the set of evaluated models and let $\mathcal{B} = \{b_1, \ldots, b_K\}$ denote the per-country sample-size bins (here $K = 4$). For each bin $b_k$, we compute the frequency-weighted accuracy $a_m(b_k) \in [0, 100]$ for each model $m \in \mathcal{M}$ by aggregating all samples whose country belongs to $b_k$.

**Rank stability across bins (Spearman).** For each bin $b_k$, define the rank vector $r(b_k) \in \mathbb{R}^M$ where $r_m(b_k)$ is the rank of model $m$ when models are sorted by $a_m(b_k)$ in descending order (rank 1 is best; ties receive average ranks). To quantify whether the *relative ordering* of models is stable across bins, we compute Spearman's rank correlation between bins $b_i$ and $b_j$:

$$\rho_s(b_i, b_j) = \text{corr}(r(b_i),\, r(b_j)), \tag{1}$$

where $\text{corr}(\cdot, \cdot)$ is the Pearson correlation. High $\rho_s$ (near 1) indicates that model rankings are preserved under different country-frequency regimes. When reported with significance, the null hypothesis is $H_0 : \rho_s = 0$ (no association between rankings across bins).

We report $\rho_s$ between adjacent bins (and optionally first vs. last bin) as a direct measure of ranking stability, to validate that comparative conclusions are stable under geographic imbalance and that higher-resource countries yield systematically higher accuracy.

**Statistical validation of stability and trends.** To verify that our conclusions are robust to geographic imbalance, we conduct two complementary statistical tests on the bin-wise results in Table 17. Also, Table 18 shows that model rankings are stable across adjacent frequency bins. First, Spearman rank correlations between adjacent frequency bins are consistently

*Table 18.* Spearman rank correlation of model rankings across per-country frequency bins. High correlations between adjacent bins indicate stable relative model ordering as country sample size increases; lower correlation between extreme bins reflects higher variance in low-resource regimes.

| Bin Pair | Spearman $\rho$ | $p$-value |
|---|---|---|
| 1–10 vs. 11–30 | 0.821 | 0.023 |
| 11–30 vs. 31–60 | 0.786 | 0.036 |
| 31–60 vs. $\geq$ 61 | 0.821 | 0.023 |
| 1–10 vs. $\geq$ 61 | 0.607 | 0.148 |

high ($\rho = 0.79$–$0.82$, $p < 0.05$), indicating stable relative ordering of models as per-country sample size increases. The correlation between the lowest- and highest-resource bins is lower ($\rho = 0.61$, $p = 0.15$), reflecting increased variance in extremely low-resource regimes rather than a reversal of model rankings. Together, these results demonstrate that while absolute accuracy is sensitive to country frequency, the qualitative behavior and comparative ranking of models remain stable across low-, medium-, and high-resource geographic regimes.

**On statistical power and low-resource countries.** While per-country sample sizes are limited for some regions, our conclusions do not rely on country-specific point estimates. Instead, we analyze frequency-weighted accuracy aggregated across country-frequency bins and explicitly test stability and monotonicity. Spearman rank correlations show that relative model ordering is stable across adjacent bins ($\rho \approx 0.8$, $p < 0.05$). These results indicate that low-resource countries increase variance but do not introduce spurious trends or reverse comparative conclusions, mitigating the risk that individual samples or texture biases dominate the observed behavior.

## H. Improving Geo-Temporal Reasoning Capabilities

To enhance geo-temporal reasoning performance on TIMESPOT, we explore supervised fine-tuning (SFT).

### H.1. Supervised Fine-Tuning (SFT)

#### H.1.1. SETUP AND IMPLEMENTATION

To assess whether supervised adaptation improves joint geo-temporal reasoning, we fine-tuned the vision–language model `unsloth/Qwen2.5-VL-3B-Instruct` on TIMESPOT using a parameter-efficient supervised fine-tuning (SFT) protocol. Given the structured, multi-field prediction format of TIMESPOT, SFT was applied to encourage alignment between visual cues and explicit geo-temporal labels while maintaining the base model's general capabilities. As TIMESPOT evaluates 12 attributes spanning two core domains, we select representative tasks from each domain for supervised fine-tuning, each one separately: *country prediction* from the geo-location understanding domain, and *local time prediction accuracy* from the temporal understanding domain. These tasks capture complementary aspects of spatial categorization and continuous temporal inference, providing a focused yet informative view of SFT effects.

We employed Low-Rank Adaptation (LoRA) (Hu et al., 2021) to update a small subset of parameters while freezing the backbone. To operate within constrained GPU memory, the model was loaded in 4-bit precision, and LoRA adapters were inserted into both attention and feed-forward projection layers. Gradient checkpointing was enabled to further reduce memory usage. All experiments used fixed random seeds to ensure reproducibility. The LoRA configuration is summarized in Table 19.

*Table 19.* LoRA Hyperparameters for SFT

| Hyperparameter | Value | Notes |
|---|---|---|
| LoRA rank ($r$) | 16 | Low-rank update capacity |
| LoRA alpha | 16 | Scaling factor |
| LoRA dropout | 0 | No dropout applied |
| Gradient checkpointing | Enabled | Library-specific flag |
| LoRA random seed | 3407 | Adapter initialization |

Training examples were formatted using the native Qwen-2.5 chat template to preserve consistency between fine-tuning and evaluation. Each sample was serialized into a single instruction-following sequence predicting the full geo-temporal schema. Dynamic padding was applied via `DataCollatorForSeq2Seq`. We used a per-device batch size of 1 with gradient accumulation over 4 steps, yielding an effective batch size of 4. Optimization employed a paged AdamW optimizer with a linear learning-rate schedule. Full trainer and optimization settings are reported in Table 20.

*Table 20.* Trainer and Optimization Hyperparameters for SFT

| Hyperparameter | Value |
| --- | --- |
| Base model | unsloth/Qwen2.5-VL-3B-Instruct |
| Max sequence length | 2048 |
| Load in 4-bit | True |
| Per-device train batch size | 1 |
| Gradient accumulation steps | 4 |
| Effective batch size | 4 |
| Learning rate | 2e-4 |
| Optimizer | paged_adamw_8bit |
| Weight decay | 0.01 |
| LR scheduler | linear |
| Logging steps | 1 |
| Seed | 42 |
| Tokenizer template | Qwen-2.5 |

Prompts for SFT are available in Figure 8 and 9.

### SFT: Prompt for Answering Model (Country)

You are a geo-spatial-temporal understanding assistant. From the given image, give a country from where the image you think belongs, in an exact format below.
Format:
  - country : value
Give an answer regardless of certainty. Do not refuse.

### SFT: Prompt for Judging Model (Country)

You are a strict evaluator.
You will receive the ground truth and a model's answer (both in the same bullet-list format).
Compare country. Treat abbreviations and long forms as equivalent (for example: USA == United States).
Return a JSON object where each field maps to a nested object:
{
 "country": {
  "ground_truth": "<ground truth string>",
  "model_ans": "<model answer string>",
  "evaluation": 1 or 0
 },
}
Include "N/A" if the model answer is malformed. Output only valid JSON.

*Figure 8.* Prompts for SFT (Country).

Fine-tuning was conducted on a stratified 40% subset of TIMESPOT, ensuring balanced coverage across continents, climate zones, and temporal conditions. Training proceeded for four epochs, with a checkpoint saved after each epoch and evaluated on the held-out 60% split. This design allows us to isolate the effects of supervised adaptation on geo-temporal inference while avoiding leakage from uneven geographic distributions. Overall, this setup enables efficient supervised adaptation of a mid-scale VLM for structured geo-temporal prediction, providing a controlled testbed for analyzing the strengths and limitations of SFT on physically grounded reasoning tasks.

---

## SFT: Prompt for Answering Model (Time)

You are a geo-spatial-temporal understanding assistant. From the given image, try to provide the exact local time when you think the image is taken, in an exact format below.
Format:
- time_of_day : HH:MM (in 24-hour local time)
Give an answer regardless of certainty. Do not refuse.

## SFT: Prompt for Judging Model (Time)

You are a strict evaluator.
You will receive two inputs in the same bullet list format:
1. Ground truth time
2. Model predicted time

Your task is to compare the times.

Scoring rule:
- Assign a score of 1 if the model answer is within plus or minus 1 hour or 60 minutes of the ground truth.
- Assign a score of 0 otherwise.

Examples:
- Ground truth: 5:40, model answer: 5:20 → score 1
- Ground truth: 5:40, model answer: 6:40 → score 1
- Ground truth: 5:40, model answer: 4:30 → score 0
- Ground truth: 5:40, model answer: 7:00 → score 0

Always include the time evaluation, even if the model answer is malformed. Malformed means score 0.

Return a JSON object where each field maps to a nested object:
{
  "time": {
    "ground_truth": "<ground truth string>",
    "model_ans": "<model answer string>",
    "evaluation": 1 or 0
  },

}
Output only valid JSON.

*Figure 9.* Prompts for SFT (Time).

### H.1.2. SUPERVISED FINE-TUNING (SFT) PERFORMANCE

Figure 4 summarizes the effect of supervised fine-tuning on TIMESPOT across training epochs. Starting from the pretrained baseline (epoch 0), SFT yields consistent improvements in *country accuracy*, rising from 14.20% to 19.24% by epoch 4, with performance remaining stable thereafter. This trend indicates that supervised adaptation effectively strengthens categorical geo-semantic discrimination, enabling the model to better associate visual cues with country-level identity.

Temporal prediction shows a different trajectory. *Time accuracy* improves rapidly in early epochs, increasing from 20.27% to 23.37% after a single epoch, and continues to fluctuate upward to 24.79% by epoch 5. Unlike country accuracy, temporal performance exhibits non-monotonic behavior across epochs, suggesting that SFT captures some coarse temporal regularities but struggles to consistently refine fine-grained time inference from illumination, shadow geometry, and solar dynamics.

The asymmetric gains between spatial and temporal fields highlight an important distinction in learnability. Country prediction benefits more directly from supervised exposure, likely because it aligns with discrete, semantically clustered visual patterns that are well represented in pretraining. In contrast, local-time prediction requires continuous reasoning over physical cues that are weakly supervised and more sensitive to noise, making improvements less stable under standard SFT

objectives.

Overall, these results show that SFT provides a meaningful and reliable boost to geo-temporal performance on TIMESPOT, particularly for structured categorical outputs. At the same time, the fluctuating temporal gains indicate that fine-tuning alone is insufficient to fully resolve continuous temporal grounding. This positions SFT as a valuable baseline enhancement, while motivating future work on complementary approaches such as physics-informed inductive biases, auxiliary supervision for solar geometry, or architecture-level mechanisms designed for temporal reasoning.

### H.1.3. CROSS-TASK EVALUATION

To assess whether single-task SFT generalizes across domains, we evaluate each fine-tuned checkpoint on the complementary task. As shown in Table 24, fine-tuning on country prediction reduces time accuracy from 22.06% to 21.78%, while fine-tuning on time prediction reduces country accuracy from 13.47% to 12.98%. This interference reflects that country prediction requires invariance to illumination (for lighting-robust place identity), while time prediction requires sensitivity to illumination (for shadow and sky state inference). Under shared LoRA parameters, these competing gradient directions reduce task-specific specialization.

*Table 21.* Cross-task SFT evaluation on TIMESPOT (Qwen-VL2.5-3B-Instruct). Each row shows a model fine-tuned on one task and evaluated on both tasks at epoch 4.

| Fine-tuned on | Country Ac. (%) | Time Ac. (%) |
|---|---|---|
| Baseline (no SFT) | 13.47 | 22.06 |
| Country | 19.24 | 21.78 |
| Time | 12.98 | 24.79 |

**Joint SFT.** To assess whether simultaneous optimization of spatial and temporal objectives reduces interference, we fine-tune a single model jointly on country and time prediction for 5 epochs, repeated across 3 runs to assess stability. Table 22 summarizes the averaged results. Joint training improves over the zero-shot baseline (country: $14.23\% \rightarrow 15.72\%$; time: $20.27\% \rightarrow 22.36\%$), but falls below the single-task peaks for both objectives (country: 19.24%; time: 24.79%). Time accuracy is more volatile across runs than country accuracy, consistent with the higher sensitivity of illumination cues to gradient noise. These results indicate that joint optimization under LoRA-based adaptation is limited by capacity constraints and competing gradient directions, suggesting that future work should explore gradient-disentangled or task-modular fine-tuning (Buschoff et al., 2025; Binz et al., 2025).

*Table 22.* Joint SFT results on TIMESPOT (Qwen-VL2.5-3B-Instruct, averaged over 3 runs). Accuracy reported at end of each epoch.

| Epoch | Avg Country (%) | Avg Time (%) |
|---|---|---|
| 0 | 14.23 | 20.27 |
| 1 | 14.80 | 20.62 |
| 2 | 15.14 | 20.80 |
| 3 | 15.10 | 21.86 |
| 4 | 15.46 | 21.90 |
| 5 | 15.72 | 22.36 |

Future work should consider RL-based training strategies (Chen et al., 2025) that allow more flexible task-specific reasoning rather than enforcing shared representations through SFT alone.

### H.2. Joint Supervised Fine-Tuning

Single-task SFT treats country and time prediction as independent objectives, which leaves open the question of whether joint optimization over both tasks yields a more capable geo-temporal model. To investigate this, we fine-tune `unsloth/Qwen2.5-VL-3B-Instruct` simultaneously on country and time prediction under the same LoRA configuration and data split described in §H.1, extending training to five epochs. To assess result stability, we repeat the experiment across three independent runs (differing only in data ordering) and report per-run and averaged results.

**Results.** Table 23 reports per-run and averaged accuracy for each epoch, with learning curves shown in Figures 10 and 11. Joint training improves over the zero-shot baseline on both tasks: averaged country accuracy rises from 14.23% to 15.72%, and averaged time accuracy rises from 20.27% to 22.36% by epoch 5. However, both values fall substantially below the corresponding single-task peaks (country: 19.24%; time: 24.79%), indicating a clear cost to joint optimization.

*Table 23.* Joint SFT results on TIMESPOT (`Qwen2.5-VL-3B-Instruct`). Each row reports country accuracy (%) and time accuracy (%) at the end of the corresponding epoch for three independent runs and their average. Epoch 0 is the pretrained zero-shot baseline (identical across runs).

| Epoch | Run 1 | | Run 2 | | Run 3 | | Average | |
|---|---|---|---|---|---|---|---|---|
| | Country | Time | Country | Time | Country | Time | Country | Time |
| 0 | 14.23 | 20.27 | 14.23 | 20.27 | 14.23 | 20.27 | 14.23 | 20.27 |
| 1 | 14.36 | 20.82 | 14.64 | 20.96 | 15.40 | 20.07 | 14.80 | 20.62 |
| 2 | 14.57 | 20.48 | 15.05 | 20.62 | 15.81 | 21.31 | 15.14 | 20.80 |
| 3 | 14.57 | 22.27 | 15.33 | 21.44 | 15.40 | 21.86 | 15.10 | 21.86 |
| 4 | 15.19 | 21.99 | 15.40 | 21.99 | 15.81 | 21.72 | 15.46 | 21.90 |
| 5 | 15.81 | 22.34 | 15.53 | 22.54 | 15.81 | 22.20 | 15.72 | 22.36 |

**Cross-task evaluation.** Table 24 compares the zero-shot baseline, single-task SFT, and joint SFT on both metrics simultaneously. When fine-tuned on country alone, time accuracy drops from 22.06% to 21.78%; when fine-tuned on time alone, country accuracy drops from 13.47% to 12.98%. Joint SFT recovers both tasks above their degraded single-task cross-evaluation values but remains below single-task peaks.

*Table 24.* Cross-task evaluation on TIMESPOT. Each row shows a model evaluated on both country and time accuracy at the checkpoint with the best task-specific performance. Joint SFT values are averages over three runs at epoch 5.

| Training objective | Country Ac. (%) | Time Ac. (%) |
|---|---|---|
| Baseline (no SFT) | 13.47 | 22.06 |
| Single-task: Country | 19.24 | 21.78 |
| Single-task: Time | 12.98 | 24.79 |
| Joint SFT (avg, ep. 5) | 15.72 | 22.36 |

**Analysis.** The performance gap between joint and single-task SFT reflects a structural conflict between the two objectives. Country prediction requires the model to be invariant to illumination: the same country must be recognized across daytime, nighttime, and seasonal conditions. Time prediction requires the opposite: sensitivity to illumination-dependent signals such as shadow length, sun elevation, sky luminance, and color temperature. Under LoRA-based adaptation, both objectives share the same low-rank update matrices. Satisfying their competing gradient directions simultaneously forces a compromise that suppresses task-specific specialization in both domains.

The asymmetry between country and time learning curves provides further evidence of this conflict. Country accuracy improves slowly and saturates early across all runs, indicating that the model struggles to acquire illumination-invariant place identity when the time objective simultaneously pulls adapter weights toward illumination sensitivity. Time accuracy, by contrast, shows a smoother upward trend but still falls below its single-task ceiling, reflecting that illumination-sensitive updates are partially dampened by the simultaneous country objective.

**Implications and future directions.** These findings are consistent with known limitations of standard SFT on heterogeneous objectives (Buschoff et al., 2025; Binz et al., 2025): enforcing a single shared representation tends to suppress the task-specific reasoning patterns that each objective individually supports. RL-based training approaches such as GRPO-style optimization (Chen et al., 2025) could mitigate this by assigning per-task outcome rewards, allowing the model to develop more flexible reasoning strategies without being constrained to a single intermediate representation. Task-modular architectures with separate adapters per objective, or curriculum strategies that alternate between tasks at the gradient level, represent additional directions worth exploring in future work. Overall, joint SFT provides a useful multi-task baseline

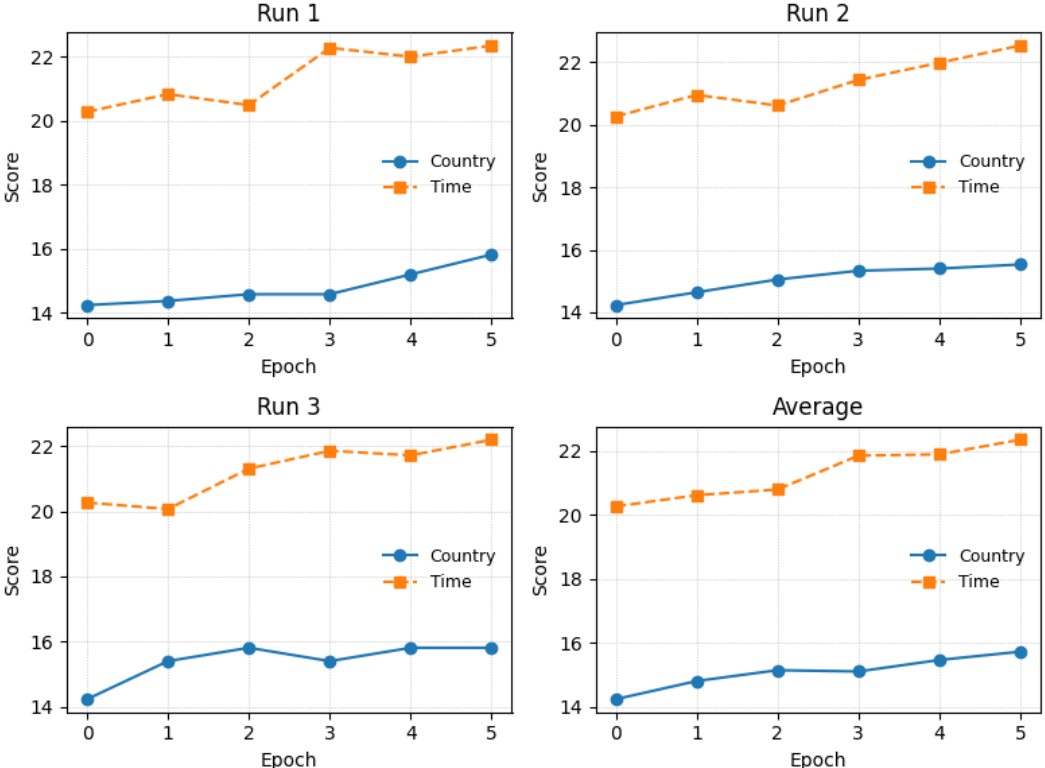

*Figure 10.* Joint SFT learning curves per run on TIMESPOT (`Qwen2.5-VL-3B-Instruct`). Each panel shows country accuracy (blue, solid) and time accuracy (orange, dashed) across epochs for one of the three independent runs, plus the averaged curves (bottom-right). Country accuracy improves slowly and saturates early in all runs; time accuracy shows a smoother but more variable upward trend, reflecting the competing gradient directions induced by illumination-invariant and illumination-sensitive objectives sharing the same LoRA parameters.

that confirms both tasks are learnable under shared supervision, while the gap relative to single-task training quantifies the interference cost and motivates more principled multi-task training designs.

# I. Current Limitations and Future Research Directions

This appendix consolidates the key limitations revealed by TIMESPOT and outlines concrete research directions motivated by empirical error patterns across geography, climate, seasonality, daylight phases, and supervised fine-tuning.

### I.1. Limitations of Current Vision–Language Models

**Weak temporal grounding and physical inconsistency.** Across models, temporal reasoning is substantially weaker than spatial categorization. Predictions of local time, season, and daylight phase frequently violate basic physical constraints, including solar elevation limits, hemisphere–season alignment, and daylight–time compatibility. Errors are especially pronounced at night, during twilight, and in high-latitude or high-altitude settings, indicating that current VLMs underutilize continuous physical cues such as shadow geometry, sky luminance gradients, and seasonal illumination dynamics.

**Over-reliance on coarse spatial priors.** Spatial predictions often succeed at continent or broad regional levels while failing at finer scales, leading to systematic neighboring-country substitutions and near-miss geography. Models rely heavily on semantic and stylistic cues such as architectural appearance, vegetation type, and text-like artifacts, rather than metric grounding in latitude, longitude, coastline geometry, or elevation. This reliance produces heavy-tailed spatial errors that remain hidden when only mean accuracy is reported.

**Climate, season, and daylight asymmetries.** Performance varies sharply across climate zones, seasons, and daylight phases. Tropical, Arid, and Temperate regions are comparatively easy, while Continental and Polar zones remain highly

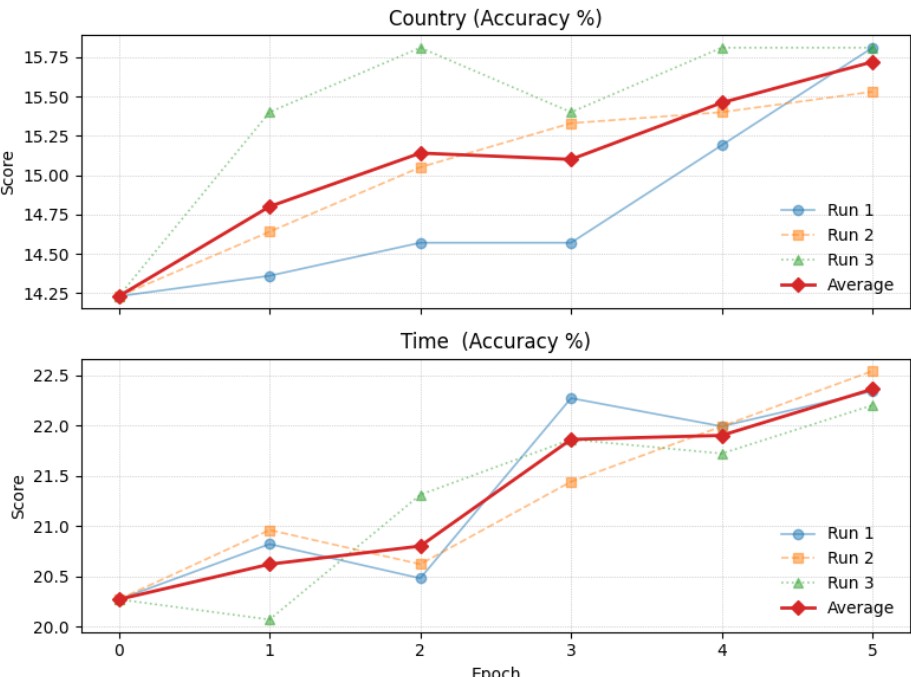

*Figure 11.* Joint SFT learning curves aggregated across three runs on TIMESPOT. Top: country accuracy (%) per run and average. Bottom: time accuracy (%) per run and average. The average curve (red, bold) shows that country accuracy saturates near 15.7% while time accuracy reaches 22.4% by epoch 5, both well below the corresponding single-task peaks of 19.24% and 24.79%, quantifying the interference cost of joint optimization under shared LoRA parameters.

error-prone. Summer is inferred reliably, but spring and winter are substantially harder, and autumn collapses entirely across all models, exposing a major weakness in phenological and color-based reasoning. Daylight phase prediction shows strong specialization but poor robustness across the diurnal cycle, with night and sunrise consistently challenging.

**Sensitivity to scene conditions and data imbalance.** Urban canyons, low-light conditions, twilight scenes, and heterogeneous inland regions amplify both spatial and temporal failures. Geographic imbalance further increases variance for low-resource countries, though our stability analysis shows that relative model ordering remains largely consistent. Nevertheless, small per-country sample sizes limit the reliability of country-specific point estimates and highlight the importance of aggregation and diagnostic metrics.

**Limits of supervised fine-tuning.** Supervised fine-tuning provides consistent gains for categorical geo-semantic prediction but yields smaller and less stable improvements for continuous temporal inference. Single-task SFT induces cross-task interference: country fine-tuning reduces time accuracy and vice versa, because the two tasks require competing visual sensitivities under shared LoRA parameters. Joint SFT partially mitigates this but falls below single-task peaks due to gradient conflict. SFT tends to reinforce frequent patterns aligned with structured labels, while struggling to induce robust reasoning over solar geometry and illumination physics, motivating architectural and training-level interventions beyond standard fine-tuning (Chen et al., 2025; Buschoff et al., 2025).

### I.2. Future Research Directions

**Explicit physical inductive biases.** Future models should incorporate explicit representations of solar geometry, seasonal cycles, and illumination physics, for example by conditioning predictions on latitude, day-of-year, and sun–earth relationships or by introducing auxiliary heads for sun elevation, horizon detection, and shadow orientation. Such inductive biases could substantially reduce temporal inconsistencies and twilight-related failures.

**Joint, constraint-aware reasoning.** Our findings motivate decoding and training strategies that enforce soft or hard consistency constraints across geo-temporal fields, such as month–season–hemisphere alignment and daylight–time compatibility. Constraint-aware objectives and structured decoding could prevent physically implausible combinations even when

individual cues are ambiguous.

**Improved supervision for continuous temporal variables.** Temporal inference requires learning continuous relationships rather than discrete categories. Future work should explore regression-based objectives, uncertainty-aware losses, and curriculum strategies that progressively refine temporal resolution, as well as richer supervision derived from ephemerides and environmental simulation.

**Micro-geographic and environmental cues.** Reducing neighboring-country confusions will likely require explicit modeling of micro-geo signals such as signage typography, license plates, road furniture, coastline topology, and elevation cues. Incorporating geo-linguistic and topographic priors, either through auxiliary tasks or specialized modules, could improve fine-grained spatial grounding.

**Data diversity and augmentation.** Climate- and altitude-aware augmentations, photometric normalization for extreme lighting regimes, and balanced sampling across seasons and latitudes are critical to improving robustness. In particular, autumn and high-latitude winter scenes require targeted data curation to address current blind spots.

**Beyond fine-tuning: architectural and training innovations.** While SFT improves baseline performance, our results indicate that deeper progress will require architectural changes and training paradigms that explicitly model time as part of world representation. Promising directions include hybrid perception–physics models, modular temporal reasoning components, and multi-task training that couples spatial perception with temporal prediction. Our cross-task interference findings also motivate RL-based approaches such as GRPO-style training (Chen et al., 2025), which can support more flexible, task-aware reasoning strategies than SFT by operating on outcome-level rewards rather than enforcing shared intermediate representations. Multi-task SFT with explicit task routing or gradient-isolated adapters (Buschoff et al., 2025; Binz et al., 2025) is another promising direction.

Overall, TIMESPOT reveals that geo-temporal reasoning failures are systematic rather than incidental. Addressing them will require moving beyond static scene recognition toward models that explicitly reason about the physical processes governing how the world changes over space and time.

## J. Human Performance and Comparison

### J.1. Human Evaluation Protocol and Observed Patterns

We conducted a controlled human evaluation of TIMESPOT using six participants divided into two cohorts: three undergraduate students (ages 20–22, generalists with no formal domain specialization) and three domain experts (Master's students and senior PhD researchers with backgrounds in geomorphology, climatology, and geosciences). All participants completed the full benchmark under identical conditions, and their responses were evaluated using the same metrics as the VLMs.

The undergraduate cohort demonstrates moderate performance with strengths in coarse environmental recognition and daylight perception, but exhibits clear limitations in fine-grained localization and temporal precision. In contrast, the expert cohort consistently achieves high accuracy across both geo-spatial and temporal tasks, with particularly strong performance in climate classification, season inference, and time estimation.

Across both groups, two general patterns emerge: (i) human performance improves systematically with domain knowledge, especially for tasks requiring physical and environmental reasoning, and (ii) humans exhibit lower variance and more consistent cross-task performance compared to models, indicating a unified underlying reasoning framework rather than task-specific heuristics.

### J.2. Detailed Comparison

The comparative evaluation between Vision-Language Models (VLMs) (Table 3) and human cohorts (Table 25) on TIMESPOT reveals systematic and interpretable differences in how artificial and biological systems process geo-temporal information. While models demonstrate strong performance in discrete classification and large-scale memorization tasks, humans consistently outperform in domains requiring causal reasoning, physical intuition, and temporal inference. Below, we present a structured analysis across six key dimensions.

**Geopolitical Recognition vs. Human Heuristics.** VLMs exhibit a clear advantage in country-level classification. The best-performing proprietary and reasoning models achieve accuracies exceeding 75%, surpassing both undergraduate (45.98%) and expert (67.89%) human averages. This performance reflects the models' ability to encode high-resolution

*Table 25.* Performance of VLMs and Projected Human Baselines on **TimeSpot** by questions. **Cnt.** → Continent, **Cou.** → Country, **Clim.** → Climate Zone; **Env.** → Environment Type, **Lat.°** → Latitude in degree, **Long.°** → Longitude in degree, **Dist.(km)** (MD) → mean distance from actual location in kilometers, **DLP** → Day-light phase. **Time** (Ac.) denotes accuracy, if the model predicted the time accurately within 1 hour window. **Time** (MAE) shows mean error in HH:MM format. Ac. denotes accuracy.

| Model / Subject | Geo-location Understanding | | | | | | | Temporal Understanding | | | | |
| --- | --- | --- | --- | --- | --- | --- | --- | --- | --- | --- | --- | --- |
| | Cnt. | Cou. | Clim. | Env. | Lat.° | Long.° | Dist.(km) | Season | Month | Time | Time | DLP |
| | Ac.(↑) | Ac.(↑) | Ac.(↑) | Ac.(↑) | MAE (↓) | MAE (↓) | MD (↓) | Ac.(↑) | Ac.(↑) | Ac.(↑) | MAE (↓) | Ac.(↑) |
| *Human Baselines* | | | | | | | | | | | | |
| Undergrad 1 (Generalist) | 81.57 | 46.28 | 68.56 | 75.04 | 12.57 | 32.16 | 2750.47 | 68.58 | 28.56 | 42.14 | 2:41 | 78.56 |
| Undergrad 2 (Traveler) | 84.06 | 52.18 | 71.26 | 78.57 | 11.26 | 28.57 | 2450.18 | 72.06 | 31.07 | 45.57 | 2:26 | 81.07 |
| Undergrad 3 (Less Traveled) | 77.08 | 39.56 | 64.07 | 73.58 | 14.86 | 38.46 | 3200.86 | 66.08 | 25.57 | 38.06 | 2:56 | 74.07 |
| Average (Undergrad) | 80.89 | 45.98 | 67.96 | 75.71 | 12.89 | 33.06 | 2800.49 | 68.89 | 28.39 | 41.92 | 2:41 | 77.89 |
| Expert 1 (Geomorphology) | 95.56 | 68.57 | 85.06 | 88.56 | 4.56 | 11.57 | 980.57 | 84.57 | 44.06 | 58.06 | 1:36 | 92.56 |
| Expert 2 (Climatology) | 92.07 | 64.06 | 89.57 | 86.07 | 5.87 | 14.27 | 1250.27 | 89.06 | 48.57 | 54.57 | 1:46 | 90.07 |
| Expert 3 (Field Geologist) | 94.57 | 71.08 | 84.56 | 87.56 | 5.06 | 10.86 | 890.47 | 86.07 | 45.57 | 61.07 | 1:26 | 94.07 |
| Average (Expert) | 94.06 | 67.89 | 86.39 | 87.39 | 5.16 | 12.22 | 1040.42 | 86.56 | 46.06 | 57.89 | 1:36 | 92.22 |

statistical associations between visual patterns and geopolitical labels.

Humans, in contrast, rely on coarse semantic cues such as language, architectural style, or cultural artifacts. These cues are often insufficient for distinguishing between geographically or culturally similar regions. Consequently, human errors tend to cluster at geopolitical boundaries, where visual differences are subtle or non-existent.

This indicates that VLMs operate as high-capacity retrieval systems for geographically indexed visual features, whereas human reasoning is constrained by semantic abstraction and limited exposure to global diversity.

**Environmental Semantics and Climate Reasoning.** When the task shifts to climate zone and environment classification, the performance gap reverses. Human experts achieve 86.39% accuracy in climate classification, significantly exceeding all model categories, where the best results plateau around 73%. This difference highlights the importance of domain knowledge and multi-scale reasoning. Humans integrate vegetation structure, soil characteristics, and atmospheric conditions into coherent environmental interpretations. These features are continuous and interdependent, requiring abstraction beyond pixel-level correlations. VLMs, despite strong performance in environment type (up to 64.47%), struggle with climate categorization due to the absence of explicit supervision linking visual features to scientific taxonomies. This suggests that current models lack robust internal representations of ecological systems.

**Metric Localization and Error Distribution.** Latitude, longitude, and mean distance (MD) provide insight into spatial precision. Top VLMs achieve lower MD values (e.g., 892.54 km) compared to experts (1040.42 km) and significantly outperform undergraduates (2800.49 km). However, this advantage must be interpreted cautiously. Model predictions are often anchored to high-confidence country-level guesses, leading to outputs near geographic centroids. This reduces average error but does not necessarily reflect true spatial reasoning. In contrast, human errors are less frequent but more dispersed, resulting in higher variance when incorrect. Notably, expert humans achieve latitude MAE ($5.16°$) comparable to leading models ($\sim 3°–4°$), suggesting that humans employ physically grounded estimation strategies, such as solar elevation and vegetation gradients.

**Temporal Inference and Solar Geometry.** Temporal prediction represents the most significant failure mode for VLMs. Across all models, time-of-day accuracy remains below 35%, with mean absolute errors consistently around 4 hours. In contrast, undergraduates achieve 2:41 MAE, and experts reduce this further to 1:36. This gap reflects the absence of explicit physical reasoning in current VLM architectures. Accurate time estimation requires interpreting shadow orientation, shadow length, atmospheric scattering, and human activity patterns. These cues are governed by deterministic physical laws, particularly solar geometry. Humans naturally integrate these signals into a coherent temporal estimate, while models treat them as weak statistical features. The resulting errors often manifest as systematic ambiguities, such as confusion between morning and late afternoon lighting conditions.

**Seasonality and Biological Signals.** Season classification further exposes differences in temporal reasoning. Human experts achieve 86.56% accuracy, substantially outperforming all models (best $\sim 65\%$). Even undergraduates (68.89%) remain competitive with top reasoning models. Season identification depends on phenological cues, including vegetation cycles, snow cover, and atmospheric clarity. These signals require longitudinal understanding of environmental change, which humans acquire through experience and education. Models, lacking explicit temporal grounding, rely on superficial cues

such as color distributions. This leads to systematic weaknesses, particularly in transitional seasons where visual differences are subtle and context-dependent. Month prediction remains challenging for both groups due to the increased granularity (12 classes), but humans still maintain a consistent advantage, especially at the expert level ($46.06\%$ vs. $\sim 24\%$ for top models).

**Daylight Phase and Perceptual Coherence.** Daylight phase (DLP) accuracy provides a coarse measure of temporal understanding. Humans again demonstrate a strong advantage, with experts reaching $92.22\%$ and undergraduates $77.89\%$, compared to a maximum of $64.09\%$ among models. This task requires integrating global illumination characteristics, including color temperature, shadow softness, and sky luminance gradients. These features are perceptually salient to humans but remain weakly encoded in current VLM representations. The discrepancy suggests that human perception is inherently calibrated to natural lighting conditions, enabling robust classification even under ambiguous scenarios.

The results collectively indicate a fundamental distinction between two modes of intelligence:

- **VLMs:** High-dimensional statistical systems optimized for pattern recognition and large-scale memorization. Their strengths lie in discrete classification tasks with strong visual-textual correlations.

- **Humans:** Physically grounded reasoning systems capable of integrating spatial, temporal, and environmental cues into coherent world models.

VLMs excel in tasks that resemble retrieval from a global visual database, such as country recognition. However, they lack the causal and geometric reasoning required for accurate temporal inference and environmental understanding. In contrast, humans interpret images as projections of dynamic physical processes. This enables robust performance in tasks involving time, climate, and environmental context, even with limited exposure to specific locations. Bridging this gap will require future models to incorporate explicit representations of physical laws, temporal dynamics, and environmental systems, moving beyond static visual pattern matching toward integrated spatio-temporal reasoning.

## K. Examples from TIMESPOT

Here we provide examples from TIMESPOT with image, ground truth and predictions of three models: *Intern-VL3.5-2B, GPT-5-mini,* and *Intern-VL2.5-72B* with correct (green) and wrong (red) markings.

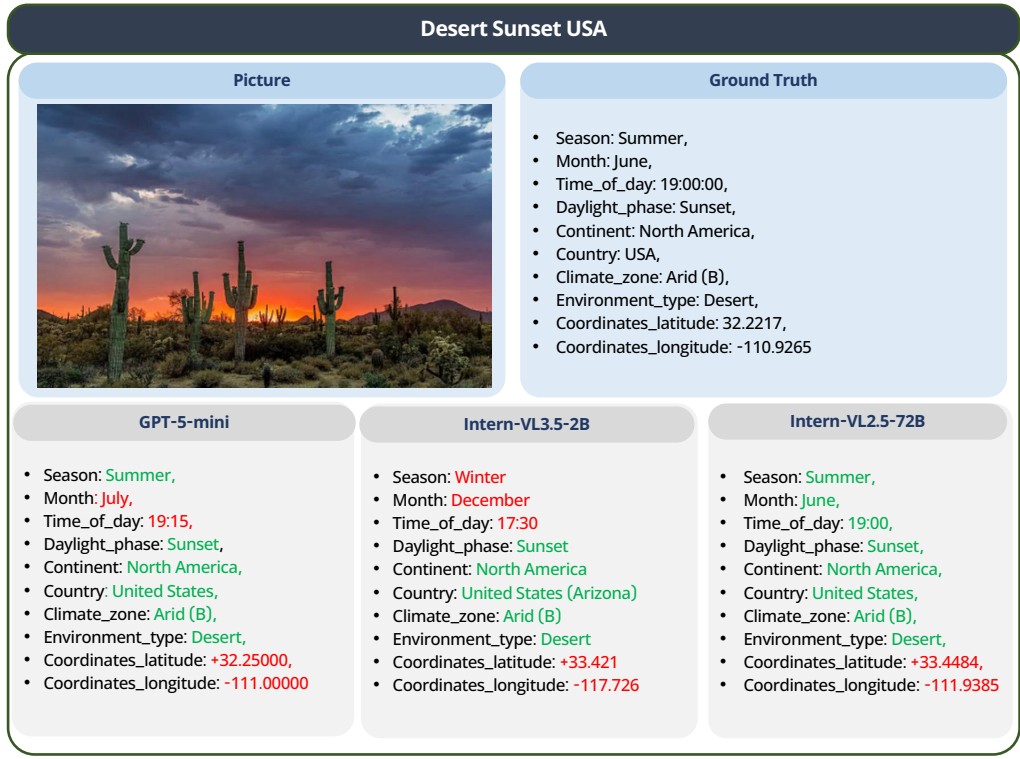

*Figure 12.* Example of TIMESPOT dataset: Desert Sunset in USA.

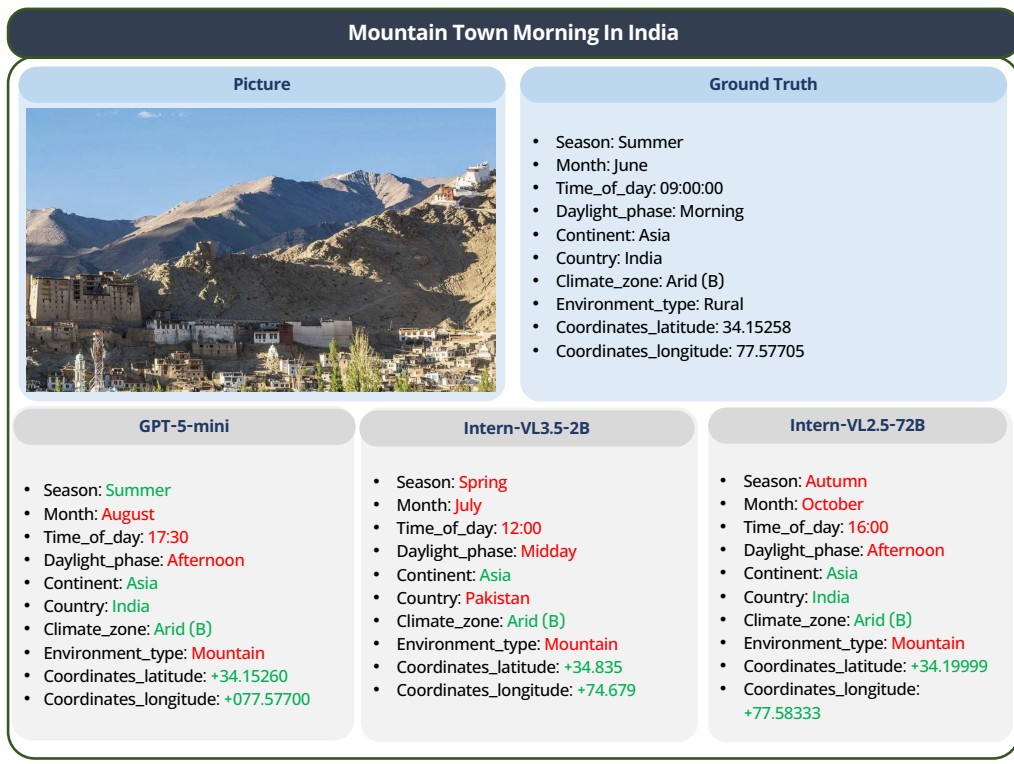

*Figure 13.* Example of TIMESPOT dataset: Mountain Town Morning in India.

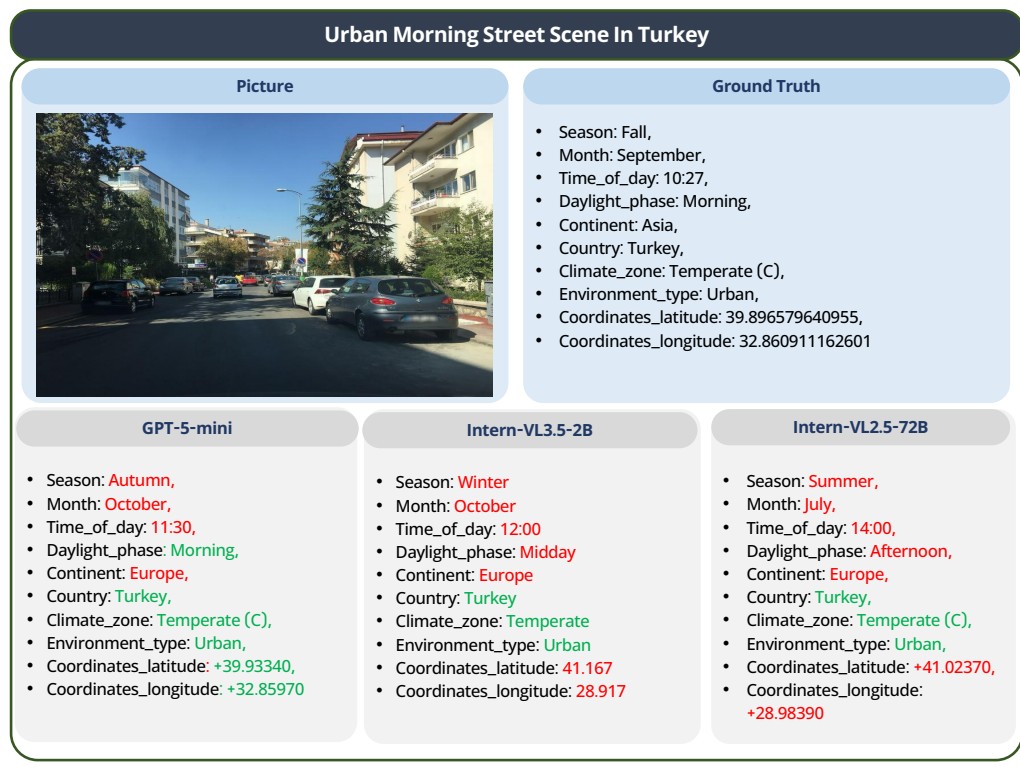

*Figure 14.* Example of TIMESPOT dataset: Urban Morning Street Scene in Turkey.

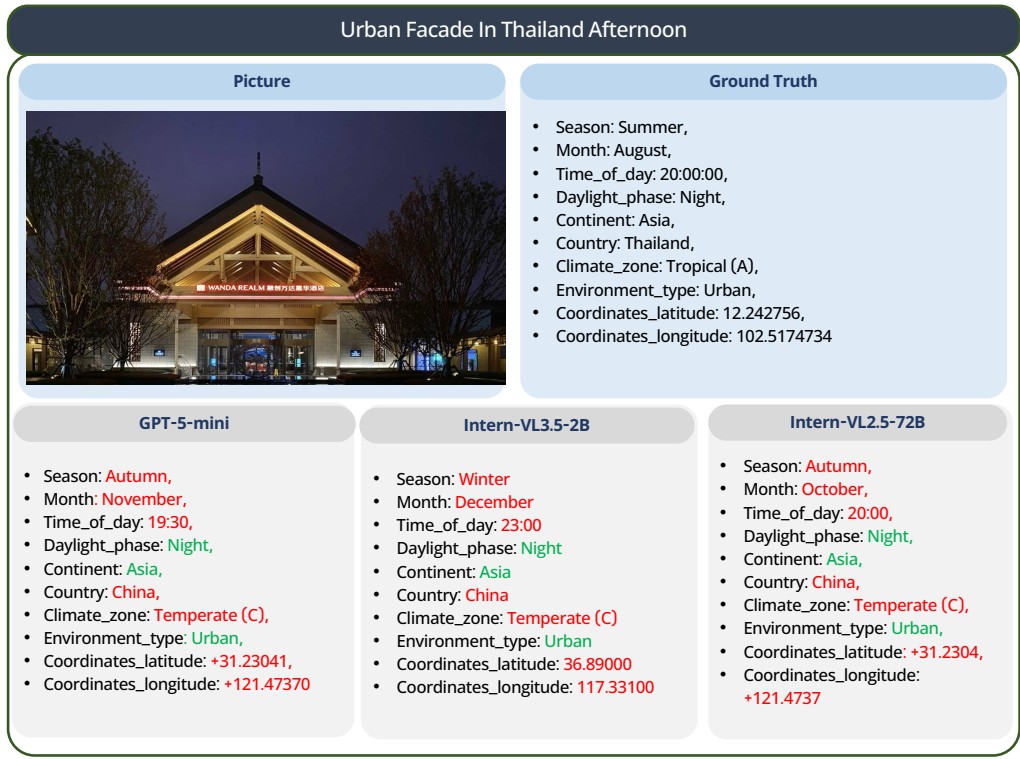

*Figure 15.* Example of TIMESPOT dataset: Urban Facade in Thailand (Afternoon).

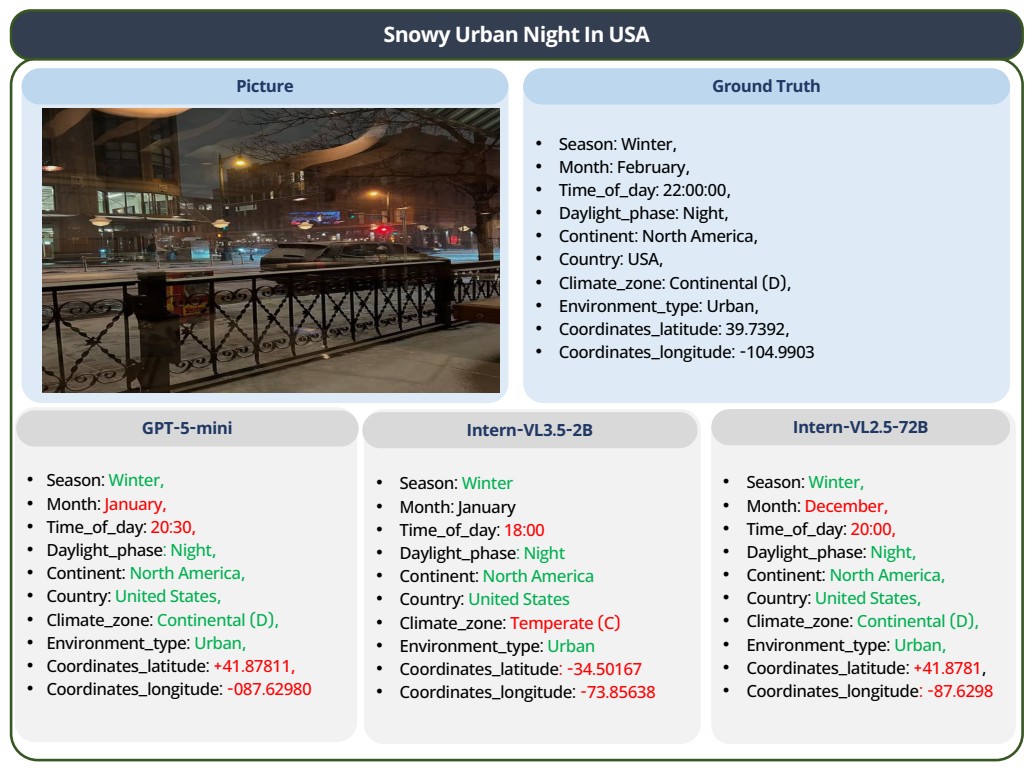

*Figure 16.* Example of TIMESPOT dataset: Snowy Urban Night in USA.

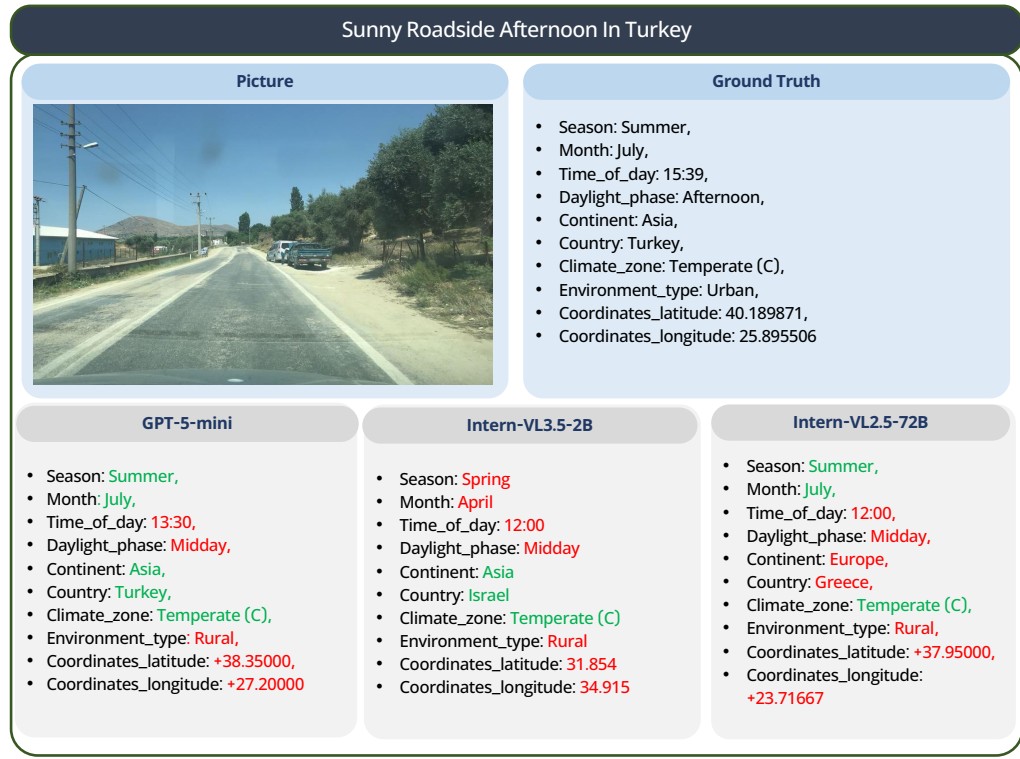

*Figure 17.* Example of TIMESPOT dataset: Sunny Roadside Afternoon in Turkey.

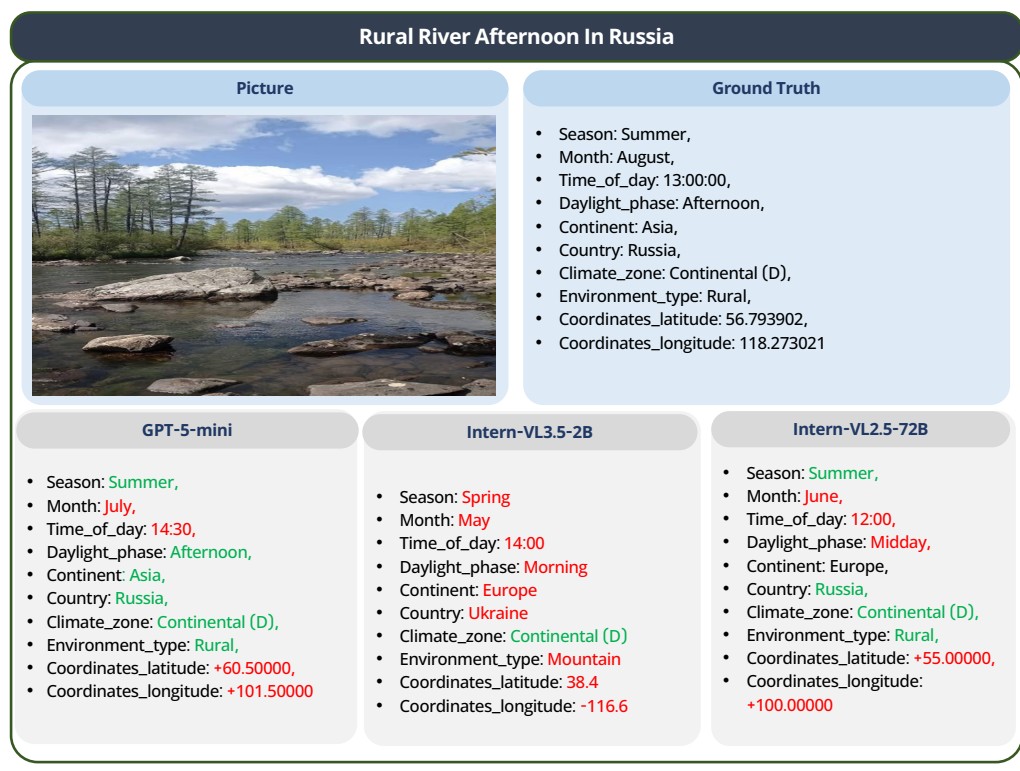

*Figure 18.* Example of TIMESPOT dataset: Rural River Afternoon in Russia.

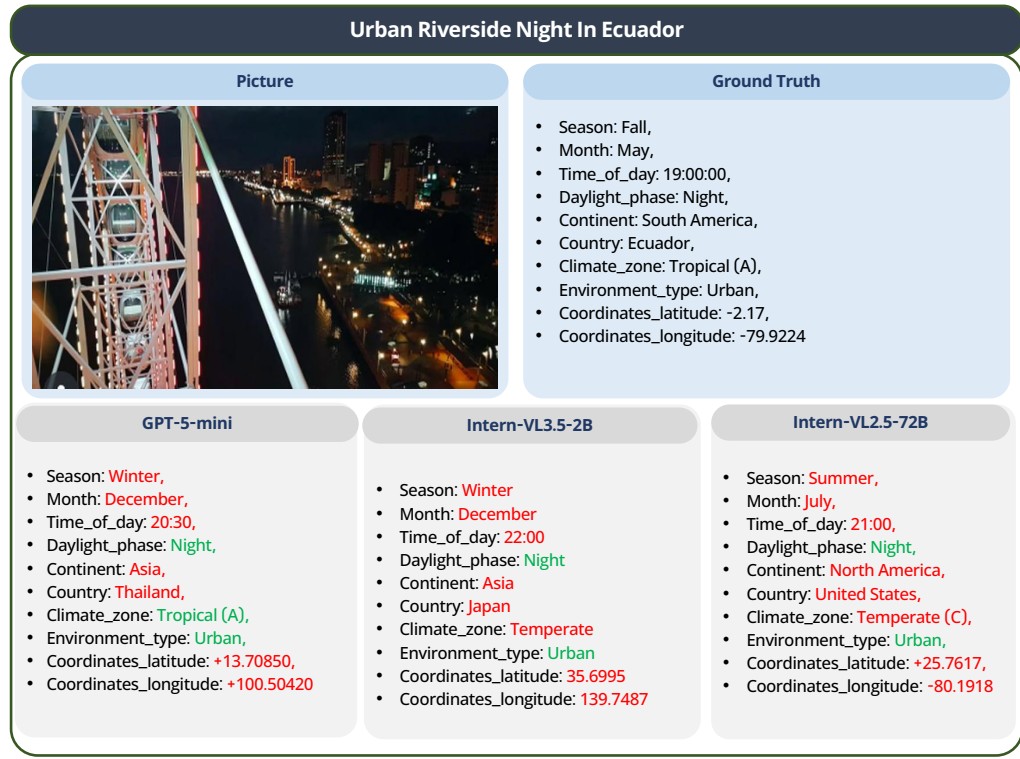

*Figure 19.* Example of TIMESPOT dataset: Urban Riverside Night in Ecuador.

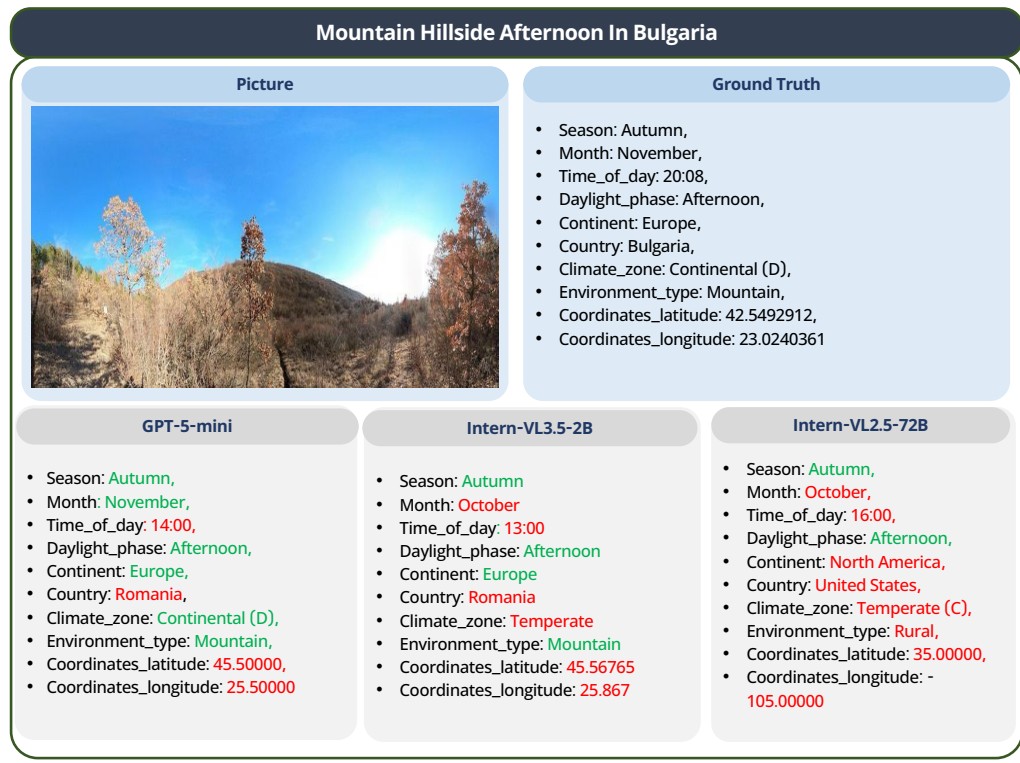

*Figure 20.* Example of TIMESPOT dataset: Mountain Hillside Afternoon in Bulgaria.

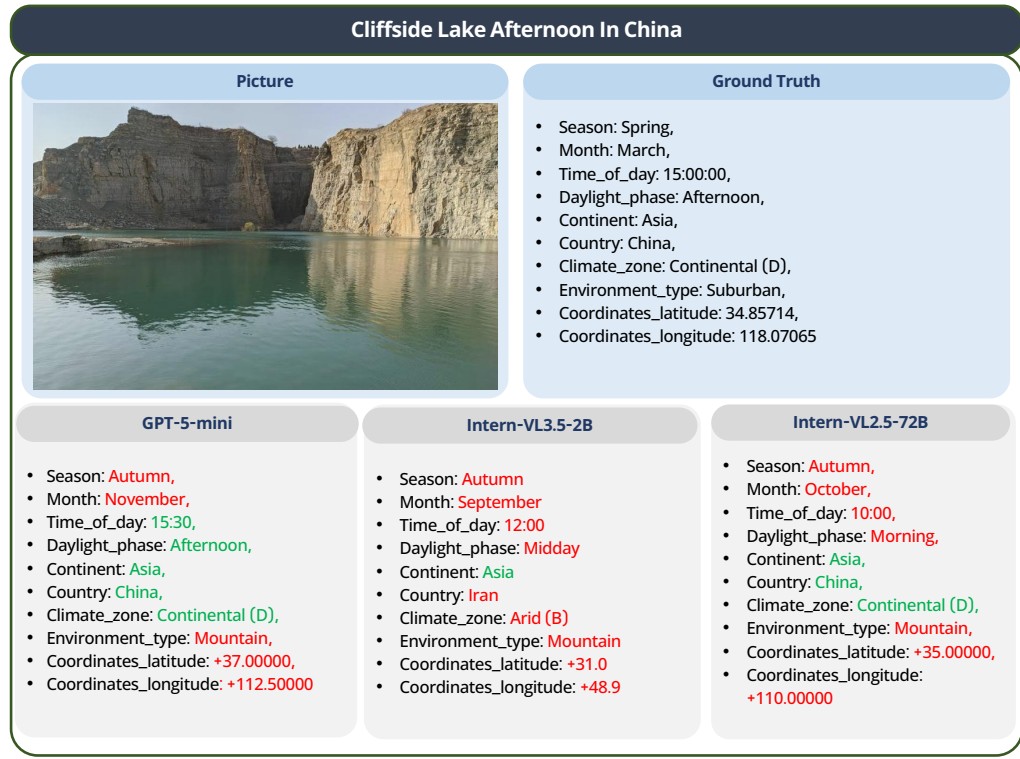

*Figure 21.* Example of TIMESPOT dataset: Cliffside Lake Afternoon in China.

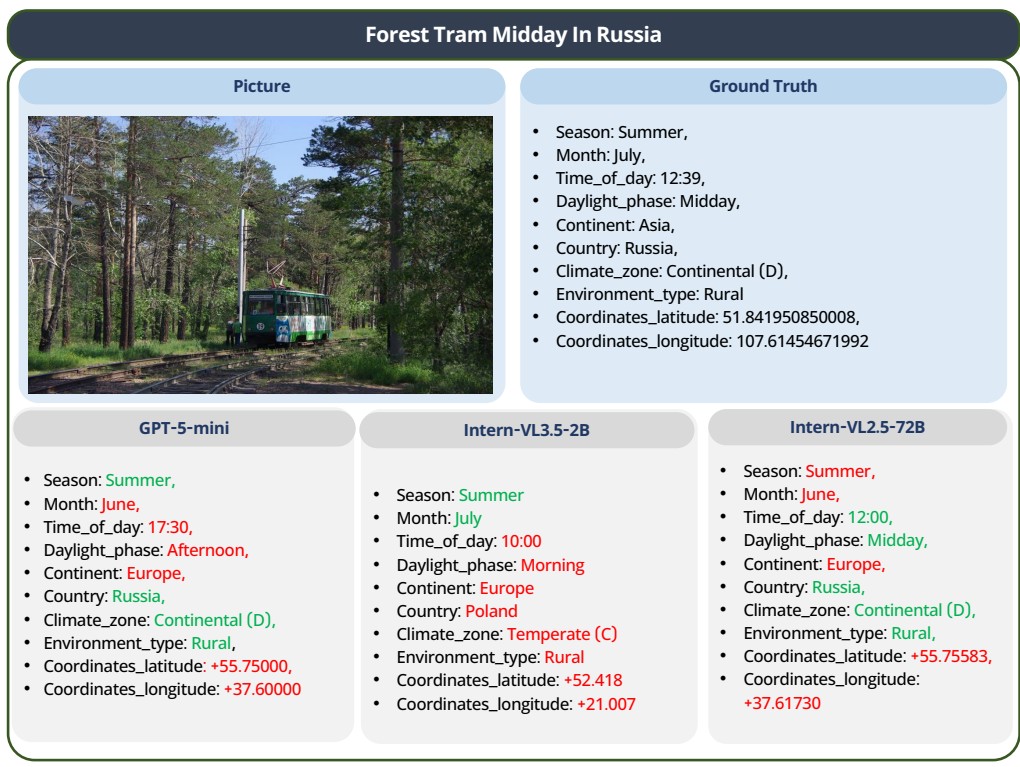

*Figure 22.* Example of TIMESPOT dataset: Forest Tram Midday in Russia.

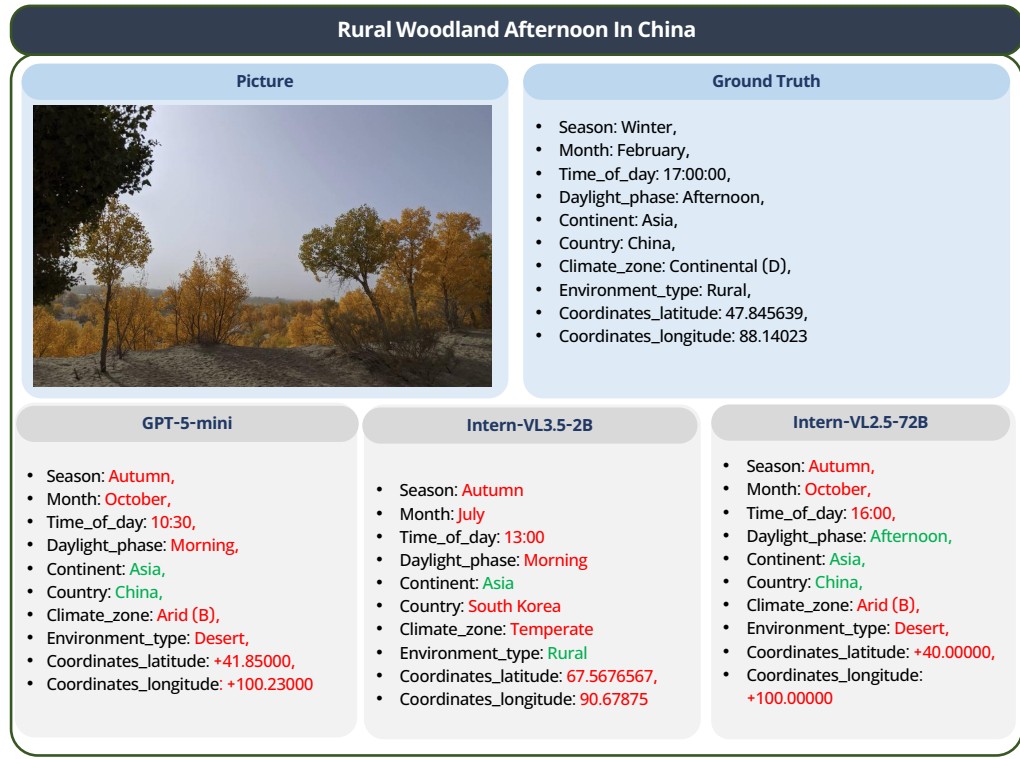

*Figure 23.* Example of TIMESPOT dataset: Rural Woodland Afternoon in China.

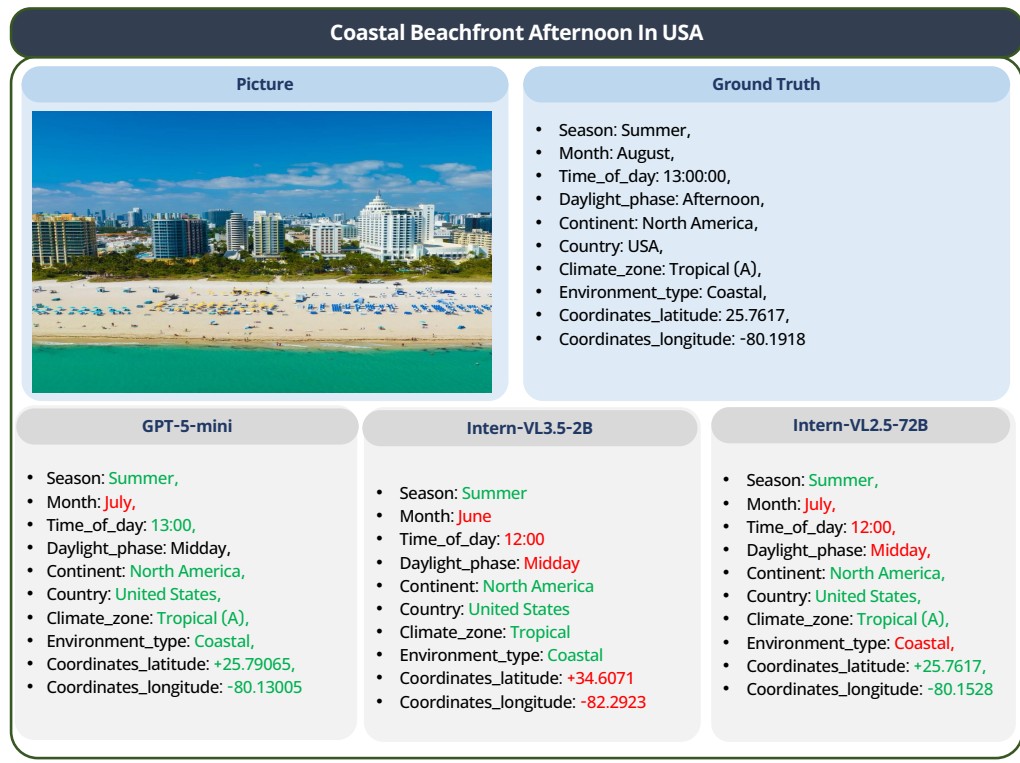

*Figure 24.* Example of TIMESPOT dataset: Coastal Beachfront Afternoon in USA.

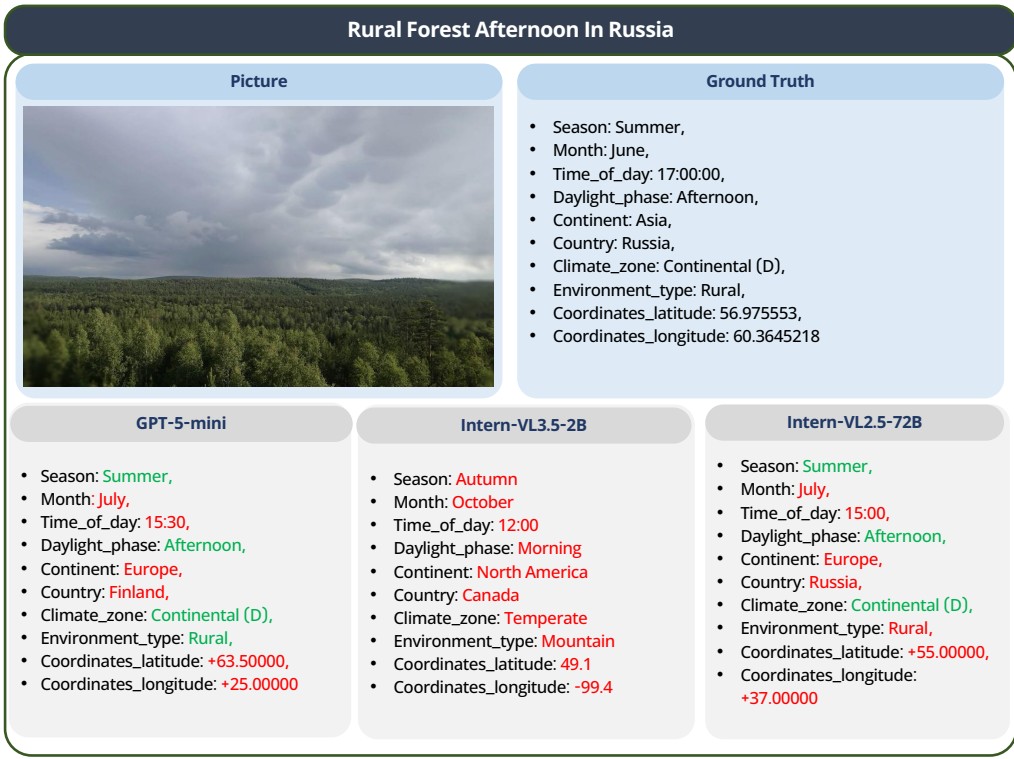

*Figure 25.* Example of TIMESPOT dataset: Rural Forest Afternoon in Russia.

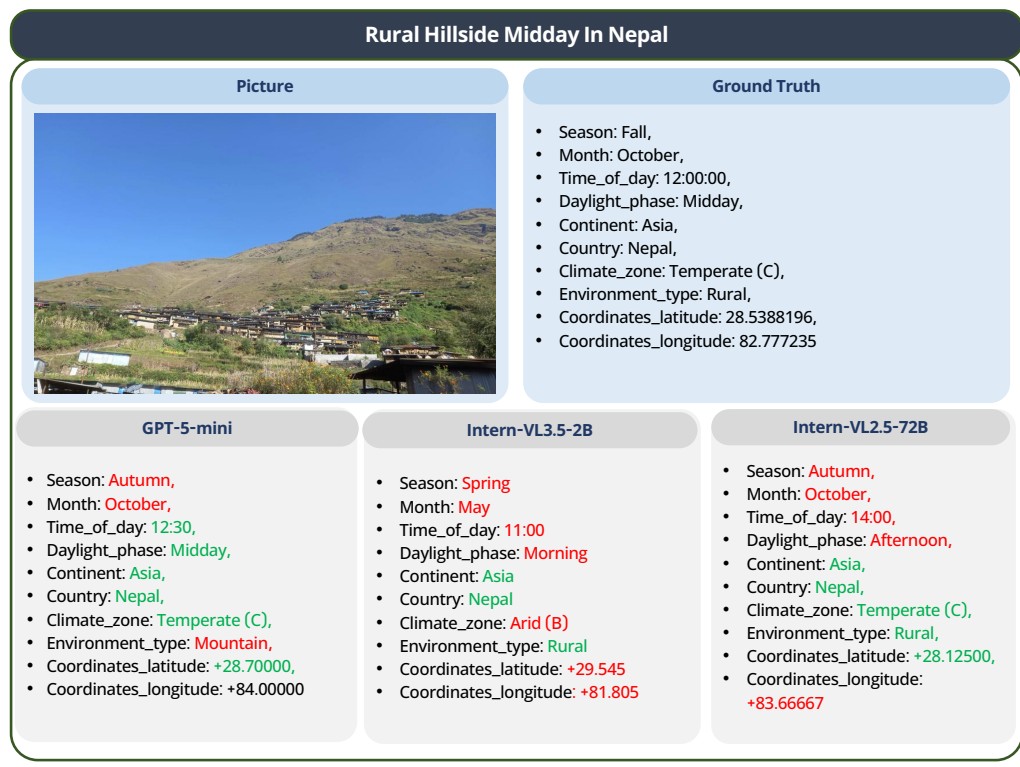

*Figure 26.* Example of TIMESPOT dataset: Rural Hillside Midday in Nepal.

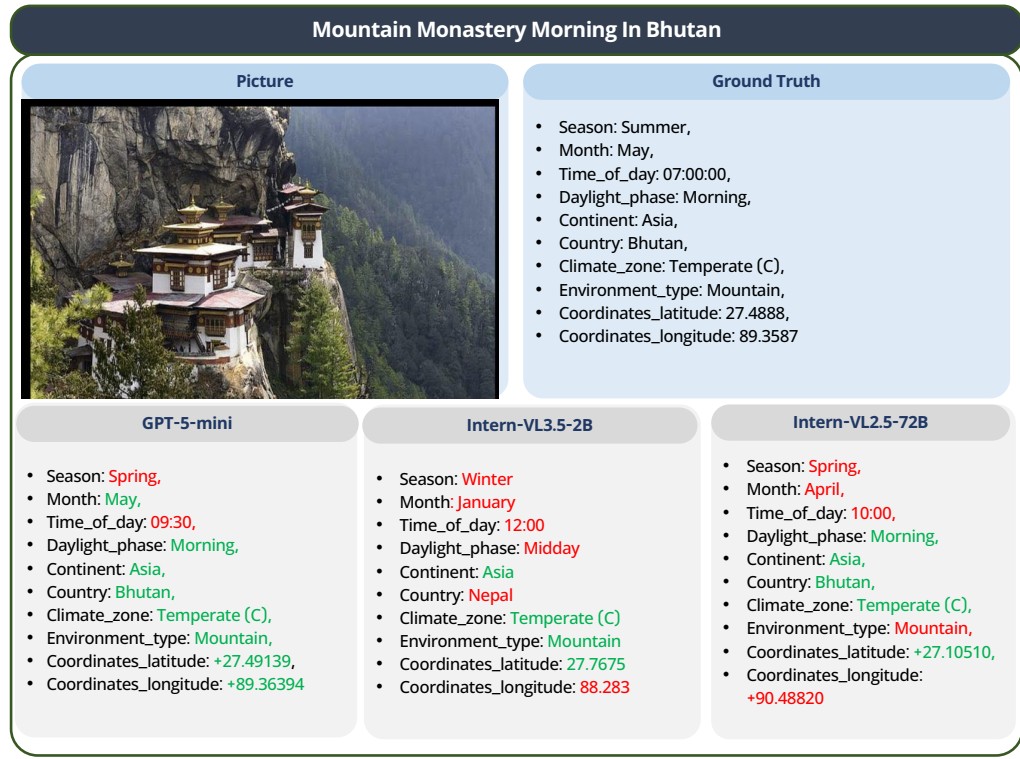

*Figure 27.* Example of TIMESPOT dataset: Mountain Monastery Morning in Bhutan.

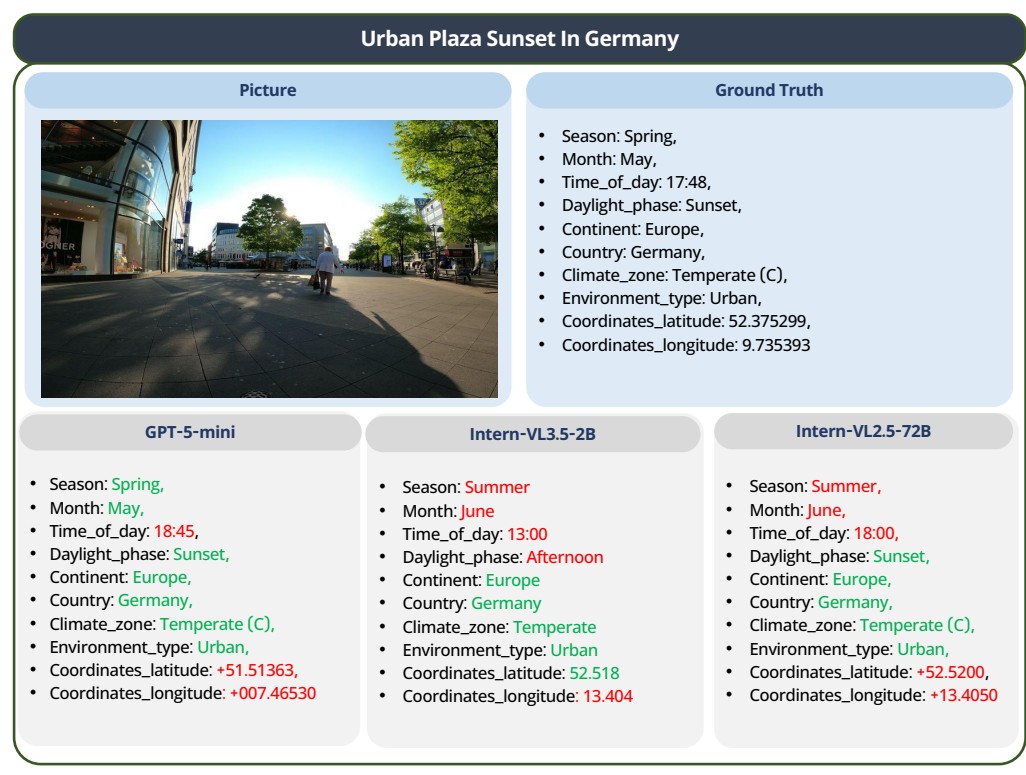

*Figure 28.* Example of TIMESPOT dataset: Urban Plaza Sunset in Germany.

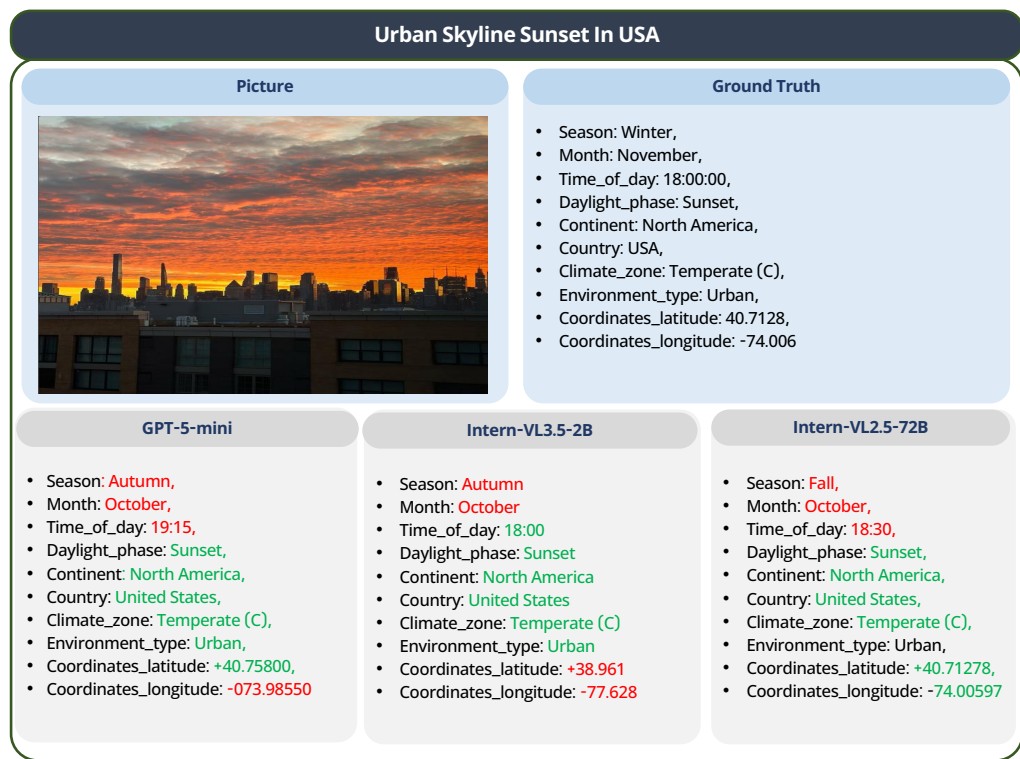

*Figure 29.* Example of TIMESPOT dataset: Urban Skyline Sunset in USA.

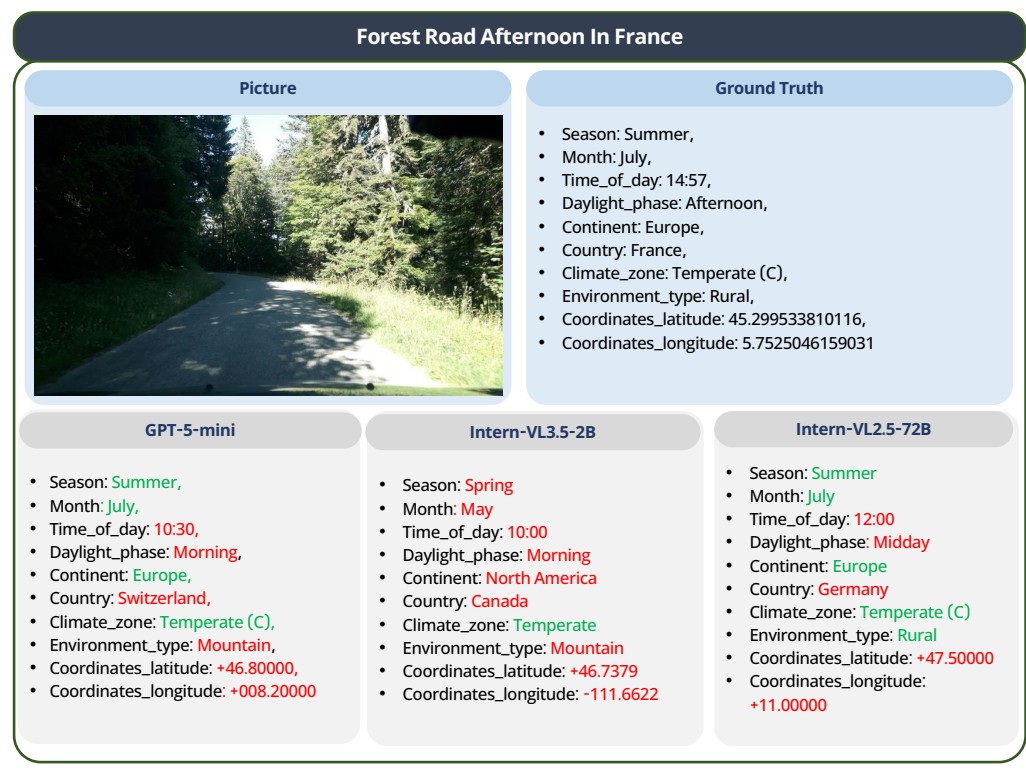

*Figure 30.* Example of TIMESPOT dataset: Forest Road Afternoon in France.

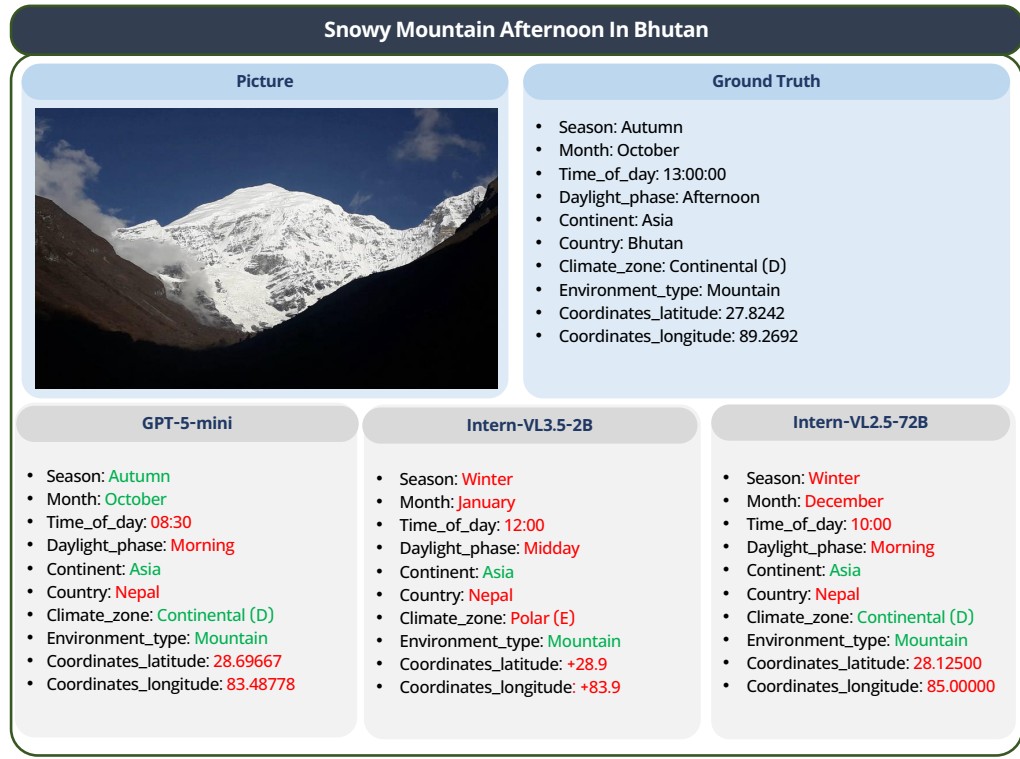

*Figure 31.* Example of TIMESPOT dataset: Snowy Mountain Afternoon in Bhutan.

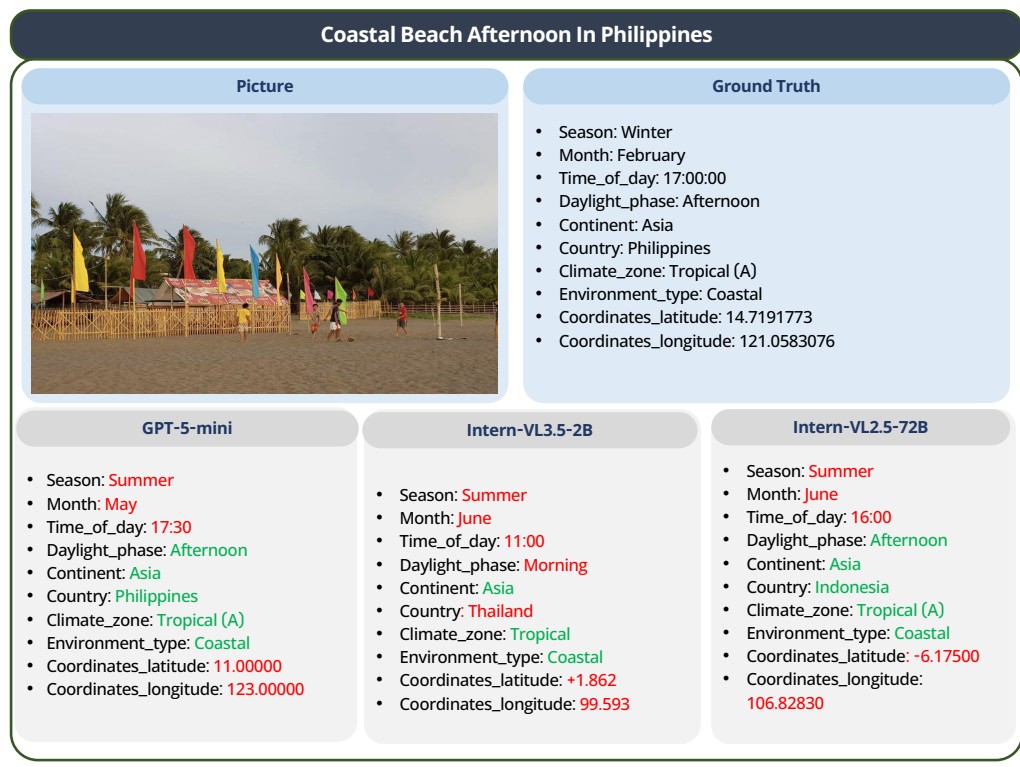

*Figure 32.* Example of TIMESPOT dataset: Coastal Beach Afternoon in Philippines.

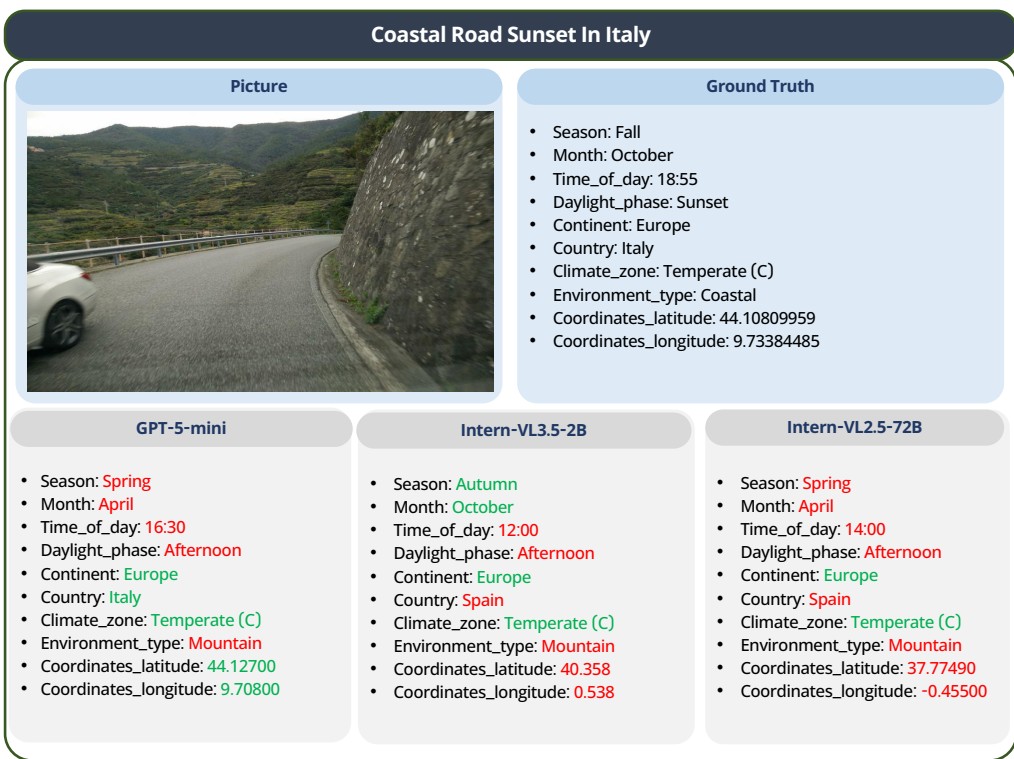

*Figure 33.* Example of TIMESPOT dataset: Coastal Road Sunset in Italy.

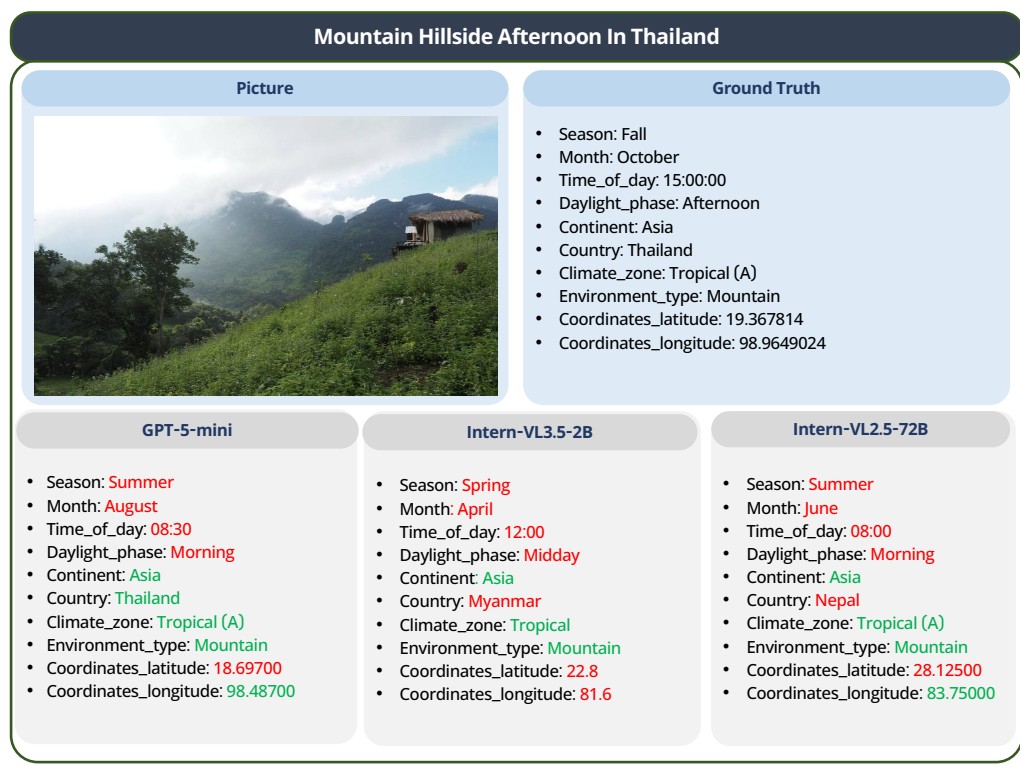

*Figure 34.* Example of TIMESPOT dataset: Mountain Hillside Afternoon in Thailand.

## L. Examples of JSON record with Response Analysis

Two such examples from the JSON record are given below:

```
Example: 1
{
    "image_file": "1eda0bc6-d820-440f-8c71-f0289d56de1c.jpg",
    "ground_truth": {
      "season": "Summer",
      "month": "June",
      "time_of_day": "21:36",
      "daylight_phase": "Night",
      "continent": "Europe",
      "country": "Poland",
      "climate_zone": "Continental (D)",
      "environment_type": "Urban",
      "coordinates_latitude": "53.0533603",
      "coordinates_longitude": "16.7307758"
    },
    "raw_model_answer": "- season : Summer\n- month : July\n- time_of_day : 11:00\n-
        daylight_phase : Morning\n- continent : Europe\n- country : Poland\n- climate_zone
        : Continental\n- environment_type : Suburban\n- coordinates_latitude : 52.25000\n
        - coordinates_longitude : 19.00000",
    "model_parsed": {
      "season": "Summer",
      "month": "July",
      "time_of_day": "11:00",
      "daylight_phase": "Morning",
      "continent": "Europe",
      "country": "Poland",
      "climate_zone": "Continental",
      "environment_type": "Suburban",
      "coordinates_latitude": "52.25000",
      "coordinates_longitude": "19.00000"
    },
    "judge_output_raw": "json\n{\n  \"season\": {\n    \"ground_truth\": \"Summer\",\n
        \"model_ans\": \"Summer\",\n    \"evaluation\": 1  },\n  \"month\": {\n    \"
        ground_truth\": \"June\",\n    \"model_ans\": \"July\",\n    \"evaluation\": 0\n
        },\n  \"time_of_day\": {\n    \"ground_truth\": \"21:36\",\n    \"model_ans\":
        \"11:00\",\n    \"evaluation\": 0\n  },\n  \"daylight_phase\": {\n    \"
        ground_truth\": \"Night\",\n    \"model_ans\": \"Morning\",\n    \"evaluation\":
        0\n  },\n  \"continent\": {\n    \"ground_truth\": \"Europe\",\n    \"model_ans\":
        \"Europe\",\n    \"evaluation\": 1\n  },\n  \"country\": {\n    \"ground_truth\":
        \"Poland\",\n    \"model_ans\": \"Poland\",\n    \"evaluation\": 1\n  },\n  \"
        climate_zone\": {\n    \"ground_truth\": \"Continental (D)\",\n    \"model_ans\":
        \"Continental\",\n    \"evaluation\": 1\n  },\n  \"environment_type\": {\n    \"
        ground_truth\": \"Urban\",\n    \"model_ans\": \"Suburban\",\n    \"evaluation\":
        0\n  },\n  \"coordinates_latitude\": {\n    \"ground_truth\": \"53.0533603\",\n
        \"model_ans\": \"52.25000\",\n    \"evaluation\": 0\n  },\n  \"
        coordinates_longitude\": {\n    \"ground_truth\": \"16.7307758\",\n    \"model_ans
        \": \"19.00000\",\n    \"evaluation\": 0\n  }\n}\n```",
    "judge_output_parsed": {
      "season": {
        "ground_truth": "Summer",
        "model_ans": "Summer",
        "evaluation": 1
      },
      "month": {
        "ground_truth": "June",
        "model_ans": "July",
        "evaluation": 0
      },
      "time_of_day": {
        "ground_truth": "21:36",
        "model_ans": "11:00",
```

```
            "evaluation": 0
          },
          "daylight_phase": {
            "ground_truth": "Night",
            "model_ans": "Morning",
            "evaluation": 0
          },
          "continent": {
            "ground_truth": "Europe",
            "model_ans": "Europe",
            "evaluation": 1
          },
          "country": {
            "ground_truth": "Poland",
            "model_ans": "Poland",
            "evaluation": 1
          },
          "climate_zone": {
            "ground_truth": "Continental (D)",
            "model_ans": "Continental",
            "evaluation": 1
          },
          "environment_type": {
            "ground_truth": "Urban",
            "model_ans": "Suburban",
            "evaluation": 0
          },
          "coordinates_latitude": {
            "ground_truth": "53.0533603",
            "model_ans": "52.25000",
            "evaluation": 0
          },
          "coordinates_longitude": {
            "ground_truth": "16.7307758",
            "model_ans": "19.00000",
            "evaluation": 0
          }
        },
        "metrics": {
          "daylight_abs_error_minutes": 636,
          "daylight_abs_error_hm": "10:36",
          "daylight_within_1h": 0,
          "lat_abs_error_deg": 0.8033603000000014,
          "lon_abs_error_deg": 2.2692242,
          "latlon_distance_m": 177222.27432500423
        }
      },
Example 2:
  {
    "image_file": "ffc56015-feb8-49db-b74c-4fc681fee5d9.jpg",
    "ground_truth": {
      "season": "Spring",
      "month": "April",
      "time_of_day": "11:00:00",
      "daylight_phase": "Morning",
      "continent": "Asia",
      "country": "Bangladesh",
      "climate_zone": "Tropical (A)",
      "environment_type": "Suburban",
      "coordinates_latitude": "23.8191",
      "coordinates_longitude": "90.4526"
    },
    "raw_model_answer": "- season : Summer\n- month : July\n- time_of_day : 11:30\n-
        daylight_phase : Midday\n- continent : Asia\n- country : Singapore\n- climate_zone
```

```
            : A\n- environment_type : Urban\n- coordinates_latitude : +1.35208\n-
      coordinates_longitude : +103.81983",
  "model_parsed": {
    "season": "Summer",
    "month": "July",
    "time_of_day": "11:30",
    "daylight_phase": "Midday",
    "continent": "Asia",
    "country": "Singapore",
    "climate_zone": "A",
    "environment_type": "Urban",
    "coordinates_latitude": "+1.35208",
    "coordinates_longitude": "+103.81983"
  },
  "judge_output_raw": "```json\n{\n  \"season\": {\n      \"ground_truth\": \"Spring\",\n
        \"model_ans\": \"Summer\",\n      \"evaluation\": 0\n  },\n  \"month\": {\n      \"
      ground_truth\": \"April\",\n      \"model_ans\": \"July\",\n      \"evaluation\": 0\n
      },\n  \"time_of_day\": {\n      \"ground_truth\": \"11:00:00\",\n      \"model_ans\":
      \"11:30\",\n      \"evaluation\": 0\n  },\n  \"daylight_phase\": {\n      \"
      ground_truth\": \"Morning\",\n      \"model_ans\": \"Midday\",\n      \"evaluation\":
      0\n  },\n  \"continent\": {\n      \"ground_truth\": \"Asia\",\n      \"model_ans\":
      \"Asia\",\n      \"evaluation\": 1\n  },\n  \"country\": {\n      \"ground_truth\": \"
      Bangladesh\",\n      \"model_ans\": \"Singapore\",\n      \"evaluation\": 0\n  },\n
      \"climate_zone\": {\n      \"ground_truth\": \"Tropical (A)\",\n      \"model_ans\":
      \"A\",\n      \"evaluation\": 1\n  },\n  \"environment_type\": {\n      \"ground_truth
      \": \"Suburban\",\n      \"model_ans\": \"Urban\",\n      \"evaluation\": 0\n  },\n
      \"coordinates_latitude\": {\n      \"ground_truth\": \"23.8191\",\n      \"model_ans
      \": \"+1.35208\",\n      \"evaluation\": 0\n  },\n  \"coordinates_longitude\": {\n
        \"ground_truth\": \"90.4526\",\n      \"model_ans\": \"+103.81983\",\n      \"
      evaluation\": 0\n  }\n}\n```",
  "judge_output_parsed": {
    "season": {
      "ground_truth": "Spring",
      "model_ans": "Summer",
      "evaluation": 0
    },
    "month": {
      "ground_truth": "April",
      "model_ans": "July",
      "evaluation": 0
    },
    "time_of_day": {
      "ground_truth": "11:00:00",
      "model_ans": "11:30",
      "evaluation": 0
    },
    "daylight_phase": {
      "ground_truth": "Morning",
      "model_ans": "Midday",
      "evaluation": 0
    },
    "continent": {
      "ground_truth": "Asia",
      "model_ans": "Asia",
      "evaluation": 1
    },
    "country": {
      "ground_truth": "Bangladesh",
      "model_ans": "Singapore",
      "evaluation": 0
    },
    "climate_zone": {
      "ground_truth": "Tropical (A)",
      "model_ans": "A",
      "evaluation": 1
```

```
        },
        "environment_type": {
          "ground_truth": "Suburban",
          "model_ans": "Urban",
          "evaluation": 0
        },
        "coordinates_latitude": {
          "ground_truth": "23.8191",
          "model_ans": "+1.35208",
          "evaluation": 0
        },
        "coordinates_longitude": {
          "ground_truth": "90.4526",
          "model_ans": "+103.81983",
          "evaluation": 0
        }
      },
      "metrics": {
        "daylight_abs_error_minutes": 30,
        "daylight_abs_error_hm": "0:30",
        "daylight_within_1h": 1,
        "lat_abs_error_deg": 22.467019999999998,
        "lon_abs_error_deg": 13.367229999999992,
        "latlon_distance_m": 2883380.4407346263
      }
    },
```

