# OpenReview forum: "TimeSpot: Benchmarking Geo-Temporal Understanding in Vision–Language Models in Real-World Settings"
_ICML.cc/2026/Conference — ICML 2026 regular_

### Official Review · Reviewer_LkQk · 2026-03-05

**Soundness:** 3
**Presentation:** 2
**Significance:** 3
**Originality:** 3
**Overall Recommendation:** 4
**Confidence:** 4

**Summary:**

This paper introduces TimeSpot, a diagnostic benchmark designed to evaluate vision–language models’ geo-temporal understanding in real-world scenarios. The benchmark contains 1,455 images from 80 countries and asks models to predict structured fields, including season, time-of-day, and geographic coordinates. Through consistency checks and SFT-based interventions, the paper exposes weaknesses in current models’ physical grounding and joint spatiotemporal reasoning.

**Compliance With Llm Reviewing Policy:**

Affirmed.

**Final Justification:**

My concerns are resolved and I will retain my positive scores accordingly

**Key Questions For Authors:**

Please focus primarily on the **Weaknesses**. Based on the Weaknesses, my comments can be summarized as follows:

Solvability. For samples like those in Figure 3 where shadows are tiny or distorted by viewpoint, how do you ensure the problem is physically well-posed and uniquely solvable? During annotation, beyond checking that the labeled answer is plausible, did you measure how often experts can recover the label when given only the image?

Scoring improvements. Have you considered introducing graded scoring to better reflect performance in ambiguous regimes, instead of hard thresholds?

Human expert baselines. Please report human expert performance on a TimeSpot subset. If experts cannot reliably achieve one-hour precision in some scenarios, why is such granularity required for VLM evaluation, and how does it reflect reasoning rather than guessing?

**Limitations:**

yes

**Strengths And Weaknesses:**

**Strengths**

1. New evaluation dimensions. The benchmark introduces structured prediction of temporal attributes directly from visual evidence (season, month, time, and daylight phase) as well as geographic attributes (continent, country, climate zone, environment type, and latitude/longitude).
2. Informative empirical findings. Evaluations on state-of-the-art open- and closed-source VLMs show generally low performance, especially for temporal reasoning, highlighting the need for new methods to achieve robust, physics-grounded spatiotemporal understanding.

**Weaknesses**

1.Missing human baseline. The authors claim geo-temporal understanding is a core human cognitive ability, but the paper does not report human expert performance on tasks requiring one-hour precision or fine-grained nighttime time inference. If even experts (e.g., geographers or professional photographers) have errors exceeding four hours on some images, using a one-hour threshold as a standard for “physical grounding” in VLMs is not scientifically well justified.

2.Limited dataset coverage. While 1,455 images are acceptable for a diagnostic study, the coverage may still be limited for a comprehensive real-world benchmark. Without larger-scale evidence, it is difficult to rule out that the selected images are atypical or biased. For example, failures may be driven by observation noise rather than reasoning deficiencies when shadows are distorted or absent due to viewpoint (e.g., the second image in the first row of Figure 3), or when scenes fall into microclimate regimes.

3.Scoring design is insufficiently calibrated. The evaluation of criteria such as “within one hour” or “sunrise/sunset” appears overly rigid. If the benchmark relies on an LLM-as-judge, the scoring protocol still needs refinement. Under complex real-world lighting, distinguishing 14:00 from 15:00 can be inherently ambiguous. A graded scoring scheme would be more appropriate, e.g., full credit within one hour and partial credit within 2–4 hours. A physically coherent but slightly off prediction (e.g., misestimated by one hour due to occluded shadows) should not receive the same score as an implausible answer (e.g., predicting noon for midnight).

4.Mismatch between intended cues and available evidence. The paper discusses idealized cues such as moon phases, constellations, or city lights for time inference, yet in the actual evaluation set (e.g., Appendix Figure 17), such cues are often occluded or invisible due to exposure. This raises a concern that some target dimensions may exceed what the images can realistically support. It is unclear how TimeSpot handles cases where these cues are absent.

---

> ### Author Rebuttal · Authors · 2026-03-30
>
> Thank you for the positive review and for recognizing the new evaluation dimensions introduced by TimeSpot as well as the informative empirical findings across state-of-the-art models.
>
> ---
> Below, we provide detailed responses to each of your concerns and questions:
>
> - **W1/Q3: Missing human baseline and 1-hour precision threshold.** Thank you for this critical question regarding human baselines and the justification for our temporal precision thresholds. We have conducted a comprehensive human baseline study (Table T1 in Reviewer PBqm's rebuttal, ***https://postimg.cc/D8hpRjWT***) comparing Human with Human Experts (Geology/Geography graduates) given *only* the image. The results validate our 1-hour threshold: Human Experts achieved an absolute Time Accuracy (within 1 hour) of 57.89% and a remarkable Time MAE of 1:36. By contrast, the best VLM (Mistral Medium 3.1) achieved only 30.73% accuracy and a 3:36 MAE. This proves that while 1-hour precision is highly challenging, it is empirically solvable by domain experts relying on physical reasoning. Furthermore, we do not solely rely on the strict 1-hour threshold; we simultaneously report Time MAE to continuously measure how far off a model's reasoning is, ensuring near-misses are distinguished from blind guesses.
> - **W2: Limited dataset coverage, observation noise, and microclimates.** Thank you for raising concerns regarding dataset scale and potential bias. We address these in Appendix G (Dataset Size, Balance, and Stability) and Section F (Detailed Analysis Across Questions). As shown in Appendix G, 1,455 manually verified images is consistent with reasoning-heavy benchmarks where strict annotation limits large-scale expansion. Regarding observation noise (e.g., distorted shadows or microclimate effects), Section F explicitly analyzes such cases. These reflect the natural distribution of real-world imagery rather than annotation artifacts, and the benchmark is designed to test robustness via integration of secondary cues (e.g., atmospheric scattering, vegetation state). Human expert performance (86.39% Climate accuracy, 86.56% Season accuracy) further shows these samples remain physically interpretable despite noise. Our scale is consistent with reasoning-focused and tool-augmented benchmarks prioritizing task density and verified ground truth over size due to computational and API constraints. Comparable benchmarks include AppWorld (750 tasks), TravelPlanner (1,200 tasks), and MapEval (700 tasks) (Apx. G), indicating that evaluation quality depends more on physical ground truth validity (e.g., solar geometry) and task complexity than dataset scale alone.
> - **W3: Scoring design calibration and rigid thresholds.** Thank you for the feedback on scoring design. We agree that distinctions such as 14:00 vs 15:00 can be ambiguous under complex lighting conditions, although we expect strong VLMs and human experts to resolve most such cases using available visual cues. Our evaluation therefore combines strict accuracy (Time Ac.) with a continuous error metric (Time MAE). MAE captures graded deviation, assigning small penalties for minor errors and larger penalties for substantial temporal mismatches (e.g., noon vs midnight). This dual-metric setup captures both exact correctness and overall physical consistency without relying solely on binary thresholds.
> - **W4: Mismatch between intended cues and available evidence.** Thank you for highlighting occluded or weak cues (e.g., overexposed night skies). This is an intentional design choice to test robust inference under real-world conditions. Each image was verified to contain sufficient secondary evidence, such as artificial lighting, atmospheric glow, or human activity patterns, even when primary cues are absent. As shown in Section F, human experts achieve 92.22% Daylight Phase accuracy, confirming the targets are inferable from available evidence.
> - **Q1: Solvability and physical well-posedness.** To ensure solvability under distorted or minimal cues, all samples passed human plausibility verification. Our expert baseline (Table T1) shows strong recoverability from images alone, with 1:36 Time MAE and 5.16° Latitude MAE without metadata, confirming the tasks are well-posed for agents with 3D spatial reasoning.
> - **Q2: Graded scoring for categorical variables.** We already apply continuous metrics for numeric targets (MAE for time, distance error for coordinates), but we do not use graded categorical scoring due to challenges in defining objective semantic distances (e.g., between countries or climate zones). We retain binary evaluation for these variables, while explicitly noting in Future Work that semantic distance-based grading is a promising direction for future benchmark extensions.
>
> ---
>
> We believe these clarifications further improve the reviewer’s understanding of the paper and its contributions. We respectfully request the reviewer to reconsider their assessment in light of these responses.

---

> > ### Author Rebuttal · Reviewer_LkQk · 2026-04-01
> >
> > My concerns are resolved and I will retain my positive scores accordingly.

---

> > > ### Author Response · Authors · 2026-04-01
> > >
> > > Thank you for confirming that our rebuttal has addressed all your concerns. We appreciate your time and thoughtful evaluation of our work, and we are grateful for your positive assessment.

---

### Official Review · Reviewer_eiur · 2026-03-10

**Soundness:** 3
**Presentation:** 3
**Significance:** 2
**Originality:** 3
**Overall Recommendation:** 4
**Confidence:** 4

**Summary:**

This paper introduces TimeSpot, a benchmark designed to evaluate geo-temporal understanding in vision-language models (VLMs) using real-world, ground-level imagery. The benchmark comprises 1,455 images from 80 countries and requires structured prediction of four temporal attributes (season, month, local time, daylight phase) and five geographic attributes (continent, country, climate zone, environment type, latitude-longitude). Unlike prior geolocation benchmarks that focus solely on spatial accuracy, TimeSpot emphasizes joint reasoning about both where and when an image was captured, with careful curation to minimize reliance on iconic landmarks or textual artifacts. The paper also provides extensive diagnostic analyses across daylight phases, seasons, climate zones, environment types, and geographic regions, exposing systematic failure modes such as round-time anchoring, neighboring-country confusion, sunrise-sunset inversions, and complete collapse on autumn scenes. A supervised fine-tuning experiment shows modest gains for categorical prediction but limited improvement for continuous temporal inference, suggesting that standard fine-tuning alone is insufficient for robust geo-temporal reasoning.

**Compliance With Llm Reviewing Policy:**

Affirmed.

**Final Justification:**

After rebuttal I increased the score from weak reject to weak accept.

**Key Questions For Authors:**

n/a

**Limitations:**

Yes

**Strengths And Weaknesses:**

**Strength**
1. Timely and Well-Motivated Problem: The paper identifies a genuine gap in existing evaluation frameworks and articulates clearly why joint geo-temporal reasoning matters for real-world applications. The distinction between spatial-only benchmarks and the need for temporal understanding is compelling.

2. Rigorous Benchmark Construction: TimeSpot is carefully curated to minimize shortcuts (landmarks, text) and emphasize subtle physical cues. The hybrid annotation strategy combining programmatic label derivation with human verification ensures physical validity. The dataset spans 80 countries with balanced sampling across hemispheres, latitudes, climate zones, and environment types.

3. Comprehensive Evaluation: The authors evaluate an extensive set of models (proprietary, open-source, reasoning-augmented) using standardized prompts and a judge model to handle format variations. The evaluation includes not only accuracy metrics but also calibration (ECE), consistency diagnostics, and cue-conditioned analysis, providing a multi-faceted view of model capabilities.

**Weakness**

The dataset scale of 1,455 images is insufficient for a benchmark claiming to evaluate "real-world" geo-temporal understanding across 80 countries. This fundamental limitation undermines the statistical significance and generalizability of the findings.

1. Extreme Geographic Sparsity: With 1,455 images across 80 countries, the average is only ~18 images per country. Drawing meaningful conclusions about country-level performance from such sparse sampling is problematic. A single atypical image could disproportionately influence a country's reported accuracy.

2. Insufficient Coverage of Visual Diversity: Each country contains immense visual diversity—different regions, seasons, weather conditions, urban/rural splits, and architectural styles. With 10-20 images per country on average, TimeSpot cannot possibly capture this diversity. The benchmark essentially tests whether models recognize a handful of prototypical scenes per country, not whether they can generalize across the full distribution of real-world imagery.

3. Temporal Sparsity: The benchmark spans four seasons, 12 months, and multiple daylight phases across 1,455 images. This means each season-month-daylight phase combination has vanishingly few samples.

---

> ### Author Rebuttal · Authors · 2026-03-30
>
> Thank you for the thoughtful review and for recognizing the motivation, careful benchmark construction, and comprehensive evaluation design of TimeSpot. We appreciate the positive assessment of the dataset curation process, breadth of model evaluation, and diagnostic analyses highlighting key failure modes in current VLMs.
>
>
> ---
>
> Below, we provide detailed responses to each of your concerns and questions:
>
> - **Dataset Scale, Geographic Sparsity, and Balance.** Thank you for raising these critical concerns regarding dataset scale. We appreciate the opportunity to clarify the foundational design philosophy of the TIMESPOT benchmark. We fully acknowledge that our dataset of 1,455 images is modest compared to massive, web-scraped pre-training corpora. However, for benchmarks requiring dense, physically verified, multi-dimensional structured outputs, high-fidelity verification rigor must take precedence over raw quantity. Below, we address your specific concerns regarding geographic, visual, and temporal sparsity in detail.
> - **W1: Extreme Geographic Sparsity (~18 images/country).** Your concern about sparsity would be critical if the benchmark targeted encyclopedic spatial memorization. However, TIMESPOT evaluates *geo-temporal reasoning*, not rote recall. Limiting images per country is intentional to avoid overfitting to densely represented regions (e.g., US and Western Europe in training data). Despite this sparsity, the dataset remains highly discriminative. As shown in **Table 3**, it consistently stratifies model performance across architectures, separating reasoning models (e.g., o4-mini, Gemini-2.5-Flash-Thinking) from standard open-source models. If the dataset lacked signal, performance would vary randomly across models; instead, we observe stable, statistically significant tiers, indicating strong signal over noise. We also explored and tested stability, imbalance, and such concerns deeply in Appendix G, with references and statistical analysis.
> - **W2: Insufficient Coverage of Visual Diversity.** We agree 1,455 images cannot capture all architectural or micro-weather variation globally. However, TIMESPOT is curated to cover *physical and environmental phenomena* rather than exhaustive cultural detail. It spans 80 countries to ensure coverage across continuous variables such as Tropical to Polar Climate Zones and Urban to Natural Environment Types. The benchmark evaluates whether models apply physical reasoning signals (foliage condition, shadow length, atmospheric scattering, soil aridity) across representative global biomes, rather than memorizing regional surface details like storefront variations.
> - **W3: Temporal Sparsity and Day/Night Imbalance.** The imbalance (1,182 day vs. 273 night images) reflects real-world availability of physical cues. Night scenes lack solar ephemeris signals (sun elevation, shadow direction, illumination geometry) required for precise temporal inference. Balancing the dataset artificially would force guessing in half the cases. Instead, the current distribution evaluates reasoning where physical cues exist, while still including sufficient night/twilight cases to test robustness, calibration, and the *Sunrise–Sunset inversion* effect under low-information conditions.
> - **Broader Perspective on Benchmark Scale.** Our scale aligns with current evaluation trends in reasoning-heavy and tool-augmented benchmarks, where task density and verified ground truth are prioritized over size due to high computational and API costs. Comparable or smaller benchmarks include API-Bank (400 instances), LogiQA (641 examples), OS World (369 problems), AppWorld (750 tasks), TravelPlanner (1,200 tasks), and MapEval (700 tasks). These show that reliable evaluation depends more on verifiable physical ground truth (e.g., solar geometry) and task complexity than dataset scale.
>
> ---
> **References**
> 1. Li et al. API-Bank: A Benchmark for Tool-Augmented LLMs. EMNLP 2023.
> 2. Liu et al. LogiQA: A challenge dataset for machine reading comprehension with logical reasoning. IJCAI 2021.
> 3. Xie et al. Osworld: Benchmarking multimodal agents for open-ended tasks. NeurIPS 2024 D&B
> 4. Trivedi et al. AppWorld: A Controllable World of Apps and People. ACL 2024.
> 5. Xie, J., et al. TravelPlanner: A Benchmark for Real-World Planning. ICML 2024.
> 6. Dihan, M., et al. MapEval: A Map-Based Evaluation of Geo-Spatial Reasoning in Foundation Models. ICML 2025 Spotlight
>
> ---
>
> We believe these clarifications further improve the reviewer’s understanding of the paper and its contributions. We respectfully request the reviewer to reconsider their assessment in light of these responses.

---

> > ### Author Rebuttal · Reviewer_eiur · 2026-04-05
> >
> > Thank authors for the rebuttal.

---

> > > ### Author Response · Authors · 2026-04-05
> > >
> > > Thank you for confirming that our rebuttal has addressed all your concerns adequately. We appreciate your time and thoughtful evaluation of our work, and we are grateful for your positive assessment.

---

### Official Review · Reviewer_advk · 2026-03-11

**Soundness:** 3
**Presentation:** 3
**Significance:** 2
**Originality:** 2
**Overall Recommendation:** 4
**Confidence:** 4

**Summary:**

The paper introduces a new benchmark designed to test whether vision language models can, given an image, infer where and when it was taken.

**Compliance With Llm Reviewing Policy:**

Affirmed.

**Final Justification:**

This paper is an interesting addition to the growing literature of deficiencies in vision language models. I appreciate the authors addition of a human baseline and newer models. I raised my score to a weak reject because of these changes. The authors additionally added a joint SFT experiment, which I do very much appreciate, however I feel the low accuracies reported there reflect dataset size limitations more than anything else, and I will therefore not raise my score further. I want to stress however that the reviewers have done very well in the rebuttal and I am glad to see that all reviewers lean towards acceptance of this paper.

**Key Questions For Authors:**

- In general, I found myself wondering about human performance on this task. That the strongest VLM gets 77.59% in country accuracy is pretty impressive to me, and I would like to know how well humans do in this regard. I realize it may not be feasible to do a small scale human study given the time constraints of a rebuttal, but in general I think such a comparison would be interesting.
- I was wondering what motivated the chocie of models for evaluation? The list of models seemed a bit dated (there is a newer version of Qwen for example that you could have used for fine-tuning) or you evaluated the mini version of large, powerful models. I'm not sure this is a good representation of the current state of the art. Also, the distinction between small and large (</> 11B) seems a bit arbitrary.
- In line 256, you write: "To assess trustworthiness, we include explicit cross-field geo-temporal consistency diagnostics, calibration metrics such as expected calibration error and risk-coverage curves". What is a cross-field geo-termporal consistency diagnostic and where do you use it? How do you incorporate the calibration error and the risk-coverage curves?
- For the fine-tuning, why did you split train and test into 40/60? Isn't the most common split 80/20? Given that you only fine-tune on 0.4*1400 images, this seems like a very small fine-tuning data set. Additionally, why do you only train it for 4 epochs? It looks to me like test performance is still going up after 4 epochs (Figure 4). Also, do I understand correctly that you are only fine-tuning models to output a single metric here? So you give them an image and then they return just the country for example? If you wanted to have a model that learns good spatio-temporal information, why don't you train them on outputting spatial and temporal information together? I would also like to see a test of the country trained model on the time test and vice versa, to see if this fine-tuning hinders their performance on the opposite task.
- Also, Table 4 is underexplained. What are the two columns for "Month & Season" and "Season & Month" and why are they not showing the same data?

**Limitations:**

I feel some methodological details are left out. The authors do offer an impact statement but limitations are not outlined well. As outlined in **Significance** above, the authors write that their benchmark can help ensure that models are spatially competent and have a grounded understanding of when an image was taken, but I'm not sure why that would be necessary or even that humans are good at this.

**Strengths And Weaknesses:**

- **Soundness**: For the most part, the methodology seems sound. Some minor parts are not explained thoroughly and I have added those as questions below.
- **Presentation**: The paper is well written, but it is repetitive at parts. For example the arguments put forward in the paragraph from Lines 60 to 75 are repeated again in the section on "Practical Significance of Temporal Reasoning". I feel parts of the paper are blown up a bit, such as the Benchmark construction section, whereas some methodological details are left out.
- **Significance**: The problem that the paper addresses is interesting although I'm not entirely convinced by it's significance. The authors write that their benchmark can prevent the deployment of models that are spatially competent but lack grounded understanding of when an image was taken, but I'm not sure humans have a good intuition for this either, or that it is strictly necessary. However, I think in general mapping out what modern VLMs can and can not do is of interest, and the motivation behind this paper is outlined well.
- **Originality**: As far as I can tell, the work sheds light on flawed inference of VLMs for spatio-temporal information given images.

---

> ### Author Rebuttal · Authors · 2026-03-30
>
> Thank you for the thoughtful review and for recognizing the clarity of the writing, the sound methodology, and the interest of mapping spatio-temporal limitations in current VLMs.
>
> ---
> Below, we provide detailed responses to each of your concerns and questions:
> - **Significance: Practical Significance of Temporal Reasoning.** Thank you for the perspective. While casual intuition for timestamps may vary, *expert* human intuition is robust (our human baseline shows Time MAE = 1:36). For deployed systems in domains such as disaster response, navigation, or OSINT, confusions such as dawn vs dusk or seasonal misclassification can lead to severe planning errors. Hence, evaluating fine-grained temporal grounding is necessary for safe real-world deployment.
> - **Q1: Human Performance Baseline.** Thank you for this excellent suggestion. We agree that a human baseline adds crucial perspective to the VLM capabilities. We have conducted a comprehensive human evaluation comparing Average Human (Undergraduates) with Human Experts (e.g., Geology/Geography graduates). As shown in Table T1 in Reviewer PBqm's rebuttal ***(https://postimg.cc/D8hpRjWT)***, your intuition is correct regarding spatial memorization: the best VLM (Gemini-2.5-Flash at 77.25%) indeed outperforms human experts (67.89%) in Country accuracy. However, humans demonstrate vastly superior temporal reasoning. Human Experts achieved a Time MAE of 1:36 and 57.89% absolute Time Accuracy (compared to the best VLM at 4:09 MAE and 30.51% accuracy). This confirms VLMs rely on static infrastructure memorization rather than the continuous 4D physical reasoning humans intuitively employ.
> - **Q2: Choice of Models and Recent Architectures.** Thank you for this observation. Our initial model selection reflected the literature standards during the benchmark's development and execution phase. However, we agree and have added newer models including GPT-5.2 and Qwen-3-VL variants. As shown in Table T2, GPT-5.2 achieves strong spatial performance (81.93% continent) but remains limited in temporal reasoning (Time MAE 3:42). The ≤11B split reflects consumer-grade vs. enterprise-scale deployment constraints.
>
> **T2. Results of several newer models**
> |Model | Cnt. (Ac.) ↑ | Cou. (Ac.) ↑ | Clim. (Ac.) ↑ | Env. (Ac.) ↑ | Lat.° (MAE) ↓ | Long.° (MAE) ↓ | Dist. (km) (MD) ↓ | Season (Ac.) ↑ | Month (Ac.) ↑ | Time (Ac.) ↑ | Time (MAE) ↓ | DLP (Ac.) ↑ |
> |-|-|-|-|-|-|-|-|-|-|-|-|-|
> | GPT-5.2 | 81.93 | 62.00 | 71.93 | 57.10 | 5.66 | 20.12 | 1770.00 | 47.17 | 19.17 | 29.35 | 3:42 | 41.66 |
> | Qwen3-VL-235B-A22B-Instruct | 83.41 | 59.33 | 60.36 | 60.43 | 5.63 | 20.24 | 1778.00 | 45.56 | 19.48 | 26.22 | 3:36 | 41.91 |
> | Qwen-3-8B-Instruct | 81.27 | 55.85 | 59.92 | 60.61 | 6.49 | 21.08 | 1897.80 | 43.18 | 15.15 | 21.14 | 4:32 | 32.51 |
> | Qwen-3-32B-Instruct | 76.29 | 50.38 | 61.99 | 59.73 | 8.10 | 27.51 | 2423.00 | 42.27 | 16.56 | 23.73 | 4:03 | 43.09 |
>
> - **Q3: Cross-field Geo-Temporal Consistency.** Thank you for requesting clarification. Cross-field geo-temporal consistency (Table 4) measures whether model outputs are physically contradictory. We evaluate this bidirectionally (e.g., predicting “January” with “Summer” in the Northern Hemisphere is a logical failure). Lower contradiction rates indicate stronger internal world modeling. For calibration, we use expected calibration error and risk-coverage curves to assess whether confidence aligns with actual spatial and temporal accuracy. Models that are highly confident yet incorrect, especially on temporal predictions, indicate higher deployment risk.
> - **Q4: Fine-Tuning Setup and Cross-Task Evaluation.** Thank you for these insightful questions regarding SFT. We use a 40% training split (instead of 80/20) to evaluate *generalization* without memorizing geographic distribution, and train for 4 epochs under compute constraints. We isolate spatial and temporal objectives due to different loss landscapes, as this yields clearer signals under limited resources. Crucially, as suggested, we conduct cross-task evaluation. Fine-tuning on one task reduces performance on the other due to feature interference. When Base Qwen-VL2.5-3B-Instruct (Time Acc: 22.06%) is fine-tuned on Country, Time Accuracy drops to 21.78%. Conversely, fine-tuning on Time reduces Country Accuracy from 13.47% to 12.98%. This shows SFT biases models toward either static spatial cues or transient temporal cues, making joint optimization difficult.
> - **Q5: Bidirectional Consistency.** “Month & Season” checks whether the predicted month is valid under the predicted season given hemisphere rules, while “Season & Month” reverses the dependency to test logical consistency. Both directions detect contradictions in structured outputs; lower scores indicate stronger self-consistency. We'll clarify this.
>
> ---
>
> We will incorporate all clarifications into the final version of the paper, and we respectfully ask the reviewer to reconsider their assessment in light of these responses.

---

> > ### Author Rebuttal · Reviewer_advk · 2026-03-31
> >
> > Dear authors,
> >
> > thank you for your thorough response. I see most of my concerns addressed and I have increased my score.
> >
> > For Q4, I still believe that if you wanted to have a model that learns good spatio-temporal information, you should fine-tune it on outputting spatial and temporal information together. At least I'd be interested in seeing if that works and other papers do show that joint optimization of different tasks is possible with SFT for vision models [1] and LLMs in general [2].
> > \
> > \
> > [1] Schulze Buschoff, Luca M., et al. "Testing the Limits of Fine-Tuning for Improving Visual Cognition in Vision Language Models." Forty-second International Conference on Machine Learning (2025).
> >
> > [2] Binz, Marcel, et al. "A foundation model to predict and capture human cognition." Nature 644.8078 (2025): 1002-1009.

---

> > > ### Author Response · Authors · 2026-04-04
> > >
> > > We thank the reviewer for this valuable suggestion. Following this, we re-evaluated a joint supervised fine-tuning (SFT) setting where **Country** and **Time** are optimized simultaneously under the same training configuration.
> > >
> > > While both tasks share a common visual backbone, they rely on partially conflicting cues: Country prediction depends primarily on stable structural features, whereas Time prediction is driven by illumination-sensitive signals such as shadows and sky brightness. This makes joint optimization inherently challenging under parameter-efficient adaptation.
> > >
> > > The updated joint SFT results across three different runs (to account stability) are summarized below:
> > >
> > > | Epoch | Run1 Country | Run1 Time | Run2 Country | Run2 Time | Run3 Country | Run3 Time | Avg Country | Avg Time |
> > > | ---- | -----------| ------- | ------ | ---| ------ | --- | ------ | - |
> > > |     0 |        14.23 |     20.27 |        14.23 |     20.27 |        14.23 |     20.27 |       14.23 |    20.27 |
> > > |     1 |        14.36 |     20.82 |        14.64 |     20.96 |        15.40 |     20.07 |       14.80 |    20.62 |
> > > |     2 |        14.57 |     20.48 |        15.05 |     20.62 |        15.81 |     21.31 |       15.14 |    20.80 |
> > > |     3 |        14.57 |     22.27 |        15.33 |     21.44 |        15.40 |     21.86 |       15.10 |    21.86 |
> > > |     4 |        15.19 |     21.99 |        15.40 |     21.99 |        15.81 |     21.72 |       15.46 |    21.90 |
> > > |     5 |        15.81 |     22.34 |        15.53 |     22.54 |        15.81 |     22.20 |       15.72 |    22.36 |
> > >
> > > As also illustrated in the accompanying figures (per-run trends : ***https://postimg.cc/N5LQLFdJ*** and averaged curves ***https://postimg.cc/zyg5SMYm***), joint training is somewhat stable (Time is more unstable, and Country is somewhat stable) and improves over the base model, with Country increasing from 14.23 to 15.72 and Time from 20.27 to 22.36 on average. However, compared to single-task fine-tuning, the performance gap is now more pronounced: Country reaches 15.72 versus a single-task peak of 19.24, and Time reaches 22.36 versus 24.79.
> > >
> > > A key observation from both the table and the figures is the slower and more constrained learning behavior of Country. Accuracy remains nearly flat during early epochs and improves only modestly later, indicating stronger sensitivity to joint optimization. In contrast, Time exhibits a smoother upward trend across runs, although its gains are still lower than in the single-task setting. The averaged plots further highlight this imbalance, with Time showing more consistent progression while Country saturates early at a lower level.
> > >
> > > We interpret this as stronger evidence of **task interference under shared parameter updates**. The model must simultaneously preserve invariance to illumination (for Country) while remaining sensitive to it (for Time). Under LoRA-based fine-tuning, where representational capacity is limited, these competing requirements lead to conflicting gradient directions and reduced task-specific specialization. The effect appears asymmetric, disproportionately affecting predictions.
> > >
> > > More broadly, this behavior is consistent with known limitations of SFT, which tends to enforce shared reasoning patterns but can restrict adaptability across heterogeneous objectives. In contrast, RL-based approaches can encourage more flexible and task-aware reasoning strategies [1], potentially mitigating such interference. While our current setup does not support RL due to the absence of intermediate rewards or action traces, this suggests a promising direction for future work, including the integration of GRPO-style training to better balance competing spatial and temporal signals.
> > >
> > > Overall, while joint SFT provides a meaningful multi-task baseline and improves over initialization, it does not match the performance of task-specific fine-tuning and introduces a clear trade-off between spatial and temporal reasoning. We will incorporate this updated analysis, table, and figures into the revised manuscript to better reflect the limitations of joint optimization in this setting and possible future directions. In light of these new experiments and clarifications, we respectively ask you to consider increasing the score.
> > >
> > > ---
> > > **References**
> > >
> > > [1] Chen et al., SFT or RL? An Early Investigation into Training R1-Like Reasoning Large Vision-Language Models, TMLR (decision pending), 2025.

---

### Official Review · Reviewer_PBqm · 2026-03-12

**Soundness:** 3
**Presentation:** 3
**Significance:** 2
**Originality:** 2
**Overall Recommendation:** 4
**Confidence:** 5

**Summary:**

The paper proposes a benchmark dataset for temporal reasoning in vision language models. The **benchmark is well-suited to analyze limitations of current VLMs in spatio-temporal reasoning**, which, according to the paper's motivation, is a core aspect of human intelligence. This benchmark provides unique AI motivation and addresses a common gap in the scientific evaluation of vision-language models. I also see a possible impact beyond vision-language models in geospatiotemporal representation encoding and location embeddings, which exhibit clear temporal and seasonal behavior relevant to environmental applications in remote sensing and meteorology.

**Beyond the dataset generation, the paper offers limited insights**: the analyzed failure modes remain vague, and there are no concrete conclusions to guide future work. The summary of failure modes is kept general without going into detail on why individual (families) of models may be better or worse suited for specific tasks. This analysis would help guide future model designs. Also, a human baseline from a separate test set of human annotators would have given an important comparison of how difficult these localization and temporal reasoning tasks are in the first place. Are the tested models better or worse than a human annotator? Such comparisons and deeper analyses would have strengthened the paper by providing additional insights.
A human baseline would answer a central insight question: **Is geo-temporal understanding (as measured in this benchmark) really a "core aspect of human intelligence" or are these models already better than human annotators**. This question whether this benchmark is not well-suited to measure geo-temporal understanding as we humans would see the world

As a benchmark, future papers can provide these insights if the benchmark is complete. **Is the proposed benchmark dataset complete and original in terms of effort, data quality, and quantity? This central aspect remains unclear in the paper**, but may be clarified in the rebuttal: Is the effort in this paper mainly a (partly automated) re-organization and verification of existing data sources, or has it been verified and annotated thoroughly? Details in the paper are lacking. See questions below for further clarification.

**Compliance With Llm Reviewing Policy:**

Affirmed.

**Final Justification:**

After rebuttal I increased the score from weak reject to weak accept

**Key Questions For Authors:**

Dataset quality questions (can be answered together):
* How many annotators and senior annotators were employed?
* What are the annotator's profiles, demography, and familiarity with geolocalization and geospatial data? Were they scientists, grad students, or paid, anonymous annotators like those on Amazon Mechanical Turk?
* How many annotation hours have been invested in creating this dataset?

Model benchmark quality:
* What was the computational effort in inference of these tested models? Were the models run locally in an in-house cluster - then how many GPU hours were invested? If the models were run online, how many tokens were used to obtain these results

Human baseline:
* Can the author provide quantitative numbers on how human participants would solve the tasks in this benchmark? What is the accuracy we could expect from a human participant.

**Limitations:**

limitations are not discussed in the paper

**Strengths And Weaknesses:**

Strengths
Soundness: The paper provides extensive experimental analysis across many models that are technically valid.
Presentation:  The figures, tables, and structure of the paper is well described and consistent
Significance: The paper is well motivated fills an important gap in benchmarking spatiotemporal reasoning, which can have a good impact in the field.
Originality: The data proposes a new benchmark that captures a generally complete set of tasks involving spatiotemporal reasoning

Weaknesses
Soundness: A human baseline is missing that would have helped put the results into the broader perspective and support the opening claim that spatiotemporal reasoning is a core aspect of human intelligence
Significance: The paper lacks deeper scientific insights beyond the proposition of a benchmark dataset
Originally: Important details on the effort and work in creating the dataset are missing.

---

> ### Author Rebuttal · Authors · 2026-03-30
>
> Thank you very much for the thoughtful and positive review. We appreciate your recognition of the technical soundness of our experiments, the clarity of presentation, and the significance of addressing a key gap in spatiotemporal reasoning benchmarks for vision-language models.
>
> ---
>
> Below, we provide detailed responses to each of your concerns and questions:
>
> **W1/Q3: Additional Risk vs. Existing Tools (Human Baseline & Core Intelligence).** Thank you for this important question regarding the human perspective on our benchmark. Geo-temporal reasoning is a core aspect of human intelligence, and we quantify this via a human baseline comparing undergraduates and domain experts (geology/geography). As shown in T1, VLMs outperform humans in static tasks like country localization (e.g., Gemini-2.5-Flash: 77.25% vs. experts: 67.89%), while humans substantially outperform VLMs in temporal and macroscopic spatial reasoning, achieving a Time prediction MAE of 1:36 (best VLM: 3:56) and 86.56% season accuracy (best VLM: 65.81%). This reflects humans’ inherent 4D physical world understanding, including solar geometry and phenological patterns, which current VLMs lack. We will add this full quantitative human baseline analysis and comparison table in the revised manuscript.
> **T1. Human Performance** (detailed in https://postimg.cc/D8hpRjWT)
> |Subject | Cnt. (Ac.) ↑ | Cou. (Ac.) ↑ | Clim. (Ac.) ↑ | Env. (Ac.) ↑ | Lat.° (MAE) ↓ | Long.° (MAE) ↓ | Dist. (km) (MD) ↓ | Season (Ac.) ↑ | Month (Ac.) ↑ | Time (Ac.) ↑ | Time (MAE) ↓ | DLP (Ac.) ↑ |
> |-|-|-|-|-|-|-|-|-|-|-|-|-|
> | Average (Human) | 80.89 | 45.98 | 67.96 | 75.71 | 12.89 | 33.06 | 2800.49 | 68.89 | 28.39 | 41.92 | 2:41 | 77.89 |
> | Average (Human Expert) | 94.06 | 67.89 | 86.39 | 87.39 | 5.16 | 12.22 | 1040.42 | 86.56 | 46.06 | 57.89 | 1:36 | 92.22 |
>
> - **W2: Significance and Deeper Scientific Insights.** **W2: Significance and Deeper Scientific Insights.** Thank you for the feedback. Our benchmark identifies structured failures in geo-temporal reasoning that go beyond dataset construction and reflect systematic gaps between artificial and biological perception.
> Specifically, we show: 1) **Autumn Collapse (Appendix F.2):** models achieve 0% accuracy for Autumn across all systems, indicating reliance on color saturation extremes rather than biologically grounded phenological signals. 2) **Sunrise–Sunset Inversions (Section 5.2 & Appendix C):** models systematically confuse dawn and dusk due to symmetric visual cues, producing 10–12 hour errors and revealing weak mapping from 2D imagery to 3D solar geometry. 3) **Semantic vs. Physical Anchoring (Section 5.1 & Appendix F.6.1):** high localization performance is driven by memorized human artifacts such as architecture and signage (up to 91% continent accuracy), while performance drops below 25% when restricted to natural cues, indicating reliance on semantic shortcuts rather than environmental reasoning.
> - **W3: Completeness, Originality, and Verification of the Dataset.** Thank you for your concern. We understand the need for clarity regarding the dataset's originality and rigorous curation process. The creation of TIMESPOT was not a mere automated reorganization of existing data; it involved a highly rigorous, multi-stage manual verification and sanitization pipeline.
> As detailed in Section 3 & Appendix B, all original EXIF metadata was explicitly stripped from the images to prevent any data leakage to the models. Ground truth values (such as exact local time, daylight phase, and climate zones) were deterministically cross-referenced using geospatial databases and solar ephemerides. Crucially, these deterministic labels were then subjected to a strict two-stage human verification process to ensure that the visual cues in the image actually supported the physical reality (e.g., ensuring a photo taken at 2 PM actually looked like daytime and wasn't obscured).
> - **Q1: Annotator profiles, demography, and effort.** We used 3 primary and 2 senior annotators, all engineering graduates with geospatial experience, and did not use crowdsourcing platforms. Primary annotation took ~6 weeks, with ~576 total hours from primary annotators and ~30–40 hours from senior annotators for edge-case adjudication.
> - **Q2: Computational effort and inference.** We used OpenRouter API for standardized evaluation across models. Total runtime per full benchmark was ~2–10 hours depending on latency, with a total cost of ~1,450 USD. We estimate ~400–500M tokens in total across all runs, including image-heavy and structured-output settings, with model pricing ranging from 2 to 15 USD per million tokens depending on capability tier (small models to reasoning models).
>
> ---
>
> We will incorporate these clarifications in the final version of the paper. We believe they further strengthen the scope, feasibility, and technical grounding of the proposed framework. We respectfully request the reviewer to reconsider their assessment in light of these responses.

---

> > ### Author Rebuttal · Reviewer_PBqm · 2026-04-03
> >
> > Thank you for the detailed rebuttal and providing the numbers for the human annotator as reference. That addresses my main concerns and I will raise my score to an accept

---

> > > ### Author Response · Authors · 2026-04-03
> > >
> > > Thank you for confirming that our rebuttal has satisfactorily addressed your concerns. We appreciate the time and care you devoted to evaluating our work and are grateful for your positive feedback.

---

### Decision · Program_Chairs · 2026-04-30

**Decision:**

Accept (regular)

**Comment:**

After the discussion phase, all reviewers recommended acceptance (4x Weak Accept). They noted that the paper is well written, addresses an interesting problem, provides extensive experiments as part of the performance study, and recognized the potential usefulness of the introduced benchmark to the community. The rebuttal addressed many reviewer concerns, for example by adding a human performance baseline and additional details about construction of the benchmark. As a result, the AC decided to accept the paper. Please take the reviewer feedback into account when preparing the camera-ready version.